


# Evaluating the transport of surface seawater from 1956 to 2021 using $^{137}$Cs deposited in the global ocean as a chemical tracer

Yayoi Inomata[1], Michio Aoyama[2]

5   1 Institute of Nature and Environmental Technology, Kanazawa University, Kanazawa, 920-1192, Japan

Center for Research in Isotopes and Environmental Dynamics, University of Tsukuba, Tsukuba, 305-8572, Japan

Correspondence to:Yayoi Inomata (yinomata@se.kanazawa-u.ac.jp)

**Abstract.** We analysed the spatiotemporal variations in the $^{137}$Cs activity concentrations in global ocean surface seawater from
1956 to 2021 using the HAMGlobal2021: Historical Artificial radioactivity database in Marine environment, Global integrated version 2021. The global ocean was divided into 37 boxes. The 0.5-yr average value of $^{137}$Cs in each box, except in the northern North Atlantic Ocean and its marginal sea, decreased exponentially in 1970–2010, immediately before the Fukushima Nuclear Power Plant (F1NPS) accident. The $^{137}$Cs inventory in the surface mixed layer in 1970 was estimated to be $187 \pm 26$ PBq. In 1975 and 1980, the $^{137}$Cs inventory increased to $201 \pm 28$ and $210 \pm 12$ PBq, respectively, due to direct discharge from the
Sellafield and La Hague nuclear fuel reprocessing plants. In 2011, the $^{137}$Cs inventory in the global ocean mixed layer increased to $48.1\pm12.1$ PBq compared to that before the F1NPS accident, in which the contribution from the accident was estimated to be approximately $15.5 \pm 3.9$ PBq. The distribution and variation in $^{137}$Cs in global surface seawater reflect basin-scale or global-scale transport. Mass balance analysis indicates that $^{137}$Cs deposited by the global fallout in the western North Pacific Ocean moves to the eastern North Pacific Ocean. Subsequently, $^{137}$Cs is transported southwards, followed by westwards
transport in the subtropical and equatorial Pacific Ocean and inflow into the Indian Ocean via the Indonesian Archipelago. The longer apparent half residence times in the Indonesian Archipelago (36.7 years from 1973 to 1997), South Atlantic Ocean (37.0 years from 1973 to 2004), and Central Atlantic Ocean (43.5 years from 1993 to 2016) also support the interpretation of the global-scale transport of $^{137}$Cs from the western North Pacific Ocean to the Indian and Atlantic Oceans. In the northern North Atlantic Ocean and its marginal sea, $^{137}$Cs discharged from nuclear reprocessing plants is transported to the North Sea,
Barents Sea and coast of Norway, and Arctic Ocean on a decadal scale. The dataset is available at http://dx.doi.org/10.34355/CRiED.U.Tsukuba.00085 (Aoyama, 2021), http://dx.doi.org/10.34355/Ki-net.KANAZAWA-U.00149 (Inomata and Aoyama, 2022a), http://dx.doi.org/10.34355/Ki-net.KANAZAWA-U.00150 (Inomata and Aoyama, 2022b), http://dx.doi.org/10.34355/Ki-net.KANAZAWA-U.00151 (Inomata and Aoyama, 2022c).





## 1 Introduction

With a half-life of 30.17 y, $^{137}$Cs is a major fission product that originates from large-scale atmospheric weapons tests. In addition to the large scale atmospheric weapons tests, $^{137}$Cs has been released into the environment from accidents such as the Chernobyl and Fukushima Daiichi Nuclear Power Plant (F1NPS), and planned discharges from nuclear reprocessing power plants. Because the $^{137}$Cs released into the atmosphere fallout onto the ocean surface, the ocean is recognized as the largest receptor of $^{137}$Cs on Earth. $^{137}$Cs is regarded as one of the most abundant artificial radionuclides in the ocean because of its

long half-life and large fission yield. The dominant sources of $^{137}$Cs have originated from global and local fallout due to atmospheric nuclear weapon tests by the United States and Russian Federation (e.g., UNCEAR 2000; Aoyama et al., 2006; Aoyama, 2010; Inomata, 2010), direct discharges from nuclear fuel reprocessing plants at Sellafield (United Kingdom) and La Hague (France) (e.g., UNSCARE, 2000; Aarkrog, 2003; Povinec et al., 2003; IAEA, 2005), the Chernobyl accident in 1986 (e.g., Molero et al., 1999; Steinhauser et al., 2014; Miyao et al., 1998), and direct release and atmospheric deposition from the

F1NPS accident in March 2011 (Aoyama et al., 2016a,b; Busseler et al., 2017). Furthermore, other sources, such as the accidental release from nuclear facilities (the Three Mile Island nuclear power plant in 1979), sea dumping of nuclear wastes from nuclear facilities carried out in 1986 in the north–central East Sea/Japan Sea by the former Soviet Union and Russian Federation, lost nuclear weapons, and the use of radioisotopes in human activities, such as industry, medicine, and science, are recognized. These contributions in the environment are minor compared to those from the dominant sources listed above

(UNSCARE, 2000; IAEA, 2005).

The largest source of $^{137}$Cs in the global ocean is from large-scale nuclear weapons tests. $^{137}$Cs was deposited in the western part of the North Pacific Ocean at approximately 30–45°N latitude and the western part of the North Atlantic Ocean at approximately 30–50°N latitude from the late 1950s to the early 1960s (UNSCARE, 2000; Aoyama et al., 2006). By using a dataset of $^{137}$Cs measurements in rainwater, seawater, and soil, $^{137}$Cs deposition estimated by global fallout in the Northern

Hemisphere with a 10°×10° grid was 765±79 PBq (Aoyama et al., 2006). The deposition of $^{137}$Cs in the Southern Hemisphere was significantly lower than that in the Northern Hemisphere. The maximum deposition of $^{137}$Cs in the Southern Hemisphere occurred approximately one year after that in the Northern Hemisphere, owing to the stratosphere air mass exchange between the northern and southern stratosphere (Hirose et al., 2003a, b). The largest local fallout of $^{137}$Cs released into the seawater occurred in the Bikini and Enewetak atolls in the Pacific Ocean (0-30°N) and was caused by the United States. Other local

fallouts had minor contributions to the $^{137}$Cs activity concentrations in the ocean (UNSCARE, 2000).

The liquid $^{137}$Cs discharged from nuclear reprocessing plants, namely, the Sellafield plant (Irish Sea, the United Kingdom) and the La Hague facility (English Channel coast, France), are also large sources. Since 1952, the discharge of several radionuclides from the Sellafield plant has occurred in the Irish Sea. The released $^{137}$Cs amount reached a maximum in the mid- to late 1970s (Gray et al., 1995; Guegueniat et al., 1997; UNSCARE, 2000). The discharged $^{137}$Cs from the Sellafield

plant from 1951 to 2020 was estimated to be 41.4 PBq (OSPAR, 2021). The maximum discharged $^{137}$Cs from the La Hague plant occurred in 1971, and the amount of discharged $^{137}$Cs decreased over time. The total amount of $^{137}$Cs released from the



La Hague plant was estimated to be 1.04 PBq (UNSCARE, 2000; IAEA2005). These discharged [137]Cs amounts were then transported to the northern North Atlantic Ocean, its marginal seas, and the Arctic Ocean (Smith et al., 1998; Maderich et al., 2021).

The Chernobyl accident on the 26[th] of April 1986, also released [137]Cs into the environment, which was estimated to be 100 PBq (WHO, 1989). Although most of the [137]Cs derived from the Chernobyl accident was deposited on land, a significant amount of [137]Cs was released into the ocean, which was estimated to be 16 PBq (UNSCARE, 2000). In particular, Chernobyl fallout increased the [137]Cs activity concentrations in the Baltic Sea (Zaborska et al., 2014), resulting in the most radioactive contaminated area, with 4.5 PBq of the total inventory (HELSINKI COMMUNICATION, 1995). The Black Sea also received

the Chernobyl [137]Cs fallout in 1986 (Bezhenar et al., 2019; Egorov et al. 1999), and the inventory was estimated to be 2-3 PBq (European Commission, 1994). The [137]Cs released into the Black Sea flowed into the Mediterranean Sea (Bezhenar et al., 2019). In the Mediterranean Sea, the total deposition of [137]Cs from the Chernobyl accident was estimated to be 2.5 PBq in 1986 (Delfanti and Papucci, 2010). Furthermore, the fallout of [137]Cs from the Chernobyl accident also occurred as a single small pulse in the western North Pacific Ocean and Japan Sea (Miyao et al., 1998: Inomata et al., 2009), and the Chernobyl

release contributed only a few percent compared to the previous [137]Cs water column inventory in these regions (Aoyama et al., 1986).

The F1NPS accident is also recognized as a large source of [137]Cs. On the 11[th] of March 2011, large amounts of [137]Cs were released into the western North Pacific Ocean by atmospheric deposition and direct discharge of liquid-contaminated stagnant water from the F1NPS because of the extraordinary earthquake and the subsequent giant tsunami (IAEA, 2015; UNSCEAR,

2013). The atmospheric deposition of [137]Cs into the ocean was estimated to be 11.7-14.8 PBq (Aoyama et al., 2016b). Directly discharged liquid [137]Cs from the F1NPS was estimated to be 3.6 ± 0.7 PBq (Tsumune et al., 2012, 2013). The [137]Cs inventory into the North Pacific Ocean was estimated to be 15.2-18.3 PBq (Aoyama et al., 2016b, Inomata et al., 2016; Tsubono et al., 2016). In addition, the region that contains F1NPS deposition in the western North Pacific Ocean is almost the same region as those with global fallout of [137]Cs in the 1950s and 1960s.

Other sources, such as nuclear waste dumping, discharges from nuclear power plant operation, the release of radionuclides from satellite failures, local underwater nuclear tests, nuclear weapons accidents, and the use in industry and medicine, are considered minor contributors to [137]Cs concentrations in the global ocean (Livingston and Povinec, 2000; IAEA, 2005).

[137]Cs exists mainly in the dissolved form, which allows it to move with seawater. After its release into the ocean, [137]Cs undergoes radioactive decay, with a half-life of 30.17 years, during transport into the ocean by undergoing oceanic physical

processes, such as advection and diffusion. Therefore, [137]Cs has been used as a marine tracer for decades to study physical processes, such as the long-range transport of water masses (e.g., IAEA, 2005; Hirose et al., 2003a,b; Inomata et al., 2009, 2012; Nakano et al., 2010; Povinec et al., 2003; Tsumune et al., 2003, 2011). Furthermore, [137]Cs is used to assess the radioactive doses (or radiological effects) to the human body due to the uptake of marine foods containing anthropogenic radionuclides (e.g., IAEA, 2005; UNSCARE, 2013).

The spatiotemporal variations in $^{137}$Cs activity concentrations in the surface seawater in the global ocean from 1956 to 2021 were investigated in this study. This study is an extension of our previous research, in which we analysed the measured data up to 2005 (Inomata et al., 2009). The data used in this study were adopted from the HAMGlobal2021: Historical Artificial radioactivity database in the Marine environment, Global integrated version 2021 (Aoyama, 2021), which contains data from the F1NPS accident. The HAMGlobal2021 database contains information on several radionuclides ($^{134}$Cs, $^{137}$Cs, $^{90}$Sr, $^3$H,

$^{239,240}$Pu, $^{241}$Am, and $^{14}$C) in the global ocean. The data were measured from 1956 to 2021.

In this paper, we present the following new insights:

1. The distribution of $^{137}$Cs in the mixed layer in the global ocean:

1) spatiotemporal variations in $^{137}$Cs activity concentrations in the surface seawater in the global ocean from 1956 to 2021;

2) spatiotemporal variations in the $^{137}$Cs inventory in the surface mixed layer in the global ocean based on the reconstructed

two-minute longitude/latitude $^{137}$Cs deposition data and surface mixed layer depths in each box; and

3) an estimate of the apparent half residence time (Tap) in each box in the global ocean.

2. Behaviour of $^{137}$Cs in the surface mixed layer in the global ocean:

4) a comparison between the amount of $^{137}$Cs deposited from the atmosphere due to global fallout and that directly released

from nuclear reprocessing plants; the behaviour of $^{137}$Cs released from nuclear reprocessing plants in the Atlantic to the Arctic; and

5) an estimate of the amount of $^{137}$Cs released from the F1NPS in 2011 by using the inventory in each box.

3. Mass balance:

6) an estimate of the $^{137}$Cs amount in horizontal transport and downwards transport below the surface mixed layer; and

7) finally, a summary of the behaviour of the $^{137}$Cs distribution in the global ocean.

**2 Data and methods**

**2.1 Data availability; HAM global 2021 database**

The $^{137}$Cs concentration data were obtained from the HAMGlobal2021: Historical Artificial radioactivity database in Marine environment, Global integrated version 2021; this database includes data measured from 1956 to 2021. These data are available at http://dx.doi.org/10.34355/CRiED.U.Tsukuba.00085 (Aoyama, 2021). The data were measured by many organizations and institutes, including the Baltic Marine Environment Protection (MORS, Germany); the Bundesamt fur Seeschiffapparent und Hydrographie (BSH, Germany); the Korean Institute of Nuclear Safety (KINS, Korea); the Marine

Ecology Research Institute (MERI, Japan); the Ministry of Agriculture, Fisheries and Food (MAFF, the United Kingdom); the Japan Coast Guard (JCG, Maritime Safety Agency until 2000, Japan); the Norwegian Radiation Protection Authority (NRPA, Norway); the Riso National Laboratory (RISO, Denmark); and the RPA V. G. Khlopin Radium Institute (VGKRI, Russia).





Data were also obtained from various research projects, such as the Arctic Monitoring Assessment Program (AMAP), the Geochemical Ocean Sections program (GEOSECS), the South Atlantic Ventilation Experiment (SAVE), Transit Tracers in the Ocean (TTO), the World Ocean Circulation Experiment (WOCE), and the Worldwide Marine Radioactivity Studies (WOMARS). Furthermore, we included the data reported in various research papers (Aarkrog et al., 1994; Aoyama and Hirose, 1995, 2004; Aoyama et al., 2001a,b; Aoyama et al., 2008, 2011, 2013, 2016a–c, 2018a,b; Ballestra et al., 1984; Bourlat et al., 1996; Bowen et al., 1982; Broecker et al., 1966, 1968; Busseler, 2012; Busseler et al., 2017; Cochran et al., 1987; Dahlgaard et al., 1995; Delfanti et al., 2000; Folsom et al., 1960a,b, 1968, 1970, 1975, 1979; Fowler et al., 1991; Gulin and Stokozov, 2005; Hirose et al., 1987, 1991, 1999; Hirose and Aoyama, 2003a,b; Ikeuchi et al., 1999; Ito et al., 2003, 2005; Kaeriyama et al., 2013, 2014, 2015; Kamenik et al., 2013; Katsuragi, unpublished data; Kautsky et al., 1987; Kim et al., 2012; Kumamoto et al., 2014, 2015, 2016, 2017, 2018, 2019; Livingston et al., 1985, 2000; Matishov et al., 2002; Miroshnichenko and Parasiv, 2020; Miyake et al., 1960, 1961, 1962, 1963, 1968, 1978, 1988; Miyao et al., 1998; Nagaya et al., 1964a,b, 1965, 1970, 1976, 1981, 1984, 1987, 1993; Nakanishi et al., 1990, 1995; Nies et al., 1989; Noshkin et al., 1976,1978,1979,1981,1999; Pillay et al., 1964; Povinec et al., 2003, 2011; Sanchez-Cabeza et al., 2011; Shirasawa et al., 1968; Smith et al., 1998, 2017; Wong et al., 1992; Yamada et al., 2006, 2007). Additionally, we utilized the database produced by the IAEA Marine Radioactivity Information System (MARIS). Finally, all these data were compiled into a single comprehensive database for this study. This new database contains a total of 54401 datapoints corresponding to $^{137}$Cs measured in surface waters (0–20 m) in the global ocean between 1956 and 2021. The $^{137}$Cs concentration data points used in this study were significantly larger than those employed in our previous research, which included 22,368 data points obtained until 2005 (Inomata et al., 2009).

In this study, the global ocean was divided into 37 boxes to investigate the temporal variations in $^{137}$Cs activity concentrations in surface seawaters (Inomata and Aoyama, 2022a) (Figure 1). These boxes were divided based on the known ocean currents (IAEA, 2005; Open University, 2004), latitudinal and longitudinal distributions, and the locations of global fallout, reprocessing plants, and nuclear power plants, and the assumptions were that the spatial distribution of $^{137}$Cs activity concentrations is uniform, sources of $^{137}$Cs are established, and horizontal and vertical transport of ocean water are almost the same (Hirose et al., 2003; Inomata et al., 2009; IAEA, 2005).

The 37 boxes set in this study are almost identical to those in our previous study (Inomata et al., 2009). However, the box areas are slightly modified by taking into account the oceanic physical parameters. The region in the South China Sea, which was in Box 33 in Inomata et al. (2009), is divided into two boxes: Boxes 33 (the South China Sea) and 35 (the Indonesian Archipelago). The region in Antarctica, which was in Box 13 in Inomata et al. (2009), is divided into three boxes: Boxes 13 (the Pacific sector of the Antarctic Ocean), 36 (the Atlantic sector of the Antarctic Ocean) and 37 (the Indian sector of the Antarctic Ocean). The region in the Southern Ocean (Box 17) is divided into two boxes: Boxes 17 (the Southern Ocean) and 19 (the middle Southern Ocean). The boxes corresponding to the source region, such as the Irish Sea (Box 23; Boxes 23.1-23.5) for the Sellafield plant and the northern North Atlantic Ocean (Box 25; Boxes 25.1 and 25.2) and western North Pacific Ocean (Box 2; Boxes 2.0-2.6) for the F1NPS accident, were divided into several subregions.

The box numbers, their corresponding longitudes and latitudes, the box name abbreviations, and the data number are listed in Table 1. The sampling points in each box are displayed in the data set (http://dx.doi.org/10.34355/Ki-net.KANAZAWA-U.00149, Inomata and Aoyama, 2022a). Temporal variations of $^{137}$Cs activity concentrations and 0.5-yr average values in each box are displayed in the Figures in the data set (http://dx.doi.org/10.34355/Ki-net.KANAZAWA-U.00150) in Inomata and Aoyama (2022b).

(a)

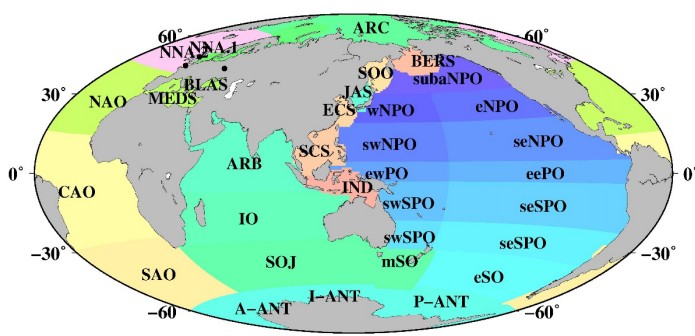

(b)                                                    (c)

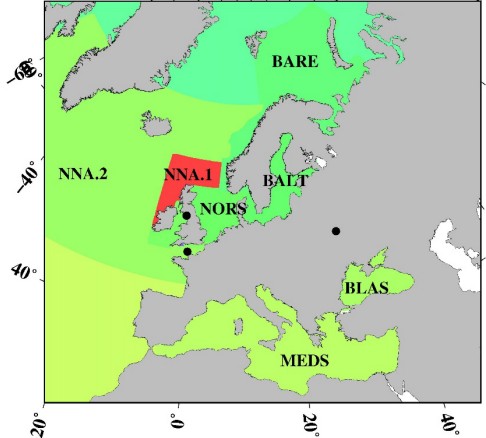
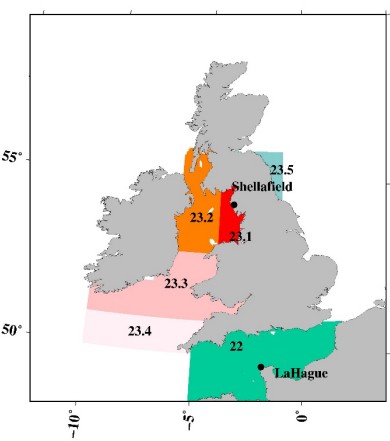

Figure 1: Boxes dividing the global ocean. (a)  Global, (b) North Atlantic Ocean and its marginal Sea, and (c) Irish Sea (Box23) and English Chanel (Box22).






Table 1. Detail of each box in the global ocean.



| Box | subbox | Area | Code | Lon_w | Lon_e | Lat_s | Lat_n | Data No |
|---|---|---|---|---|---|---|---|---|
| 1 | | subarctic North Pacific Ocean | subarctic NPO | 140.5 | 240 | 40 | 62 | 2182 |
| 2 | 0 | western North Pacific Ocean | western NPO | 128 | 180 | 25 | 27 | 3744 |
| 2 | 0 | western North Pacific Ocean | western NPO | 129 | 180 | 27 | 28 | |
| 2 | 0 | western North Pacific Ocean | western NPO | 130 | 180 | 28 | 31 | |
| 2 | 0 | western North Pacific Ocean | western NPO | 131 | 180 | 31 | 34 | |
| 2 | 0 | western North Pacific Ocean | western NPO | 132 | 180 | 34 | 35 | |
| 2 | 0 | western North Pacific Ocean | western NPO | 145 | 180 | 35 | 38.25 | |
| 2 | 0 | western North Pacific Ocean | western NPO | 141 | 180 | 38.25 | 40 | |
| 2 | 1 | western North Pacific Ocean | western NPO | 140 | 141.35 | 37.15 | 37.69 | 7466 |
| 2 | 2 | western North Pacific Ocean | western NPO | 139.5 | 141.75 | 37.69 | 38.25 | 2566 |
| 2 | 2 | western North Pacific Ocean | western NPO | 141.35 | 141.75 | 37.15 | 37.69 | |
| 2 | 2 | western North Pacific Ocean | western NPO | 140.5 | 141.75 | 36.85 | 37.15 | |
| 2 | 3 | western North Pacific Ocean | western NPO | 141.75 | 143.5 | 36.85 | 38.25 | 234 |
| 2 | 4 | western North Pacific Ocean | western NPO | 140.5 | 143.5 | 36 | 36.85 | 1365 |
| 2 | 5 | western North Pacific Ocean | western NPO | 143.5 | 145 | 36 | 38.25 | 89 |
| 2 | 6 | western North Pacific Ocean | western NPO | 140 | 145 | 35 | 36 | 2682 |
| 3 | | eastern North Pacific Ocean | eastern NPO | 180 | 255 | 25 | 40 | 953 |
| 4 | | western subtropical North Pacific Ocean | subtropical western NPO | 121 | 180 | 5 | 25 | 497 |
| 5 | | eastern subtropical North Pacific Ocean | subtropical eastern NPO | 180 | 283 | 5 | 25 | 671 |
| 6 | | western equatorial Pacific Ocean | equatorial western PO | 117 | 180 | -5 | 5 | 65 |
| 7 | | eastern equatorial Pacific Ocean | equatorial eastern PO | 180 | 285 | -5 | 5 | 156 |
| 8 | | western subtropical South Pacific Ocean | subtropical western SPO | 142 | 180 | -25 | -5 | 52 |
| 9 | | eastern subtropical South Pacific Ocean | subtropical eastern SPO | 180 | 290 | -25 | -5 | 220 |
| 10 | | western South Pacific Ocean | subtropical western SPO | 149 | 180 | -40 | -25 | 43 |
| 11 | | eastern South Pacific Ocean | subtropical eastern SPO | 180 | 290 | -40 | -25 | 109 |
| 12 | | eastern Southern Ocean | eastern SO | 180 | 291.93 | -60 | -40 | 25 |
| 13 | | Pacific sector of Antarctic | Pacific ANT | 147 | 291.93 | -90 | -60 | 3 |
| 14 | | Japan Sea | JAS | 127 | 142 | 33.4 | 52 | 2806 |
| 15 | | Arabian Sea | ARB | 32 | 119 | -10 | 30 | 60 |
| 16 | | Indian Ocean | IO | 20 | 129 | -35 | -10 | 72 |
| 17 | | Southern Ocean | SOJ | 20 | 147 | -60 | -31 | 34 |
| 18 | | Arctic Ocean | ARC | 0 | 360 | 70 | 90 | 651 |
| 19 | | Middle Southern Ocean | middle SO | 147 | 180 | -60 | -31 | 16 |
| 20 | | Barents Sea and Coast of Norway | BARE | 2 | 71 | 58 | 80 | 829 |
| 21 | | Baltic Sea | BALT | 9 | 30 | 53 | 66 | 6450 |
| 22 | | North Sea | NORS | 0 | 360 | 50 | 58 | 5452 |


Table 1. Detail of each box in the global ocean. (Continued)





| Box | subbox | Area | Code | Lon_w | Lon_e | Lat_s | Lat_n | Data No |
|---|---|---|---|---|---|---|---|---|
| 23 | 1 | Irish Sea | IRIS | 356 | 358 | 53 | 55 | 3730 |
| 23 | 2 | Irish Sea | IRIS | 353 | 356 | 53 | 56 | 2500 |
| 23 | 3 | Irish Sea | IRIS | 352 | 357 | 52 | 53 | 1452 |
| 23 | 3 | Irish Sea | IRIS | 350 | 358 | 51 | 52 | |
| 23 | 4 | Irish Sea | IRIS | 350 | 357 | 50 | 51 | 54 |
| 23 | 5 | Irish Sea | IRIS | 356 | 359 | 55 | 56 | 0 |
| 23 | 5 | Irish Sea | IRIS | 358 | 359 | 54 | 55 | |
| 24 | | English Channel | ENGC | 0 | 360 | 49 | 50.5 | 1575 |
| 25 | 1 | northern North Atlantic Ocean | NNA | 350 | 360 | 59 | 62 | 2268 |
| 25 | 1 | northern North Atlantic Ocean | NNA | 350 | 355 | 55 | 59 | |
| 25 | 1 | northern North Atlantic Ocean | NNA | 350 | 352 | 54 | 55 | |
| 25 | 1 | northern North Atlantic Ocean | NNA | 350 | 351 | 52 | 54 | |
| 25 | 1 | northern North Atlantic Ocean | NNA | 0 | 3 | 59 | 62 | |
| 25 | 2 | northern North Atlantic Ocean | NNA | 295 | 350 | 45 | 70 | 1322 |
| 25 | 2 | northern North Atlantic Ocean | NNA | 350 | 360 | 62 | 70 | |
| 25 | 2 | northern North Atlantic Ocean | NNA | 350 | 355 | 45 | 50 | |
| 25 | 2 | northern North Atlantic Ocean | NNA | 0 | 3 | 62 | 64 | |
| 25 | 2 | northern North Atlantic Ocean | NNA | 355 | 0 | 45 | 46 | |
| 25 | 2 | northern North Atlantic Ocean | NNA | 355 | 359 | 46 | 47 | |
| 25 | 2 | northern North Atlantic Ocean | NNA | 355 | 358 | 47 | 48 | |
| 25 | 2 | northern North Atlantic Ocean | NNA | 0 | 3 | 62 | 64 | |
| 25 | 2 | northern North Atlantic Ocean | NNA | 0 | 4 | 64 | 65 | |
| 25 | 2 | northern North Atlantic Ocean | NNA | 0 | 5 | 65 | 66 | |
| 25 | 2 | northern North Atlantic Ocean | NNA | 0 | 6 | 66 | 67 | |
| 25 | 2 | northern North Atlantic Ocean | NNA | 0 | 7 | 67 | 68 | |
| 25 | 2 | northern North Atlantic Ocean | NNA | 0 | 8 | 68 | 69 | |
| 25 | 2 | northern North Atlantic Ocean | NNA | 0 | 9 | 69 | 70 | |
| 26 | | Black Sea | BLAS | 27 | 42 | 41 | 48 | 88 |
| 27 | | Mediterranean Sea | MEDS | 0 | 360 | 30 | 46 | 202 |
| 28 | | North Atlantic Ocean | NAO | 262 | 360 | 15 | 45 | 136 |
| 29 | | Central Atlantic Ocean | CAO | 0 | 360 | -30 | 15 | 92 |
| 30 | | South Atlantic Ocean | SAO | 290 | 360 | -60 | -30 | 34 |
| 31 | | Sea of Okhotsk | SOO | 135 | 165 | 43 | 63 | 72 |
| 32 | | Eastern China Sea | ECS | 117 | 131 | 25 | 41 | 1189 |
| 33 | | South China Sea | SCS | 99 | 125 | -2 | 25 | 90 |
| 34 | | Berigng Sea | BERS | 162 | 203 | 52 | 66 | 71 |
| 35 | | Indonesian Archipelago | IND | 105 | 142 | -18 | 4 | 27 |
| 36 | | Atlantic sector of Antarctic | Atlantic ANT | 0 | 360 | -90 | -60 | 3 |
| 37 | | Indian sector of Antarctic | Indian ANT | 20 | 147 | -90 | -60 | 4 |



## 2.2 The 0.5-yr average values of $^{137}$Cs

### 2.2.1 The 0.5-yr average values of the surface $^{137}$Cs activity concentrations in each box

The 0.5-yr average values of the surface $^{137}$Cs activity concentrations in each box were calculated and shown in the dataset (Inomata and Aoyama, 2022c:http://dx.doi.org/10.34355/Ki-net.KANAZAWA-U.00151). These 0.5-yr average values of the surface $^{137}$Cs activity concentrations are useful to verify the general ocean circulation models (Tsumune et al., 2011; Tsubono et al., 2016) and assess the radiation doses delivered to humans through the ingestion of marine food (Aarklog et al., 1997; IAEA, 2005; UNSCARE, 2013). The 0.5-yr average values of the surface $^{137}$Cs concentrations in each box were produced by

the grid value producing command of block median programs (Wessel et al., 2013). The block median reads the arbitrary data (x, y, z) and calculates the median value in a grid defined in the setting range. We produced the dataset (box number, year, and $^{137}$Cs activity concentrations) for the open ocean (Pacific Ocean, Indian Ocean, Antarctic Ocean, and Atlantic Ocean) and the northern North Atlantic Ocean, Arctic Ocean, and its marginal seas by considering that the ocean currents and major source of $^{137}$Cs in the surface seawater had a 0.5-year time interval. These gridded data were recalculated to continuous curvature

splines with adjustable tension, and these values are regarded as the 0.5-yr average value. In the case of Box 2, significantly higher concentrations were observed only near the F1NPS (Box 2.1-2.6) because of the direct release of $^{137}$Cs. We used only the $^{137}$Cs activity data in Box 2.0 for the analysis of the 0.5-yr average values because the significantly higher values are localized and do not reflect concentrations throughout Box 2.

## 2.3 Estimates of apparent half residence times in global surface seawater

### 2.3.1 Definition of the period based on fallout characteristics and observed data

The $^{137}$Cs activity concentrations are mainly dominated by fallout into the surface seawater, radioactive decay with a 30.17-yr half-life, horizontal transport, and downwards transport below the mixed layer. As described above, the dominant source of $^{137}$Cs in seawater is considered to be large-scale nuclear weapons testing in the 1945 to the 1960s (UNSCARE, 2000; IAEA,

2005), directly discharged $^{137}$Cs in seawater from the Sellafield and La Hague nuclear reprocessing plants after 1970, deposition by the Chernobyl accident in 1986, and release by the F1NPS accident in 2011. Based on the long-term measurement of $^{90}$Sr deposition by the global monitoring network, the cumulative $^{90}$Sr deposition reached a maximum in the late 1960s to the early 1970s (UNSCARE, 2000). This suggests that the atmospheric deposition of $^{137}$Cs had a minor contribution to the surface seawater $^{137}$Cs activity concentrations after 1970 (UNSCARE, 2000; Hirose and Aoyama, 2003a; Aoyama et al., 2006).

Assuming that $^{137}$Cs in the seawater exponentially decreased by advection and diffusion with radioactive decay after 1970, the decrease in $^{137}$Cs activity concentrations in the surface seawater in each box was estimated using the corresponding fitting of the exponentially decreasing curves in each box. This decreasing rate of $^{137}$Cs activity concentrations is controlled by radionuclide decay as well as physical ocean circulation because the contribution from large-scale deposition by atmospheric nuclear tests was negligible after 1970 (UNSCARE, 2000). The regression line of the 0.5-yr average value of $^{137}$Cs for each



box was determined. However, the decreasing exponential trend was disturbed by several unexpected accidents, such as the Chernobyl accident in 1986 and the F1NPS accident in 2011, as well as direct discharge from nuclear reprocessing power plants. The Tap, therefore, was estimated for several periods, taking into account the source contribution as follows.

Tap1 is before 1970 (periods with nuclear weapon tests at a global scale), Tap2 is the period from 1970 to 1986 (until the Chernobyl accident), Tap3 is from 1990 to 2010 (after the Chernobyl accident), and Tap4 is after 2011 (F1NPS accident).

The Tap of $^{137}Cs$ for each period, from 1 to 4, in the surface seawater was calculated using the following equations:

$$^{137}Cs = {^{137}Cs_0}exp(-\lambda_{cs,\ apparent}) \tag{1}$$

$$\lambda_{Cs,apparent} = \lambda_{Cs,ocean} + \lambda_{Cs,decay} \tag{2}$$

$$Tap = 0.693/(\lambda_{Cs,apparent}) \tag{3}$$

$$Tpo = 0.693/(\lambda_{Cs,ocean}) \tag{4}$$

where $\lambda_{Cs,apparent}$, $\lambda_{Cs,ocean}$, and $\lambda_{Cs,decay}$ are the decay constants for apparent decay, physical oceanographic decay, and radioactive decay, respectively.

### 2.3.2. Case study in the Pacific Ocean and its marginal sea

The $^{137}Cs$ measurements in the surface seawater used in this study are discrete at spatial and temporal scales. To evaluate
the temporal variation in $^{137}Cs$ in the surface seawater in more detail, we focus on boxes that have much measured data, such as the western North Pacific Ocean and Japan Sea. In the Arctic Ocean (Box 18), European coastal sea (Boxes 20, 22, 23, and 24), northern North Atlantic Ocean (Box 25), and Central Atlantic Ocean (Box 28), the Tap cannot be approximated by an exponential function due to the direct discharge of $^{137}Cs$ from the two nuclear fuel reprocessing plants. We also did not estimate the Tap in the Black Sea (Box 26) and Mediterranean Sea (Box 27) because $^{137}Cs$ activity concentrations derived from global-
scale nuclear weapon tests were disturbed by $^{137}Cs$ deposition from the Chernobyl accident. Furthermore, the Tap in the North Pacific Ocean after 2011 was not estimated because of the $^{137}Cs$ released from the F1NPS.

### 2.4 F1NPS contribution to the 0.5-yr $^{137}Cs$ average values in the North Pacific Ocean

After several years following the F1NPS accident, $^{137}Cs$ activity concentrations in surface seawater gradually increased in
the Japan Sea, Eastern China Sea, subarctic North Pacific Ocean, and Bering Sea (Aoyama et al., 2016a, 2017; Inomata et al., 2018; Smith et al., 2017, Kumamoto et al., 2019). To estimate the contribution of the F1NPS accident, $^{137}Cs$ derived from the F1NPS accident ([F1NPS-$^{137}Cs$]) in 2011 was estimated using the following equation.

$$[F1NPS\text{-}^{137}Cs]each\ box = [0.5\text{-yr average } ^{137}Cs\ values]each\ box - [Global\text{-}^{137}Cs]each\ box \tag{5}$$

The "[Global-$^{137}Cs$]each box" in 2011 was estimated by extrapolating the exponential regression line from 1990 to 2010 under
the assumption that the apparent half residence time of $^{137}Cs$ activity concentrations were the same value.



## 2.5 Reconstruction with the 2-minute latitude/longitude $^{137}$Cs deposition amount in the global ocean

To estimate the $^{137}$Cs inventory in each box in the global ocean, we reconstructed the $^{137}$Cs global fallout distribution with a two-minute latitude/longitude grid based on the 10° latitude/longitude grid data of $^{137}$Cs deposition, which was constructed using $^{137}$Cs data measured in rainwater, seawater, and soil by Aoyama et al. (2006) and topography–bathymetry information based on the two-minute Gridded Global Relief Data (ETOPO2) (Earth Topography; NOAA National Geophysical Data Center, 2006), taking into account the ellipticity of the Earth (Oki et al., 1997; Suga et al., 2013).

## 2.6 Estimate of the $^{137}$Cs inventory in the surface mixed layer in the global ocean

The $^{137}$Cs in the surface seawater was transported horizontally along with the surface sea current and then transported below the surface mixed layer, after which it underwent radioactive decay (T$_{1/2}$= 30.17 yr). We estimated the $^{137}$Cs inventory in the surface mixed layer in each box and compared it with the $^{137}$Cs fallout amount until 1970 because a major input of $^{137}$Cs into surface seawater occurred in the early 1960s and reached a maximum in 1970 (IAEA, 2000; Aoyama et al., 2006). The $^{137}$Cs deposition in the surface seawater after 1970 was negligible compared to the activity concentrations in the surface seawater. Therefore, the behaviour of $^{137}$Cs in the surface seawater after 1970 is mainly controlled by radioactive decay and ocean physical transport processes. In this study, the $^{137}$Cs inventory was estimated under the assumption that $^{137}$Cs was mixed and homogeneously distributed within each box, with a 0.5-yr timescale in the surface mixed layer. However, the Irish Sea (Box 23) was subdivided into five regions (Boxes 23.1-23.5) because significantly higher values were observed around the directly discharged area from the Sellafield plant. However, there were no available data in Box 23.5. In the northern North Atlantic Ocean, higher $^{137}$Cs activity concentrations were observed in the region close to the Irish Sea or the North Sea. In this case, the northern North Atlantic Ocean was also divided into two boxes (Box 25.1 and Box 25.2).

The $^{137}$Cs surface mixed layer inventory (unit: PBq) was estimated using the following equation, assuming that $^{137}$Cs activity concentrations are almost constant in the surface mixed layer.

$$[^{137}\text{Cs inventory}]_{\text{box}} = \sum_{i=1}^{grid\ number}([\ ^{137}\text{Cs average value}]_{\text{box}}\times[\text{sea area}]_{\text{box}}\times[\text{averaged mixed layer depth}]_{\text{box}}) \quad (6)$$

The sea area in each box was calculated using the basin mask assigned to each two-minute latitude/longitude square, which was created based on the two-minute gridded global data (NOAA National Geophysical Data Center, 2006). The average mixed layer depth in each box was calculated using the two-degree latitude/longitude gridded "Mixed Layer Climatology" data constructed by the French Research Institute for the Exploitation of the Sea (IFREMER) (Montegut et al., 2004; Mignot et al., 2007). These data were regridded as the two-minute latitude/longitude data to set the same scale as the sea area dataset. In this dataset, the mixed layer depth was defined as the depth at which the surface temperature decreases by 0.2 °C and the density decreases by 0.03 kg m$^{-3}$. The mixed layer depths were estimated by 780000 profiles recorded in the World Ocean Database 09-National Oceanographic Data Center (NODC); Conductively, Temperature and Depth Profile (CTD) (1961–2008); World Ocean Circulation Experiment-3.0 Profiling Float Data (PFL) CTD (1990–2002) and ARGO PFL (1995–2008).



The mixed layer depth was the monthly time interval. The monthly mixed layer depth shows seasonal variation that is deeper in winter and shallower in summer. The maximum monthly mixed layer depth in each box was used to calculate the $^{137}$Cs

inventory in the mixed layer, taking into account that substances in the surface seawater are easily transported into the subsurface layer under the deeper mixed layer. The mixed layer depths used to estimate the inventory ranged from 33 to 182 m.

Furthermore, the $^{137}$Cs density in the surface seawater (unit: kBqm$^{-2}$) was also estimated by using the following equation.

$$[^{137}\text{Cs density in the surface seawater}]_{box} = [^{137}\text{Cs inventory}]_{box}/[\text{sea area}]_{box} \quad (7)$$

In this study, we used only data in Box 2.0 for the western North Pacific Ocean to estimate the representative value because significantly higher $^{137}$Cs values around the F1NPS causes the higher 0.5-yr average value in Box 2. However, in our previous study, we estimated the F1NPS-derived $^{137}$Cs behaviour in the subarctic North Pacific Ocean and western and eastern North Pacific Ocean by statistical optical interpretation analysis (Inomata et al., 2016). The results obtained by using optical interpretation analysis around the F1NPS were used in this study.


### 2.7. Mass balance; inflow and outflow of $^{137}$Cs from each box

In the marine environment, $^{137}$Cs activity concentrations after 1970 were dominantly controlled by radioactive decay and physical ocean processes, such as horizontal (outflow to the downstream box) and downwards transport below the surface mixed layer, except for the contribution from accidental release (the Chernobyl accident in 1986 and the Fukushima accident

in 2011) and direct discharge from nuclear reprocessing plants.

$$C_{i,box} = C_{0,box} - C_{0,box}\exp(-0.693/T_{1/2}\times\Delta t)-[\text{horizontal transport of } C_{0,box}]-$$

$$[\text{downwards transport of } C_{0,box} \text{ below the mixed layer}] \quad (8)$$

where $C_{0, box}$ is the 0.5-yr $^{137}$Cs average value in each box in the initial year and $C_{i, box}$ is the 0.5-yr $^{137}$Cs average value in each box after the $\Delta t$ year. In fact, distinguishing between horizontal and downwards-transported $^{137}$Cs amounts was very difficult

in this study. Therefore, the sum of the horizontally and downwardly transported $^{137}$Cs was estimated as the outflowed $^{137}$Cs in each box. In the northern North Atlantic Ocean, an extremely large inflow was estimated in 2005 due to the large values included in the dataset. These data were removed from the figures.

### 3 Results

Figure 2 shows the temporal variations in the 0.5-yr average value of $^{137}$Cs in each box. Table 2 lists the 0.5-yr average values of the surface $^{137}$Cs in the global ocean from 1960 to 2020 every 5 years. The 0.5-yr average values of $^{137}$Cs in each box are also listed in Inomata and Aoyama (2022c). The 0.5-yr average values of $^{137}$Cs until 2005 in each box in this study show no significant difference compared to those reported by Inomata et al. (2009), and the correlation coefficient is between 0.51 and 1. In this section, temporal variations in the 0.5-yr average $^{137}$Cs values in each box are described.


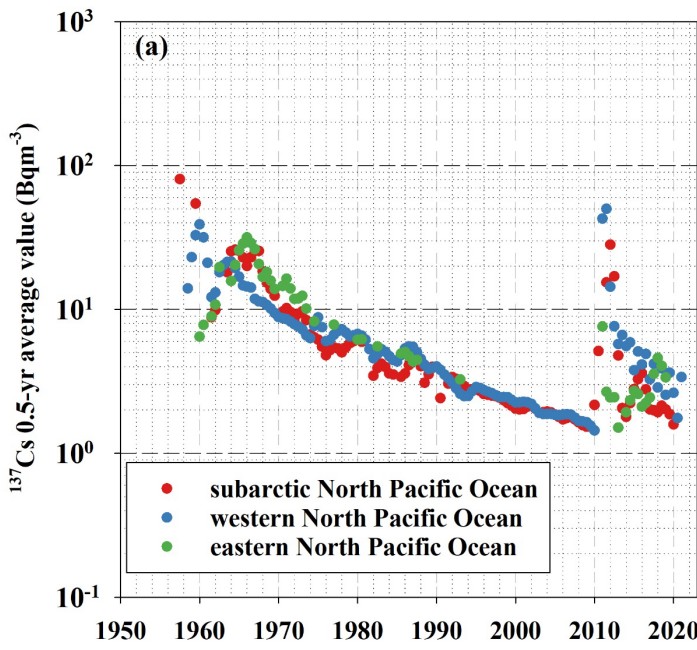

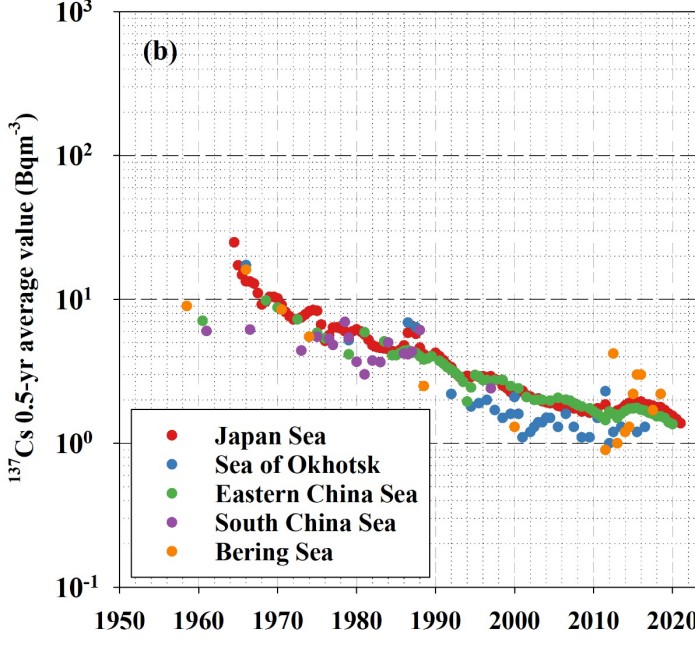



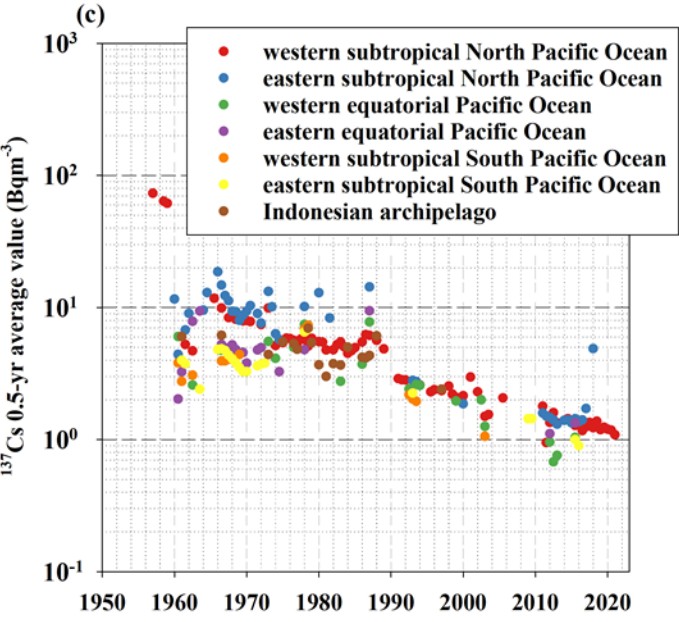


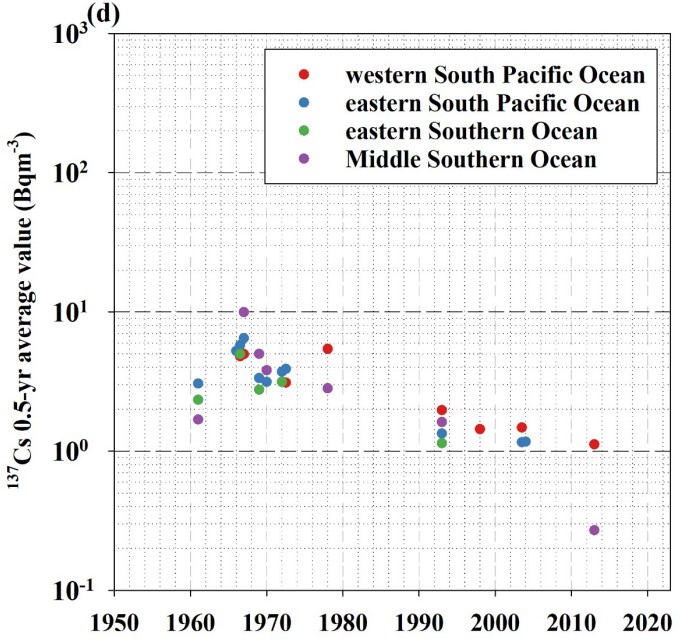





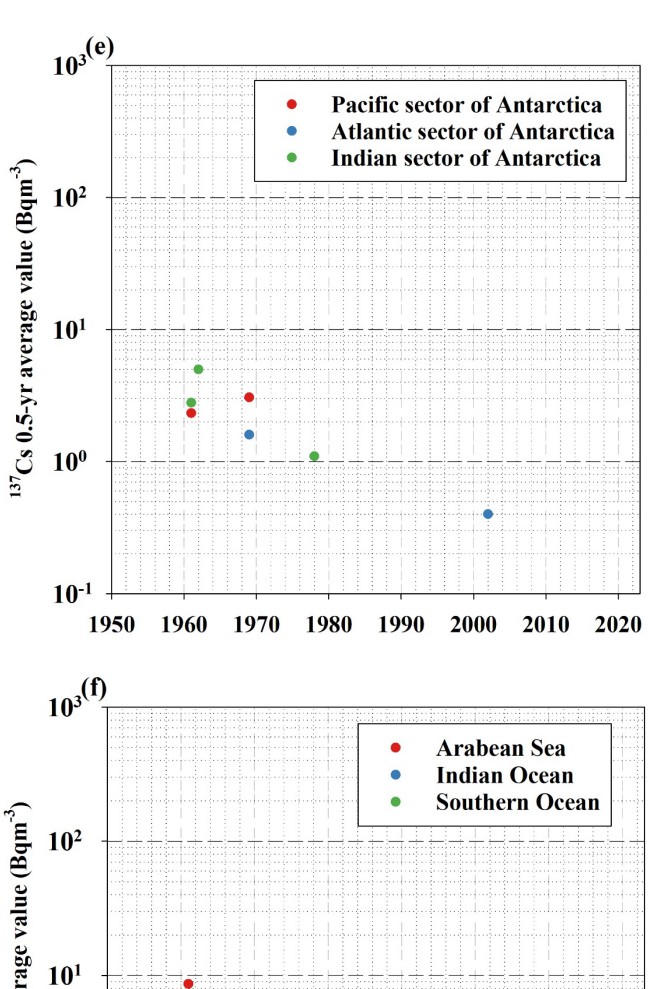




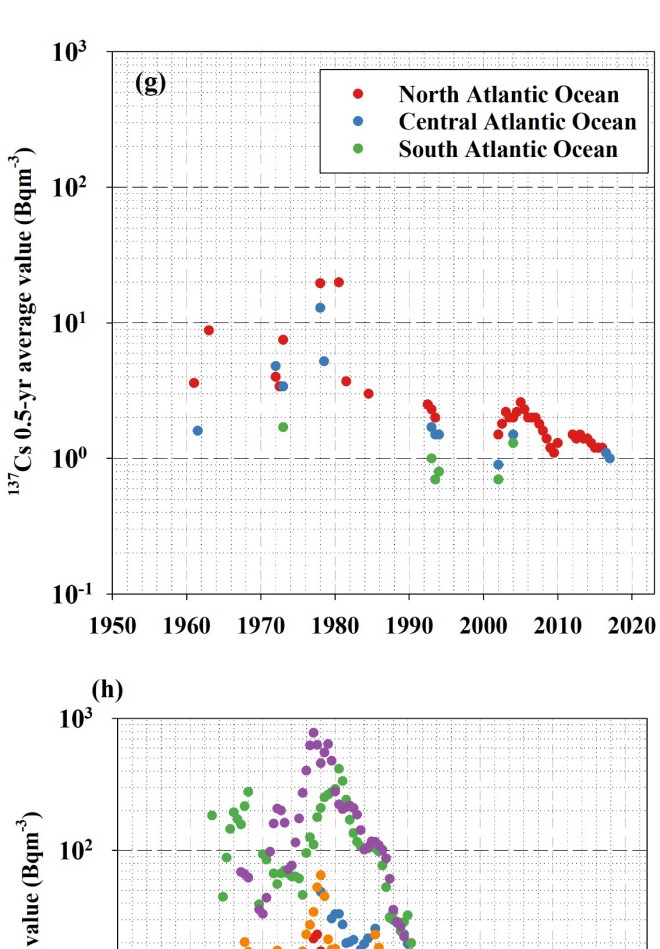

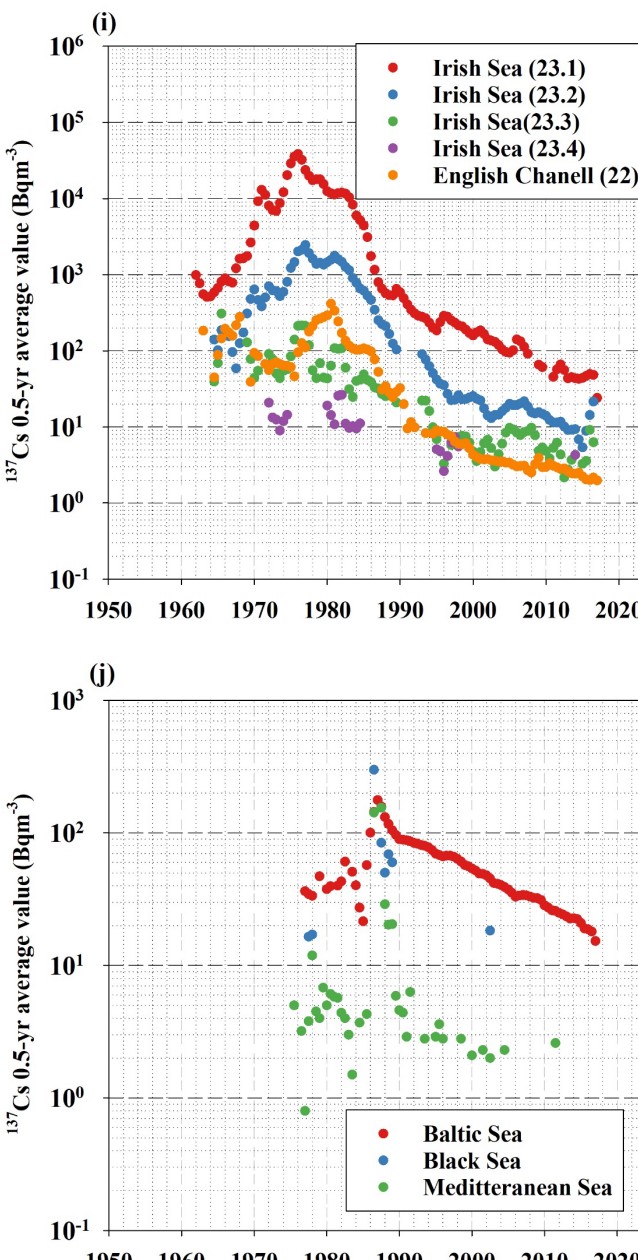


**Figure 2:** 0.5-yr average $^{137}$Cs values of each box for **(a)** Boxes 1–3 (subarctic North Pacific Ocean, western North Pacific Ocean, and eastern North Pacific Ocean), **(b)** Boxes 14 and 31–34 (Japan Sea, Sea of Okhotsk, Eastern China Sea, South China Sea, and Bering Sea), **(c)** Boxes 4-9 (western subtropical North Pacific Ocean, eastern subtropical North Pacific Ocean, western equatorial Pacific Ocean, eastern equatorial Pacific Ocean, eastern subtropical South Pacific Ocean, and western subtropical South Pacific Ocean), **(d)** Boxes 10–12 and 19 (western subtropical South Pacific




Ocean, eastern subtropical South Pacific Ocean, eastern Southern Ocean, and middle Southern Ocean), (e) Boxes 13, 36 and 17 (Pacific sector of the Antarctic Ocean, Atlantic sector of the Antarctic Ocean, and Indian sector of the Antarctic Ocean), (f) Boxes 15–17 (Arabian Sea, Indian Ocean, and Southern Ocean), (g) Boxes 28–30 (North Atlantic Ocean, Central Atlantic Ocean, and South Atlantic Ocean), (h) Boxes 18, 20, 22 and 25 (Arctic Ocean, Barents Sea and

coast of Norway, North Sea, and northern North Atlantic Ocean), (i) Boxes 23 and 24 (Irish Sea and English Channel), (j) Boxes 21, 26 and 27 (Baltic Sea, Black Sea and Mediterranean Sea).










Table 2. $^{137}$Cs deposition density, $^{137}$Cs inventory, and 0.5-yr $^{137}$Cs average value in each box.

| box | 0.5-yr $^{137}$Cs average value | | | | | | | | | | | |
| | 1960 | 1965 | 1970 | 1975 | 1980 | 1985 | 1990 | 1995 | 2000 | 2005 | 2010 | 2015 |
| | (Bq m$^{-3}$) | (Bq m$^{-3}$) | (Bq m$^{-3}$) | (Bq m$^{-3}$) | (Bq m$^{-3}$) | (Bq m$^{-3}$) | (Bq m$^{-3}$) | (Bq m$^{-3}$) | (Bq m$^{-3}$) | (Bq m$^{-3}$) | (Bq m$^{-3}$) | (Bq m$^{-3}$) |
| 1 subarctic North Pacific Ocean | 38 | 25 | 10 | 7.0 | 4.8 | 3.4 | 3.3 | 2.7 | 2.2 | 1.8 | 1.5 | 2.4 |
| 2 western North Pacific Ocean | 22 | 15 | 8.8 | 7.2 | 5.8 | 4.8 | 3.4 | 2.8 | 2.3 | 1.8 | 1.5 | 2.9 |
| 3 eastern North Pacific Ocean | 6.5 | 26 | 15 | 10 | 6.7 | 4.5 | 3.7 | 3.0 | 2.5 | 2.1 | 1.7 | 2.7 |
| 4 western subtropical North Pacific Ocean | - | - | 7.0 | 6.4 | 5.7 | 5.2 | 2.9 | 2.5 | 2.2 | 1.9 | 1.7 | 1.3 |
| 5 eastern subtropical North Pacific Ocean | 12 | - | 9.8 | 9.8 | 9.8 | 9.8 | 3.0 | 2.5 | 2.2 | 1.8 | 1.6 | 1.4 |
| 6 western equatorial Pacific Ocean | 4.6 | 4.3 | 5.4 | 5.4 | 5.4 | 5.4 | 3.0 | 2.4 | 1.9 | 1.5 | 1.2 | 0.9 |
| 7 eastern equatorial Pacific Ocean | 13 | 7.1 | 4.3 | 4.3 | 4.3 | 4.3 | 2.4 | 2.1 | 1.8 | 1.6 | 1.4 | 1.2 |
| 8 western subtropical South Pacific Ocean | 3.1 | 3.7 | 14 | 9.1 | 6.1 | 4.1 | 2.8 | 1.9 | 1.2 | 0.8 | 0.6 | 0.4 |
| 9 eastern subtropical South Pacific Ocean | 5.0 | 4.8 | 3.0 | 5.2 | 5.2 | 2.4 | 2.4 | 2.1 | 1.9 | 1.6 | 1.4 | 1.0 |
| 10 western South Pacific Ocean | - | 13 | 9.3 | 4.2 | 4.7 | 3.4 | 2.4 | 1.8 | 1.6 | 1.4 | 1.2 | 1.1 |
| 11 eastern South Pacific Ocean | 2.5 | 5.1 | 3.2 | 3.3 | 2.6 | 2.0 | 1.6 | 1.3 | 1.2 | 1.2 | 1.1 | 1.0 |
| 12 eastern Southern Ocean | 3.0 | 3.2 | 3.5 | 2.7 | 2.1 | 1.7 | 1.3 | 1.0 | 0.8 | 0.6 | 0.5 | 0.4 |
| 13 Pacific sector of Antarctic | 2.2 | 2.7 | 2.7 | 1.4 | 0.7 | 0.4 | 0.2 | 0.1 | 0.1 | 0.0 | 0.0 | 0.0 |
| 14 Japan Sea | 44 | 19 | 9.0 | 7.1 | 5.6 | 4.4 | 3.7 | 3.0 | 2.4 | 1.9 | 1.6 | 2.0 |
| 15 Arabian Sea | 8.6 | - | 4.1 | 4.0 | 3.8 | 3.5 | 2.9 | 2.4 | 1.9 | - | - | - |
| 16 Indian Ocean | 6.5 | 5.7 | 4.7 | 4.7 | 4.7 | 3.9 | 3.0 | 2.4 | 1.8 | 1.4 | 1.1 | 0.9 |
| 17 Southern Ocean | - | - | 3.5 | 3.5 | 3.5 | 2.6 | 2.1 | 1.6 | 1.3 | 1.0 | 0.8 | 0.6 |
| 18 Arctic Ocean | - | - | 1.2 | 6.1 | 11.3 | 8.2 | 6.7 | 3.2 | 1.4 | 3.6 | 1.7 | 1.5 |
| 19 Middle Southern Ocean | 0.3 | 7.2 | 4.4 | 3.3 | 2.4 | 1.8 | 1.4 | 1.0 | 0.8 | 0.6 | 0.4 | 0.3 |
| 20 Barents Sea and Coast of Norway | - | - | - | - | 33.1 | 25.6 | 19.7 | 4.4 | 4.9 | 3.7 | 2.8 | 2.1 |
| 21 Baltic Sea | - | - | - | 30 | 38 | 22 | 96 | 71 | 53 | 39 | 29 | 21 |
| 22 North Sea | - | 88 | 94 | 61 | 292 | 108 | 11 | 8.2 | 5.9 | 4.3 | 3.1 | 2.2 |
| 23.1 Irish Sea | 1138 | 670 | 4411 | 28847 | 12469 | 4425 | 587 | 185 | 159 | 94 | 55 | 44 |
| 23.2 Irish Sea | 822 | 102 | 639 | 1225 | 1457 | 616 | 96 | 42 | 25 | 20 | 14 | 5.4 |
| 23.3 Irish Sea | 498 | 69 | 44 | 84 | 43 | 49 | 19 | 6.9 | 4.8 | 9.7 | 4.8 | 3.3 |
| 23.4 Irish Sea | - | - | 15 | 13 | 10 | 8.0 | 2.9 | 2.8 | 2.7 | 2.5 | 2.4 | 2.2 |
| 23.5 Irish Sea | - | - | - | - | - | - | - | - | - | - | - | - |
| 24 English Channel | - | 14 | 59 | 24 | 6.1 | 12 | 6.7 | 5.7 | 39 | 1.7 | 1.8 | 1.5 |
| 25.1 northern North Atlantic Ocean | - | - | 41.3 | 165.0 | 278.3 | 108.1 | 6.8 | 5.1 | 3.9 | 2.9 | 2.2 | 1.7 |
| 25.2 northern North Atlantic Ocean | - | - | 8.4 | 11.9 | 18.0 | 9.4 | 5.3 | 3.3 | 6.1 | 1.9 | 1.6 | 1.3 |
| 26 Black Sea | - | - | - | - | - | - | 56.4 | 36.2 | 23.2 | 14.8 | 9.5 | 6.1 |
| 27 Mediterranean Sea | - | - | - | 5.0 | 5.0 | 4.3 | 4.3 | 3.5 | 2.9 | 2.6 | 1.3 | 1.2 |
| 28 North Atlantic Ocean | - | - | 8.7 | 8.7 | 8.7 | 3.1 | 2.5 | 2.0 | 1.7 | 1.3 | 1.1 | 0.9 |
| 29 Central Atlantic Ocean | - | - | 8.0 | 6.0 | 4.5 | 3.3 | 1.6 | 1.5 | 1.4 | 1.3 | 1.2 | 1.1 |
| 30 South Atlantic Ocean | - | - | 1.6 | 1.4 | 1.3 | 1.2 | 1.1 | 1.0 | 0.9 | 0.8 | 0.7 | 0.7 |
| 31 Sea of Okhotsk | - | 19.0 | 12 | 7.5 | 4.7 | 3.0 | 2.1 | 1.9 | 1.6 | 1.4 | 1.2 | 1.2 |
| 32 Eastern China Sea | 7.1 | - | 7.9 | 6.4 | 5.2 | 4.2 | 3.3 | 2.8 | 2.4 | 2.0 | 1.7 | 1.8 |
| 33 South China Sea | - | - | 5.6 | 7.8 | 10 | 3.9 | 2.5 | 1.6 | 1.0 | 0.6 | 0.4 | 0.3 |
| 34 Berigng Sea | - | - | 7.8 | 6.0 | 4.6 | 3.5 | 2.7 | 2.0 | 1.5 | 1.2 | 0.9 | 2.2 |
| 35 Indonesian Archipelago | 6.0 | 6.0 | 5.8 | 5.3 | 4.8 | 4.3 | 4.0 | 3.6 | 3.3 | 3.0 | 2.7 | 2.5 |
| 36 Atlantic sector of Antarctic | - | - | 1.5 | 1.2 | 1.0 | 0.8 | 0.7 | 0.5 | 0.4 | 0.4 | 0.3 | 0.2 |
| 37 Indian sector of Antarctic | 4.3 | 3.0 | 2.1 | 1.4 | 1.0 | 0.7 | 0.5 | 0.3 | 0.2 | 0.2 | 0.1 | 0.1 |




## 3.1 North Pacific Ocean

Figure 2a shows the temporal variations in the 0.5-yr average [137]Cs values in the subarctic North Pacific Ocean (Box 1), western North Pacific Ocean (Box 2), and eastern North Pacific Ocean (Box 3). The [137]Cs in this region largely originated from atmospheric deposition due to large-scale weapons tests (e.g., Aoyama et al., 2006; Inomata et al., 2012), the Chernobyl accident in 1986 (Miyao et al., 1989), and the F1NPS accident after 2011 (e.g., Aoyama et al., 2016a, b; Inomata et al., 2016). The temporal variations in the 0.5-yr average [137]Cs values were the highest in the middle and late 1950s. In the 1960s, the 0.5-yr average [137]Cs values increased gradually and reached a maximum in 1968. Then, the values decreased exponentially.

The 0.5-yr average value of [137]Cs after approximately 1970 became even smaller than those in the 1960s because the supply due to [137]Cs deposition was negligible. This implies that the variations in the [137]Cs activity concentrations after 1970 strongly depended on the physical processes in the ocean. Until the early 1960s, the 0.5-yr average values of [137]Cs in the subarctic North Pacific Ocean and western North Pacific Ocean were higher than those in the eastern North Pacific Ocean. However, the 0.5-yr average [137]Cs values in the eastern North Pacific Ocean increased in the 1960s and were slightly higher than those in the western North Pacific Ocean in the 1970s. A small peak that occurred in 1986 was caused by the deposition of [137]Cs from the Chernobyl accident. The supply of [137]Cs due to Chernobyl fallout in the western and eastern North Pacific Ocean was larger than that in the subarctic North Pacific Ocean. After 2011, the 0.5-yr average [137]Cs values increased in this region. However, the maximum values in the eastern North Pacific Ocean were several years later than those observed in the subarctic and western North Pacific Ocean due to the basin-scale transport of [137]Cs in the North Pacific Ocean.

## 3.2 Marginal seas of the western North Pacific Ocean

The temporal variations in the 0.5-yr average [137]Cs values in the marginal seas of the North Pacific Ocean (the Japan Sea (Box 14), Sea of Okhotsk (Box 31), Eastern China Sea (Box 32), South China Sea (Box 33), and Bering Sea (Box 34)) are displayed in Fig. 2b. The 0.5-yr average [137]Cs values in the marginal seas of the North Pacific Ocean also decreased exponentially. In 1986, a small peak due to [137]Cs fallout from the Chernobyl accident was observed in the Japan Sea and Sea of Okhotsk. In the 1980s, the 0.5-yr average value of [137]Cs was also high in the South China Sea. After the 1990s, the 0.5-yr average value of [137]Cs in the Sea of Okhotsk was smaller than those in the other boxes (the Japan Sea, Eastern China Sea, and South China Sea) in this region. An increase in the 0.5-yr average [137]Cs values in 2011 occurred in the Japan Sea, Sea of Okhotsk, and Bering Sea because of [137]Cs deposition originating from the F1NPS. In the Japan Sea and Eastern China Sea, the 0.5-yr average values of [137]Cs gradually increased after the F1NPS accident. In addition, an increase in the 0.5-yr average values of [137]Cs derived from the F1NPS occurred gradually in the Sea of Okhotsk and the Bering Sea.



### 3.3 Subtropical, equatorial Pacific Ocean and Indonesian Archipelago

The 0.5-yr average values of [137]Cs in the western subtropical North Pacific Ocean (Box 4) in the late 1950s were significantly high due to local fallout (Fig. 2c). In the 1970s-1980s, the 0.5-yr average values of [137]Cs in the eastern subtropical North Pacific

Ocean (Box 5), western equatorial Pacific Ocean (Box 6), eastern equatorial Pacific Ocean (Box 7), western subtropical South Pacific Ocean (Box 8), and eastern subtropical South Pacific Ocean (Box 9) were less than 20 Bq m$^{-3}$. In the 1970s and 1980s, the 0.5-yr average values of [137]Cs almost constantly varied in the eastern subtropical and western equatorial Pacific Ocean. In the eastern equatorial Pacific Ocean and western and eastern subtropical South Pacific Ocean, the values increased gradually. After the mid-1980s or the 1990s, the 0.5-yr average [137]Cs values showed an exponential decrease until 2011. The 0.5-yr

average values of [137]Cs in the Indonesian Archipelago showed almost the same range as those in the western equatorial Pacific Ocean.

### 3.4 South Pacific Ocean (Boxes 10-12 and 19)

In the South Pacific Ocean (the western South Pacific Ocean (Box 10), eastern South Pacific Ocean (Box 11), eastern Southern Ocean (Box 12), and middle Southern Ocean (Box 19)), as shown in Fig. 2d, the 0.5-yr average [137]Cs activity

concentrations in 1961 ranged from 1.4 to 2.5 Bq m$^{-3}$, whereas in 1967, the 0.5-yr average of [137]Cs increased to 4.5–9.9 Bq m$^{-3}$. Afterwards, the 0.5-yr average value of [137]Cs decreased exponentially, although the available data were limited. Moreover, substantially lower 0.5-yr average [137]Cs values of less than 1 Bq m$^{-3}$ were observed in the middle Southern Ocean in 2013.

### 3.5 Antarctic Ocean (Boxes 13, 36, and 37)

The 0.5-yr average [137]Cs activity concentrations in the Antarctic Ocean were the lowest in the global ocean, although the measurements were very limited (Fig. 2e). The 0.5-yr average [137]Cs activity concentrations from the 1960s to the 2010s decreased from 2.0 to 0.01 Bq m$^{-3}$ in the Pacific sector of the Antarctic Ocean (Box 13) and from 1.6 to 0.4 Bq m$^{-3}$ in the Atlantic sector of the Antarctic Ocean (Box 36). In the Indian sector of the Antarctic Ocean, the 0.5-yr average [137]Cs activity concentrations decreased from 5.5 to 1.1 Bq m$^{-3}$ during the period from 1961.5 to 1978 (Box 37). The decreasing rate in the

Antarctic region was larger than those in the other regions. These lowest 0.5-yr average [137]Cs values and larger decreasing rates were due to the long distance from the dominant [137]Cs fallout area. Furthermore, the upwelling of seawater from the deeper layers in the Antarctic Ocean may have caused dilution of the [137]Cs activity concentrations, resulting in the lowest [137]Cs values (Kumamoto et al., 2016).





### 3.6 Indian Ocean (Boxes 15, 16, and 17)

In the 1960s, the 0.5-yr average $^{137}$Cs activity concentration was higher in the Arabian Sea (Box 15, 8.9 Bq m$^{-3}$) than those in the Indian Ocean (Box 16, 6.7 Bq m$^{-3}$) and the Southern Ocean (Box 17, 4.0 Bq m$^{-3}$) (Fig. 2f). The average $^{137}$Cs activity concentrations in the 1960s showed a latitudinal gradient, with higher values in the northern areas and lower values in the southern areas. The 0.5-yr average $^{137}$Cs activity concentrations in the Arabian Ocean and Southern Ocean were almost constant in the 1970s, whereas those in the Indian Ocean increased slightly (3-6 Bq m$^{-3}$) in the 1970s. These values in the three boxes decreased to approximately 2 Bq m$^{-3}$ in the late 1990s and the early 2000s, although there were no available data in the 1980s. The 0.5-yr average $^{137}$Cs activity concentrations in the Indian Ocean were higher than those in the Arabian Ocean and the Southern Ocean. Note that the lowest values (0.15 Bq m$^{-3}$) were observed in 2012 in the Arabian Sea.

### 3.7 Atlantic Ocean (Boxes 28-30)

The Atlantic Ocean (the North Atlantic Ocean, Box 28; the Central Atlantic Ocean, Box 29; and the South Atlantic Ocean, Box 30) had a north–south gradient of 0.5-yr average $^{137}$Cs values, with higher values in the North Atlantic Ocean and lower values in the South Atlantic Ocean (Fig. 2g). In the North Atlantic Ocean, relatively high values were observed in the 1970s but then rapidly decreased after 1980. After 2000, an exponentially decreasing trend was not observed in the North Atlantic Ocean. The values slightly increased and reached the maximum value (2.5 Bq m$^{-3}$) in 2005, after which they gradually decreased. In the Central Atlantic Ocean, the temporal variations in the 0.5-yr average values of $^{137}$Cs exponentially decreased, although the data were very limited. In the South Atlantic Ocean, the 0.5-yr average values also decreased exponentially after 1970. Notably, the 0.5-yr average values in 2003 in the Central and South Atlantic Oceans slightly increased compared to those in 2002.

### 3.8 Arctic, northern North Atlantic Ocean and its marginal seas (Boxes 18, 20, 21, 22, and 25)

Fig. 2h shows the temporal variations in the 0.5-yr average values of $^{137}$Cs in the Arctic Ocean (Box 18), Barents Sea and coast of Norway (Box 20), North Sea (Box 22) and northern North Atlantic Ocean (Box 25.1, 25.2). The dominant sources of $^{137}$Cs in this area are the global-scale atmospheric deposition in the 1960s by large-scale nuclear weapons tests and two nuclear fuel reprocessing plants after the 1970s. In these regions, the 0.5-yr average values of $^{137}$Cs did not decrease exponentially due to the $^{137}$Cs discharged from the nuclear fuel reprocessing plants. After the mid-1970s, the 0.5-yr average $^{137}$Cs activity concentrations in the North Sea and northern North Atlantic Ocean increased rapidly. Thereafter, the 0.5-yr average $^{137}$Cs activity concentrations in these regions rapidly decreased, and the decreasing rate became small after 1990, which was associated with the reduced amount of released $^{137}$Cs.



In the Arctic Ocean, the 0.5-yr average $^{137}$Cs activity concentrations increased until the middle 1970s and then decreased. The overall 0.5-yr average $^{137}$Cs activity concentrations until the 1970s in the Arctic Ocean were lower than those in the surrounding oceans.

**3.9 Irish Sea and English Chanel (Boxes 23 and 24)**

The $^{137}$Cs activity concentrations in the Irish Sea (Box 23) and English Channel (Box 24) were primarily affected by the
discharge from the nuclear fuel reprocessing plants (Fig. 2i). Because the 0.5-yr average value of $^{137}$Cs is significantly higher than those in other regions, the scale of the y-axis changes from $10^{-1}$ to $10^{6}$ Bq m$^{-3}$. In the Irish Sea (Box 23.1), which is the discharge region, the 0.5-yr average $^{137}$Cs activity concentrations increased rapidly and reached 38532 Bq m$^{-3}$ in 1976 and then decreased rapidly. The 0.5-yr average $^{137}$Cs activity concentration decreased with increasing distance from the discharge region. In Box 23.2, the maximum value (2453 Bq m$^{-3}$) was observed in 1977. The $^{137}$Cs activity concentrations in the Celtic Sea (Box
23.4) were the lowest in this box. The decreasing gradient of the 0.5-yr average $^{137}$Cs values reflected the controlled discharge amount.

In the English Chanel, the 0.5-yr average $^{137}$Cs activity concentrations reached a maximum in 1981, and these also decreased over time. The 0.5-yr average $^{137}$Cs activity concentrations in 2017 in the English Chanel were 2.0 Bq m$^{-3}$.

**3.10 Baltic Sea (Box 21), Black Sea (Box 26) and Mediterranean Sea (Box 27)**

In the Baltic Sea, the 0.5-yr average values of $^{137}$Cs increased gradually in the 1970s due to the inflow of the $^{137}$Cs released from reprocessing plants (Fig. 2j). In 1986, the 0.5-yr average values of $^{137}$Cs increased rapidly (177 Bq m$^{-3}$) due to the deposition of $^{137}$Cs derived from the Chernobyl accident. The 0.5-yr $^{137}$Cs average values in the Baltic Sea decreased over time. The 0.5-yr average $^{137}$Cs activity concentration in the Baltic Sea in 2017 was estimated to be 15.3 Bq m$^{-3}$.
The 0.5-yr average value of $^{137}$Cs in the Black Sea in 1977 and 1978.5 was approximately 17 Bq m$^{-3}$, and in 1986, it increased to 299 Bq m$^{-3}$, which was at least 18 times higher than that before the Chernobyl accident (Fig. 2j). The 0.5-yr average value of $^{137}$Cs decreased rapidly to 60 Bq m$^{-3}$ in 1989. Although there were limited data, the 0.5-yr average $^{137}$Cs value in 2002 was almost equal (18.3 Bq m$^{-3}$) to that before the Chernobyl accident. The rapid decrease in surface $^{137}$Cs could be due to the strong intrusion of surface waters to the deep layers, $^{137}$Cs inflow into the Mediterranean Sea after passing through
the Bosporus Strait, and radioactive decay (Egorov, 1999; Delfanti et al., 2014).

In the Mediterranean Sea, the 0.5-yr average value of $^{137}$Cs varied from 0.8 to 12 Bq m$^{-3}$ before the Chernobyl accident (Fig. 2j). After the accident, it increased to 142 and 155 Bq m$^{-3}$ in 1986.5 and 1987.5, respectively. In the following years, the 0.5-yr average value of $^{137}$Cs decreased rapidly and became almost the same as that before the Chernobyl accident. This rapid decrease could have been due to the stronger intrusion of bottom water from the surface (Delfanti et al., 2000; Delfanti and
Papucchi, 2010).



### 3.11. Comparison with the 0.5-yr average $^{137}$Cs values in the Pacific Ocean, Indian Ocean, and Atlantic Ocean

Fig. 3 shows a comparison of the 0.5-yr average $^{137}$Cs values in the Pacific Ocean, Indian Ocean, and Atlantic Ocean. A significant feature is that the highest values were observed in the western North Pacific Ocean, and the values decreased

exponentially over time until 2011. In contrast, the values in the western equatorial Pacific Ocean, western subtropical South Pacific Ocean, and Indian Ocean increased gradually in the 1970s and 1980s, followed by a decrease after the 1990s. The difference in the 0.5-yr average $^{137}$Cs values in the Pacific Ocean and Indian Ocean became very small after 1980. Although the data are very limited, the 0.5-yr average $^{137}$Cs values in the Atlantic Ocean were lower than those in the Pacific Ocean and the Indian Ocean. A slight increase in the 0.5-yr average $^{137}$Cs values was detected in 2003 in the Atlantic Ocean.


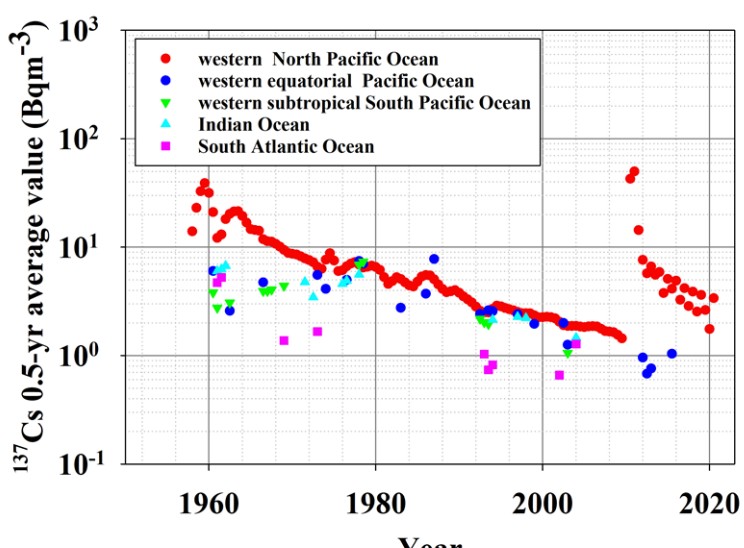

**Figure 3: Comparison with 0.5-yr average $^{137}$Cs values in the Pacific Ocean, Indian Ocean, and Atlantic Ocean.**



### 3.12. Changing Tap of the 0.5-yr average $^{137}$Cs values in the surface mixed layer

In the surface ocean, $^{137}$Cs activity concentrations are controlled by radioactive decay and horizontal transport associated with physical oceanographic processes, such as advection, diffusion, and downwards transport below the surface mixed layer. The Tap also varies in relation to these physical oceanographic processes. Considering that the half-life of $^{137}$Cs ($T_{1/2}$) is 30.17 years, the Tap should be shorter than the half-life if no source of $^{137}$Cs exists in the region of interest. A shorter Tap means that $^{137}$Cs is removed quickly in the area and/or the $^{137}$Cs inflow amount is small in the area compared with the

$^{137}$Cs outflow amount. In other words, a Tap shorter than the radioactive decay time indicates that the variations in the $^{137}$Cs activity concentrations are strongly controlled by physical ocean processes. In contrast, a longer Tap as well as a negative Tpo value means that $^{137}$Cs is preserved in the region for a longer time and/or there is an influx of water mass with higher $^{137}$Cs in the region compared to the $^{137}$Cs outflow from the region.

      Fig. 4 shows the temporal variation in the 0.5-yr average $^{137}$Cs values in the western North Pacific Ocean and the

Japan Sea as a typical case because the sequential time series varies due to the large amount of data. Compared with the western North Pacific Ocean (Fig. 4a), the Tap in the Japan Sea (Fig. 4b) is shorter in the three analysed periods (Tap1-Tap3).

      The Tap estimated in all boxes until 2010 are listed in Table 3. For Tap1, which is the longest, at 52.0 years, is estimated in the western equatorial Pacific Ocean, and Tpo in this box is negative. The shorter Tap1, which is approximately 4 years, is in the Japan Sea. In the case of Tap2, the longer Tap values are in the Indonesian Archipelago (36.7 years) and

South Atlantic Ocean (37 years). In these boxes, Tap3, which is estimated to be 36.7 years for the Indonesian Archipelago, 43.5 years for the Central Atlantic Ocean, and 37.0 years for the South Atlantic Ocean, are also longer than those in the other boxes. In the boxes with the longer Tap, Tpo is estimated to be negative.

(a) 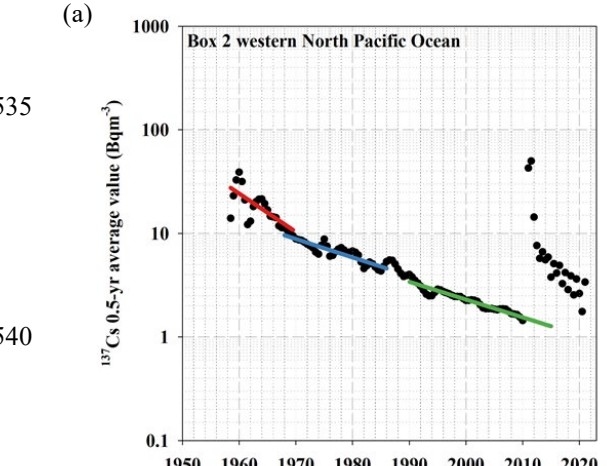 (b) 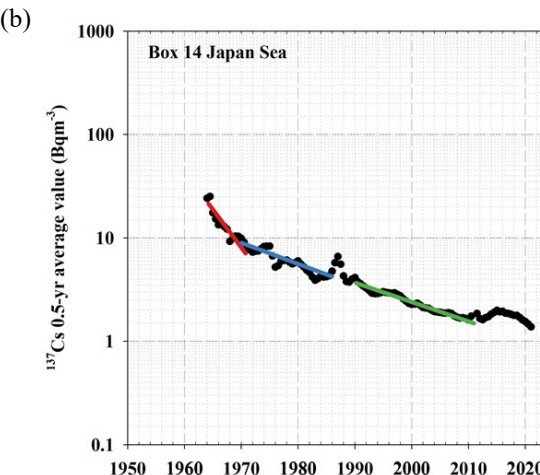

Figure 4: Temporal variation in the 0.5-yr average $^{137}$Cs values. (a) Western North Pacific Ocean, (b) Japan Sea. The lines
represent the exponential decay of the 0.5-yr $^{137}$Cs average value before 1970 (Tap1), 1970-1985 (Tap2), and 1990-2010 (Tap3).

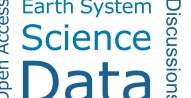

Table 3. Tap and Tpo in the global ocean.

|  | Box | Start | End | Tap | Tpo |
|---|---|---|---|---|---|
| Tap1 | subarctic North Pacific Ocean | 1950 | 1970 | 8.6 | 12.1 |
|  | western North Pacific Ocean | 1950 | 1970 | 8.1 | 11.1 |
|  | Japan Sea | 1950 | 1970 | 4.1 | 4.7 |
|  | western subtropical North Pacific Ocean | 1950 | 1970 | 4.3 | 5.0 |
|  | western equatorial Pacific Ocean | 1950 | 1970 | 52.0 | -71.8 |
|  | eastern equatorial Pacific Ocean | 1963 | 1970 | 5.8 | 7.2 |
|  | eastern subtropical South Pacific Ocean | 1963 | 1970 | 14.4 | 27.5 |
| Tap2 | subarctic North Pacific Ocean | 1970 | 1986 | 9.6 | 14.0 |
|  | western North Pacific Ocean | 1970 | 1986 | 16.9 | 38.6 |
|  | eastern North Pacific Ocean | 1970 | 1985 | 8.8 | 12.4 |
|  | Eastern China Sea | 1970 | 1985 | 16.8 | 38.0 |
|  | Japan Sea | 1970 | 1986 | 14.7 | 28.8 |
|  | Indonesian Archipelago | 1973 | 1997 | 36.7 | 36.7 |
|  | Arabian Sea | 1971 | 1978 | 25.6 | 167.6 |
|  | Central Atlantic Ocean | 1972 | 1993 | 11.9 | 19.5 |
|  | South Atlantic Ocean | 1973 | 2004 | 37.0 | -163.3 |
| Tap3 | subarctic North Pacific Ocean | 1990 | 2010 | 18.2 | 45.8 |
|  | western North Pacific Ocean | 1990 | 2010 | 17.5 | 41.9 |
|  | Eastern China Sea | 1990 | 2010 | 20.7 | 65.8 |
|  | Japan Sea | 1990 | 2010 | 16.1 | 34.4 |
|  | western equatorial Pacific Ocean | 1990 | 2010 | 15.6 | 32.5 |
|  | Indonesian Archipelago | 1973 | 1997 | 36.7 | 36.7 |
|  | Arabian Sea | 1978 | 1998 | 17.6 | 42.2 |
|  | Southern Ocean | 1972 | 1998 | 18.5 | 47.9 |
|  | Indian Ocean | 1994 | 2004 | 16.0 | 34.2 |
|  | Central Atlantic Ocean | 1993 | 2016 | 43.5 | -98.6 |
|  | South Atlantic Ocean | 1973 | 2004 | 37.0 | -163.3 |






### 3.13 Horizontal distribution of $^{137}$Cs in the surface mixing layer in the global ocean

**3.13.1 Horizontal distribution of $^{137}$Cs deposition as of the 1$^{st}$ of January 1970**

The atmospheric deposition of $^{137}$Cs due to the nuclear weapons tests in the global surface ocean as of the 1$^{st}$ of January 1970 is estimated to be $874 \pm 90$ PBq, with a two-minute latitude/longitude grid resolution. The global fallout of $^{137}$Cs in the Northern Hemisphere is $773 \pm 80$ PBq. At this time, the deposition in the global ocean is estimated to be $577 \pm 60$ PBq, which is an initial value. These results demonstrate good agreement with the estimation of the 10° latitude/longitude grid by

Aoyama et al. (2006), in which the atmospheric deposition of $^{137}$Cs derived from nuclear weapons tests in the Northern Hemisphere was $765 \pm 79$ PBq on the 1$^{st}$ of January 1970. However, these estimations are almost 1.4 times larger than those in the estimation by using a model simulation (UNSCARE, 1993), with an estimated value of 545 PBq.

Fig. 5 shows the horizontal distributions of $^{137}$Cs deposition density in each box in the global ocean. These values are also listed in Table 4. The $^{137}$Cs deposition density is high in the midlatitude region in the North Pacific Ocean (the Japan Sea,

subarctic North Pacific Ocean, Okhotsk Sea, western North Pacific Ocean, and eastern North Pacific Ocean) and the northern North Atlantic Ocean. In the North Pacific Ocean, these regions correspond to the area in which the Kuroshio Current and Kuroshio Extension are transported. In the northern North Atlantic Ocean, the higher $^{137}$Cs deposition area influences the Gulf Stream flow. The dominant features of these regions in the North Pacific Ocean and northern North Atlantic Ocean have received larger precipitation amounts and the occurrence of stratosphere–troposphere air mass exchange (Aoyama et al., 2006).

The larger air mass exchange between the stratosphere and troposphere means that the $^{137}$Cs injected into the stratosphere by the large-scale weapons tests is transported into the troposphere and deposited on the surface by precipitation. South of 5°N, the $^{137}$Cs deposition density is lower than that in the northern region and there is no significant difference between the open oceans (Pacific, Atlantic, and Indian Oceans).






(a)

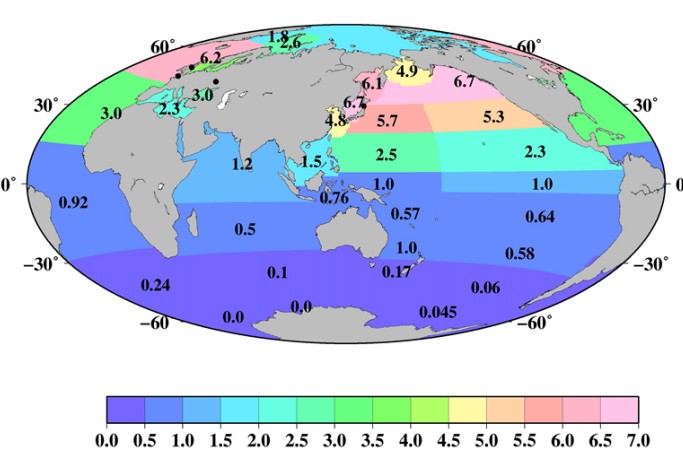

(b)

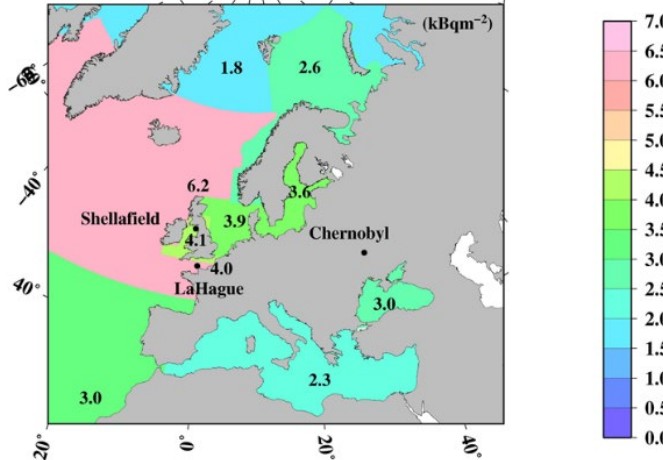

Figure 5: Horizontal distributions of $^{137}$Cs deposition density (KBq m$^{-2}$) as of the 1$^{st}$ of January 1970. (a) Global Ocean, (b) Northern North Pacific Ocean and its marginal seas. Black circles are locations of the F1NPS, Sellafield, La Hague, and Chernobyl power plants.





Fig. 6 shows the horizontal distribution of the $^{137}$Cs deposition amount as of the 1$^{st}$ of January 1970 in the surface mixed layer in the global ocean. These data are also listed in Table 4. In the Pacific Ocean, a higher $^{137}$Cs deposition amount

occurs in the subarctic North Pacific Ocean (71.6 PBq), western North Pacific Ocean (40.8 PBq), eastern North Pacific Ocean (52.4 PBq), and subtropical eastern North Pacific Ocean (47.9 PBq). In the Atlantic Ocean, a higher $^{137}$Cs deposition amount is found in the northern North Atlantic Ocean (sum of Boxes 25.1 and 25.2; 56.1 PBq) and North Atlantic Ocean (69.8 PBq). The $^{137}$Cs deposition amount in the Atlantic Ocean shows a significant latitudinal gradient, which is 27.3 PBq for the Central Atlantic Ocean and 5.1 PBq for the South Atlantic Ocean. In the Indian Ocean, the $^{137}$Cs deposition amount also has a north–

south gradient. The $^{137}$Cs deposition amount is the lowest in the Pacific sector (0.05 PBq), Atlantic sector (0 PBq), and Indian sector (0 PBq) of the Antarctic Ocean.

(a)                                                                                    (b)

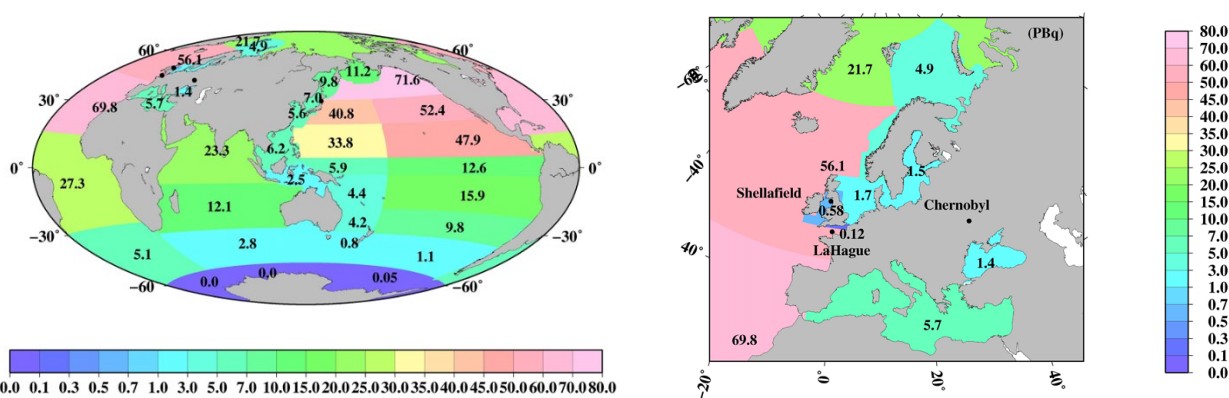

Figure 6: Horizontal distributions of the $^{137}$Cs deposition amount (PBq) in each box as of the 1$^{st}$ of January 1970. (a) Global Ocean, (b) Northern North Pacific Ocean and its marginal seas. Black circles are locations of the F1NPS, Sellafield, La Hague, and Chernobyl power plants.





Table 4. $^{137}$Cs deposition density and inventory as 1 January, 1970.

| box | $^{137}$Cs deposition density, 1970 | $^{137}$Cs deposition inventory, 1970 |
|---|---|---|
| | (kBq m$^{-2}$) | (PBq) |
| 1 subarctic North Pacific Ocean | 6.7 | 71.6 |
| 2 western North Pacific Ocean | 5.7 | 40.8 |
| 3 eastern North Pacific Ocean | 5.3 | 52.4 |
| 4 western subtropical North Pacific Ocean | 2.5 | 33.8 |
| 5 eastern subtropical North Pacific Ocean | 2.3 | 47.9 |
| 6 western equatorial Pacific Ocean | 1.0 | 5.9 |
| 7 eastern equatorial Pacific Ocean | 1.0 | 12.6 |
| 8 western subtropical South Pacific Ocean | 0.6 | 4.4 |
| 9 eastern subtropical South Pacific Ocean | 0.6 | 15.9 |
| 10 western South Pacific Ocean | 1.0 | 4.2 |
| 11 eastern South Pacific Ocean | 0.6 | 9.8 |
| 12 eastern Southern Ocean | 0.1 | 1.1 |
| 13 Pacific sector of Antarctic | 0.005 | 0.05 |
| 14 Japan Sea | 6.7 | 7.0 |
| 15 Arabian Sea | 1.2 | 23.3 |
| 16 Indian Ocean | 0.5 | 12.1 |
| 17 Southern Ocean | 0.1 | 2.8 |
| 18 Arctic Ocean | 1.8 | 21.7 |
| 19 Middle Southern Ocean | 0.2 | 0.8 |
| 20 Barents Sea and Coast of Norway | 2.6 | 4.9 |
| 21 Baltic Sea | 3.6 | 1.5 |
| 22 North Sea | 3.9 | 1.7 |
| 23.1 Irish Sea | 4.1 | 0.03 |
| 23.2 Irish Sea | 4.1 | 0.1 |
| 23.3 Irish Sea | 4.1 | 0.2 |
| 23.4 Irish Sea | 4.1 | 0.2 |
| 23.5 Irish Sea | 4.1 | 0.02 |
| 24 English Channel | 4.0 | 0.3 |
| 25.1 northern North Atlantic Ocean | 5.5 | 2.0 |
| 25.2 northern North Atlantic Ocean | 6.2 | 53.6 |
| 26 Black Sea | 3.0 | 1.4 |
| 27 Mediterranean Sea | 2.3 | 5.7 |
| 28 North Atlantic Ocean | 3.0 | 69.8 |
| 29 Central Atlantic Ocean | 0.9 | 27.3 |
| 30 South Atlantic Ocean | 0.2 | 5.1 |
| 31 Sea of Okhotsk | 6.1 | 9.8 |
| 32 Eastern China Sea | 4.8 | 5.6 |
| 33 South China Sea | 1.5 | 6.2 |
| 34 Berigng Sea | 4.9 | 11.2 |
| 35 Indonesian Archipelago | 0.8 | 2.5 |
| 36 Atlantic sector of Antarctic | 0.0 | 0.0 |
| 37 Indian sector of Antarctic | 0.0 | 0.0 |





### 3.13.2. Horizontal distribution of 0.5-yr average $^{137}$Cs values in the global ocean

Fig. 7 shows the spatial variations in the 0.5-yr average $^{137}$Cs values in the global ocean every 5 years. In 1970, the 0.5-year average $^{137}$Cs values were higher in the North Pacific Ocean and lower in the South Pacific Ocean. In particular, the highest value (14.8 Bq m$^{-3}$) is observed in the eastern North Pacific Ocean. In the South Pacific Ocean, relatively high values occur in the western and western subtropical South Pacific Ocean (9.3 and 13.6 Bq m$^{-3}$) compared to those in the eastern region (3-4.3 Bq m$^{-3}$). The 0.5-yr average $^{137}$Cs values in the Indonesian Archipelago (5.8 Bq m$^{-3}$) and the South China Sea (5.6 Bq

m$^{-3}$) are almost the same or slightly higher than those in the western equatorial Pacific Ocean (5.4 Bq m$^{-3}$). In 1975, the 0.5-yr average $^{137}$Cs values in the South China Sea were higher than those in the Indonesian Archipelago (Fig. 7b). In 1980 and 1985 (Fig. 7c, 7d), in the Pacific Ocean, higher values were found in the eastern subtropical Pacific Ocean (9.8 Bq m$^{-3}$). In the North Pacific Ocean, the 0.5-yr average $^{137}$Cs values are higher in the eastern region, whereas higher values in the South Pacific Ocean are found in the western region. In 1990, the 0.5-yr average $^{137}$Cs values decreased, although the concentration

distribution was similar to that of the 1980s (Fig. 7e). In particular, after 1990, the Indonesian Archipelago became the hot spot region, with relatively high 0.5-yr average $^{137}$Cs activity values in the Pacific Ocean and Indian Ocean. Higher values in this region were still observed in 2015 (Fig. 7j). The latitudinal gradient, which was higher in the North Pacific Ocean and lower in the South Pacific Ocean, became small, which lasted until 2010. In 2015, an increase in the 0.5-year average $^{137}$Cs values was found in the western and subarctic North Pacific Ocean due to the release of $^{137}$Cs from the F1NPS accident (Fig.

7j). The lowest value occurs in the Antarctic Ocean in the global ocean after 1970 (Fig. 7).

In the Atlantic Ocean, a latitudinal gradient that is higher in the northern North Atlantic Ocean and North Atlantic Ocean (north of 30°N) than in the Central and South Atlantic Ocean occurs. In 2015, the values in the South Atlantic Ocean (0.7 Bq m$^{-3}$) were almost equal to those in the Southern Ocean (0.6 Bq m$^{-3}$) (Fig. 7j). In the Atlantic Ocean, because of the discharged $^{137}$Cs in the surface seawater from the nuclear reprocessing plants, i.e., the Sellafield plant, significantly higher 0.5-

yr $^{137}$Cs average values occur in the Irish Sea, particularly at $^{137}$Cs discharge points (Box 23.1), as shown in Figs. 8 and 9. The discharged $^{137}$Cs is transported to the northern North Pacific Ocean (Box 25.1) and North Sea from the Irish Sea (Box 23.2). The 0.5-yr average $^{137}$Cs values after 1985 (Fig. 8d) decreased gradually in accordance with the discharged amount of $^{137}$Cs (Fig. 9c). The increase in the 0.5-yr average $^{137}$Cs values in the Baltic Sea (96 Bq m$^{-3}$) and Black Sea (56 Bq m$^{-3}$) in 1990 was caused by the deposition of $^{137}$Cs from the Chernobyl accident (Figs. 8e, f). Contamination due to the Chernobyl accident

continued in the Baltic Sea and Black Sea until 2015 (Fig. 8j).




**(a)**

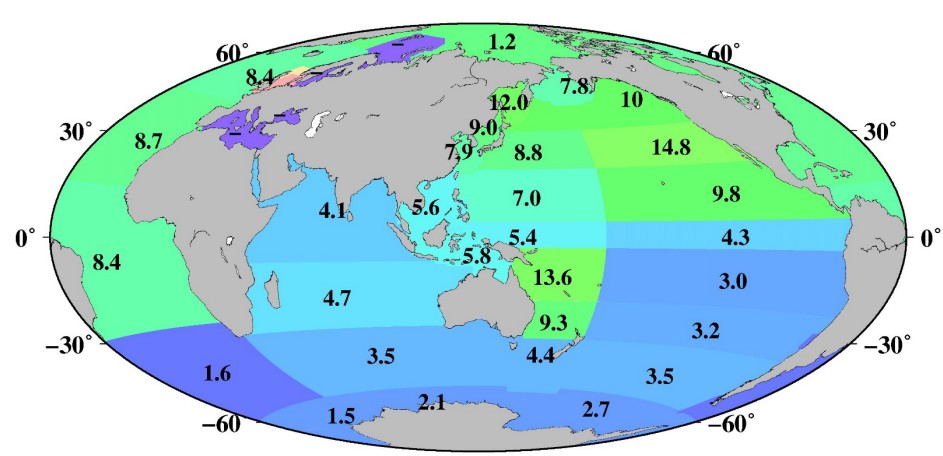

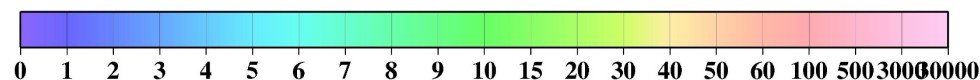

**(b)**


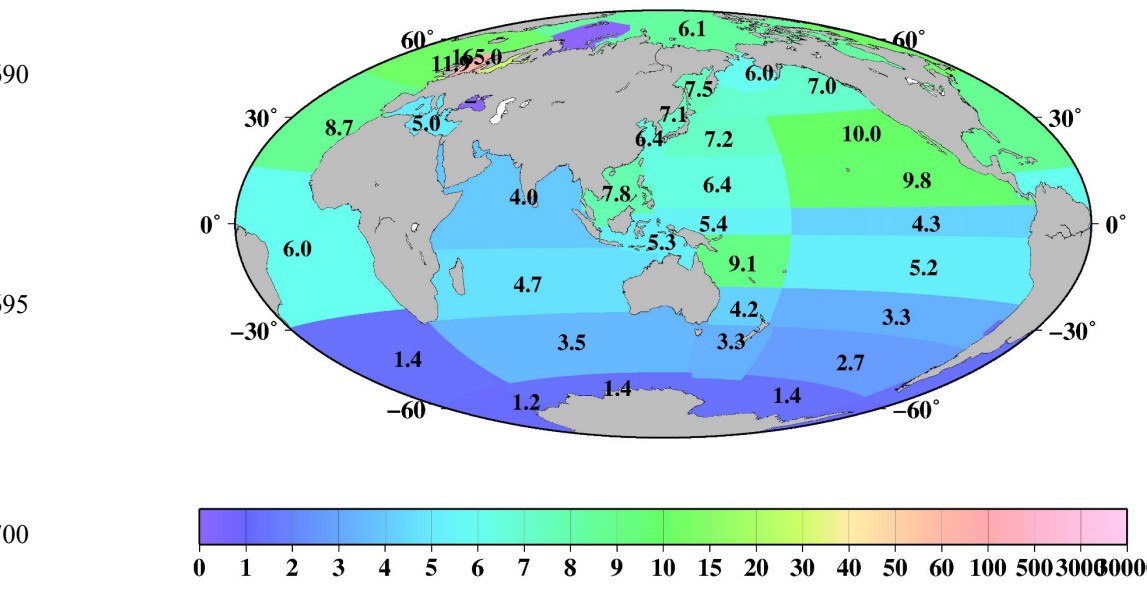





**(c)**

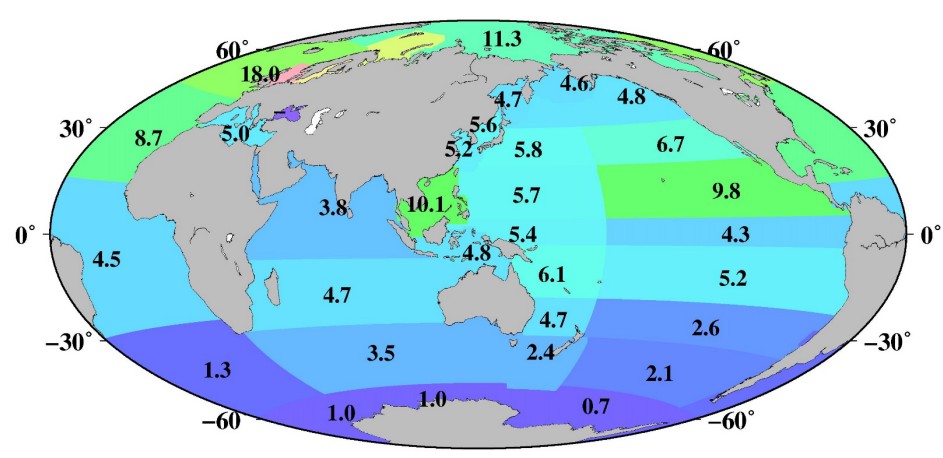

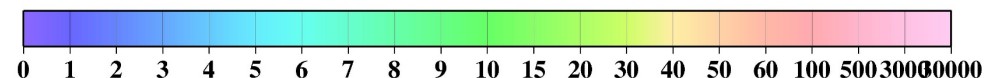


**(d)**

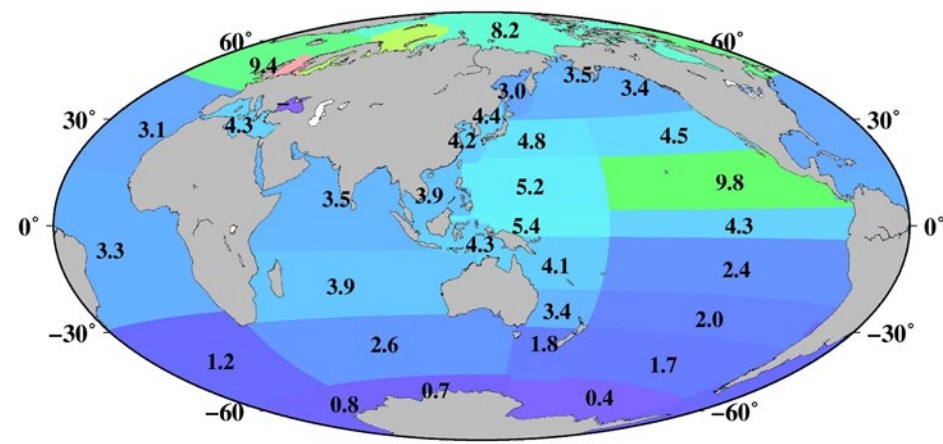


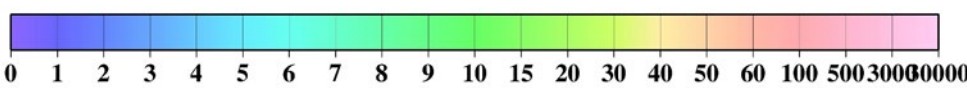






**(e)**

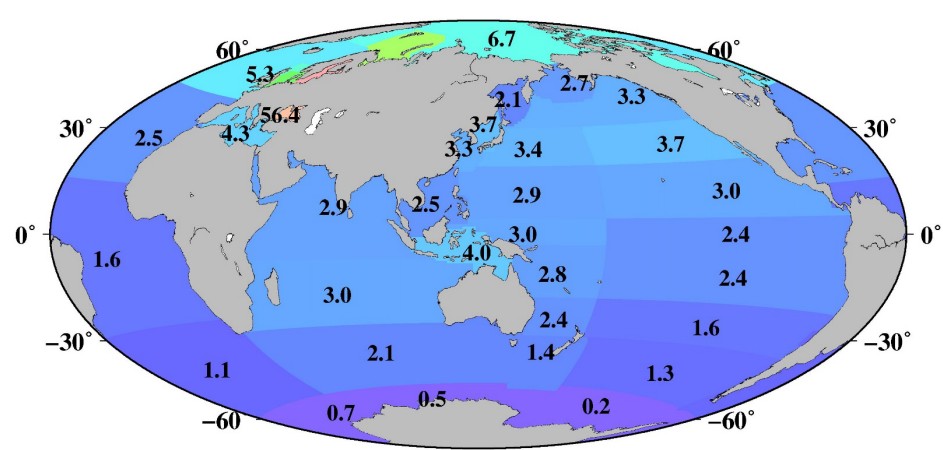

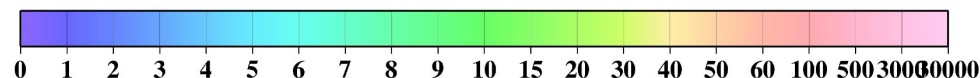


**(f)**


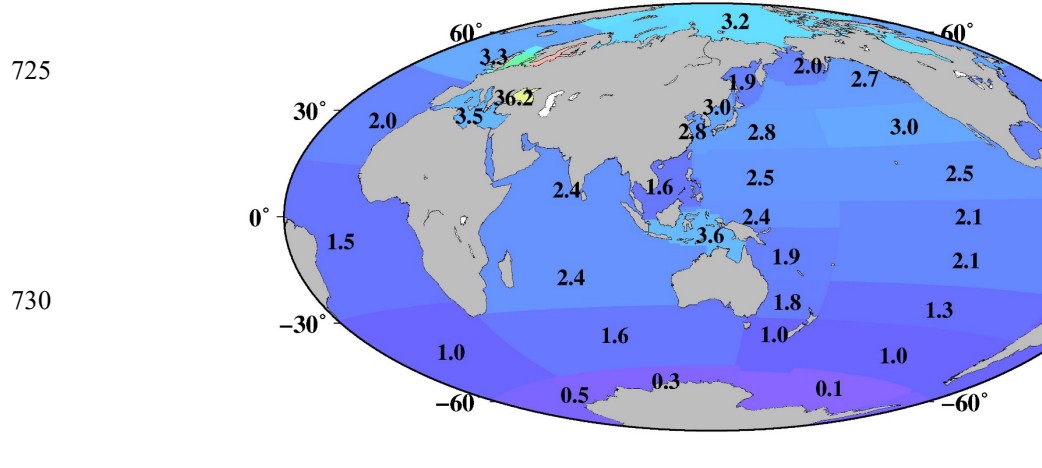





**(g)**

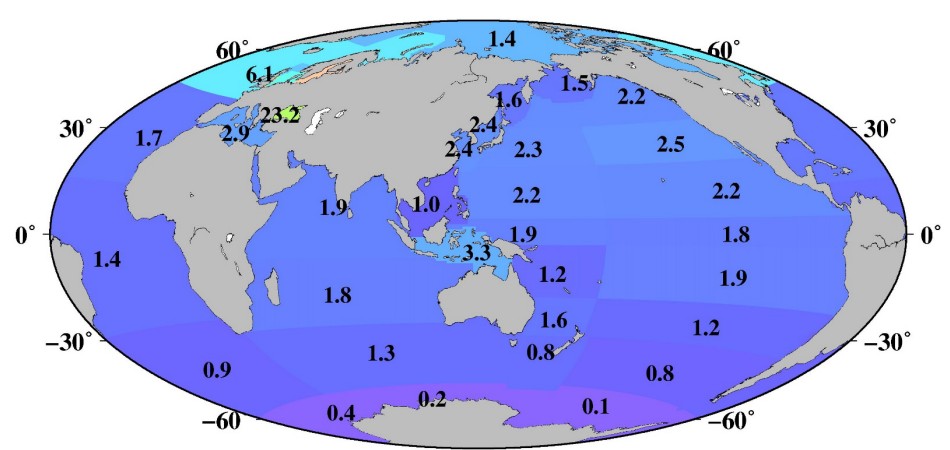

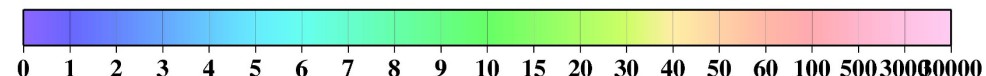

**(h)**


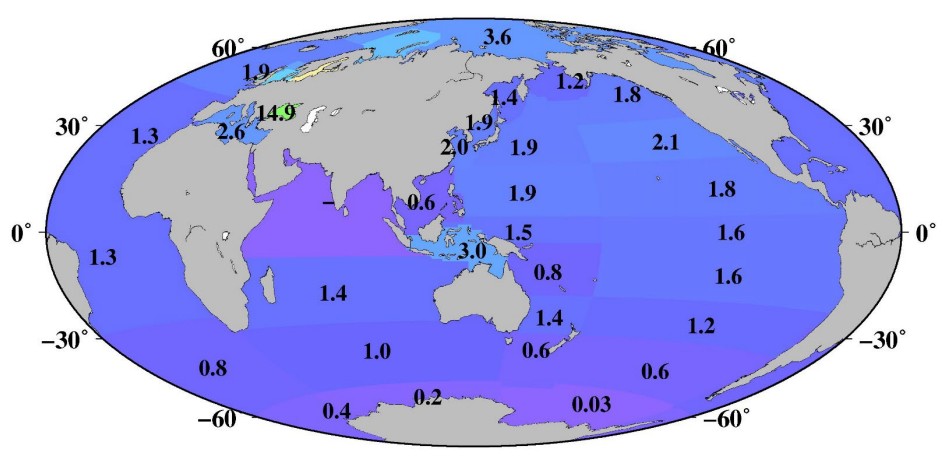


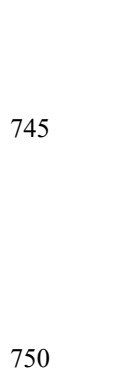


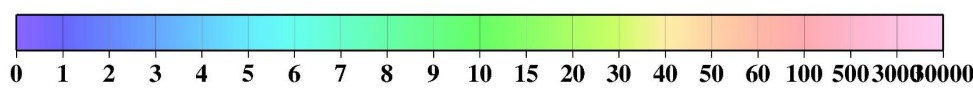

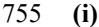



**(i)**



**(j)**

**Figure 7: Horizontal distributions of the 0.5-yr average $^{137}$Cs value in the surface mixed layer in the global ocean. The unit is Bqm$^{-3}$. (a) 1970, (b) 1975, (c) 1980, (d) 1985, (e) 1990, (f) 1995, (g) 2000, (h) 2005, (i) 2010, and (j) 2015.**



**(a)**

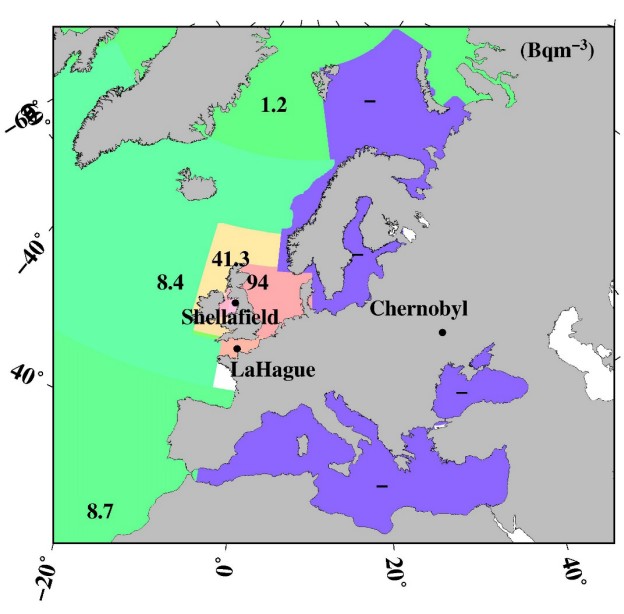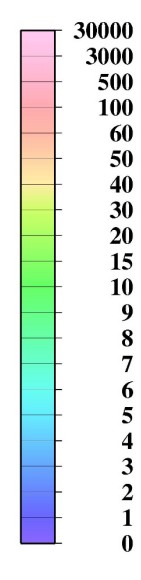

**(b)**

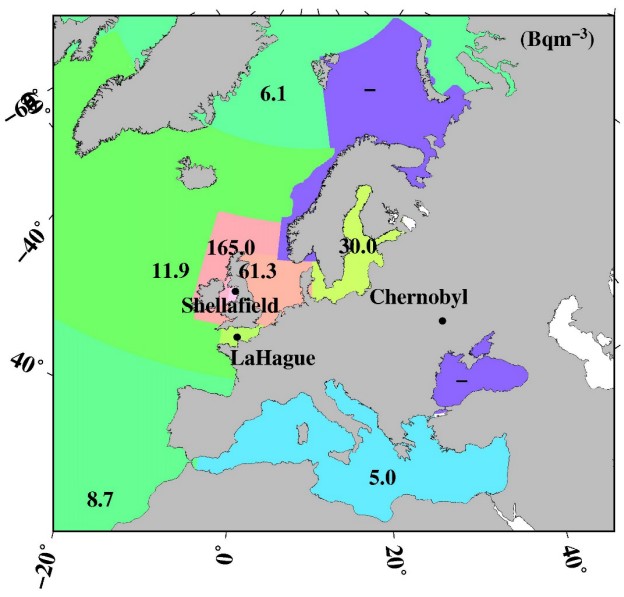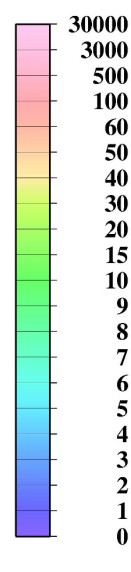



**(c)**

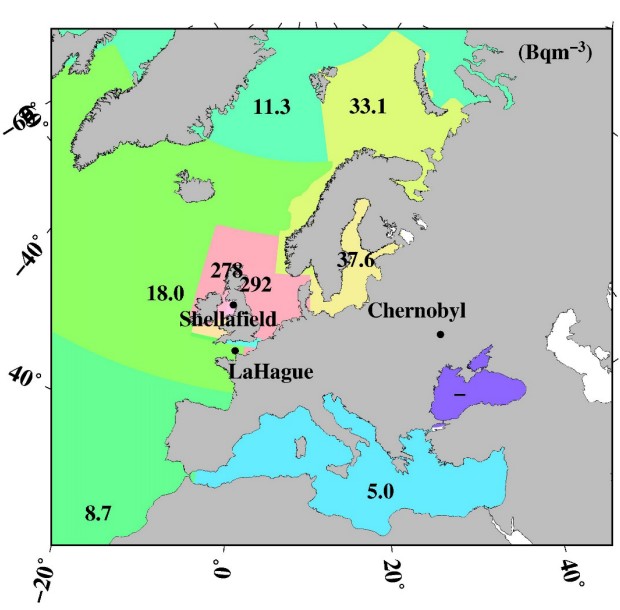
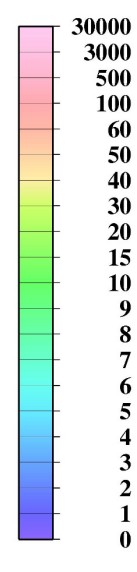

**795** **(d)**

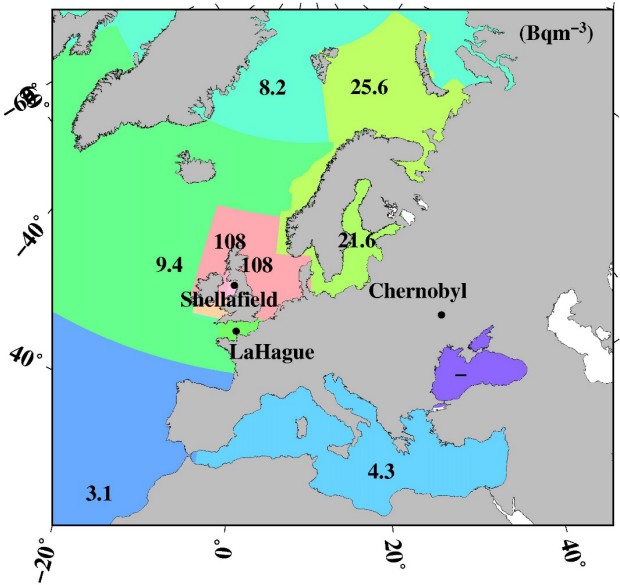
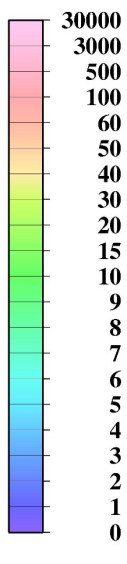





**(e)**

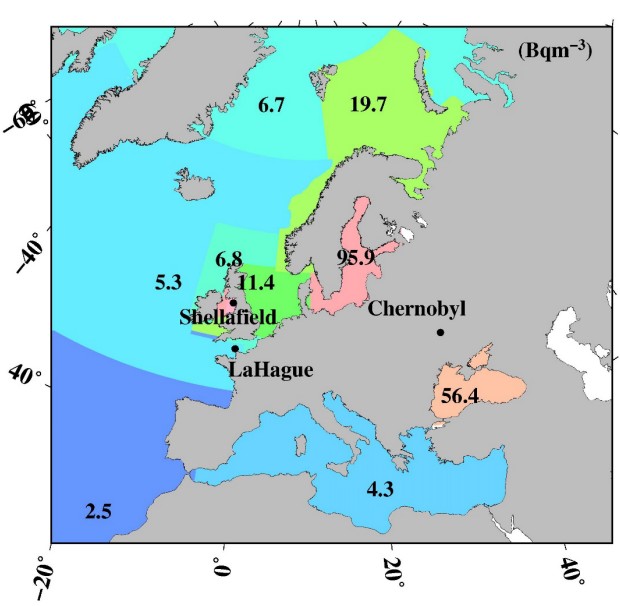
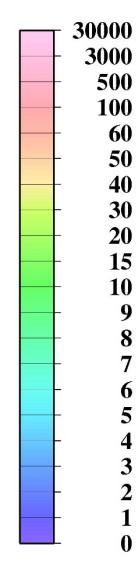

**(f)**

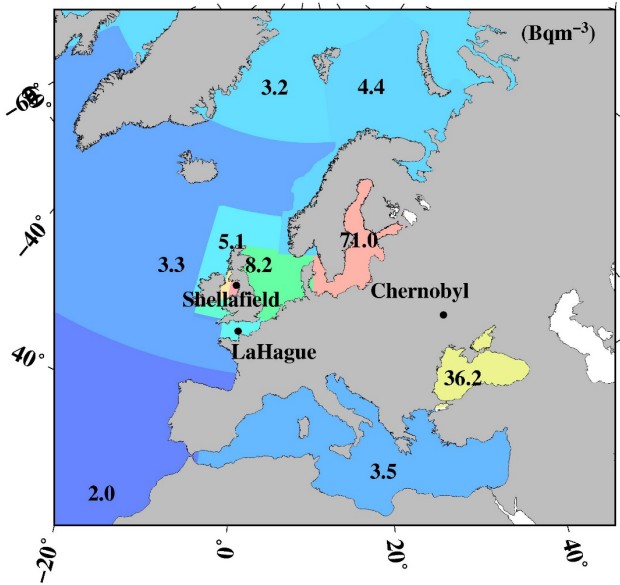
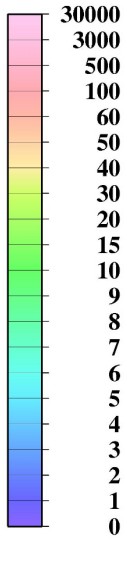



**(g)**

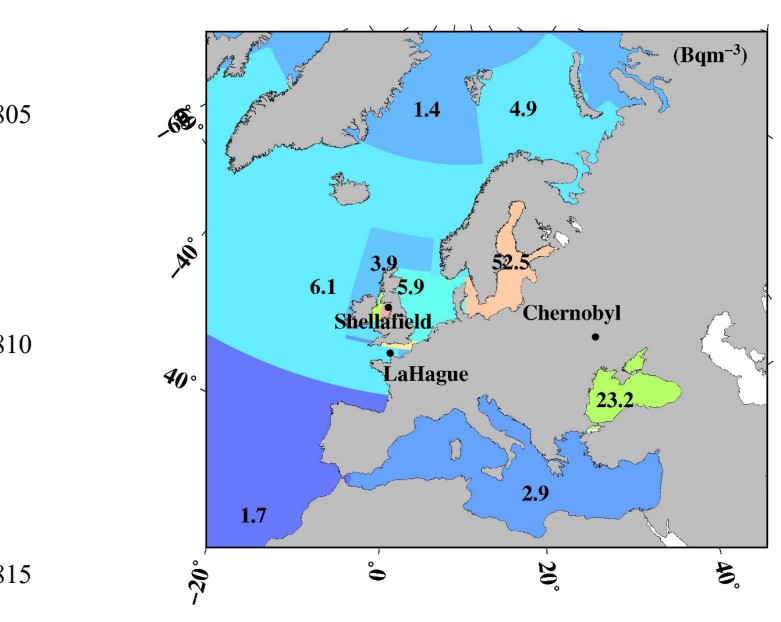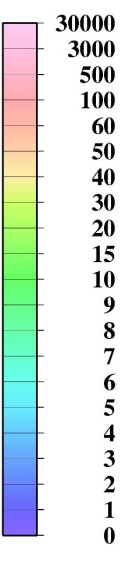

**(h)**

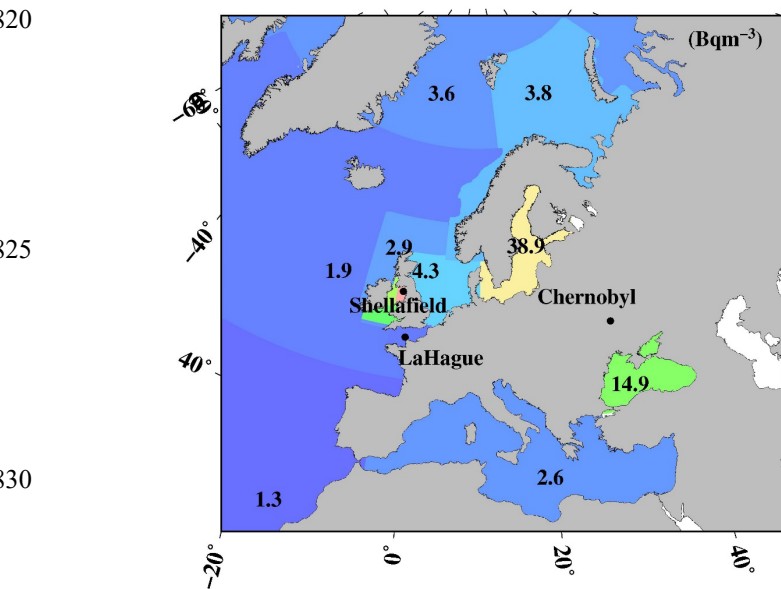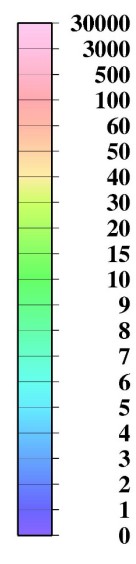



**(i)**

**137Cs.S.conc2010.EURO**



**(j)**

**137Cs.S.conc2015.EURO**




**Figure 8: Horizontal distributions of the 0.5-yr average $^{137}$Cs value in the surface mixed layer in the northern North Atlantic Ocean and its marginal seas. The unit is Bqm$^{-3}$. (a) 1970, (b) 1975, (c) 1980, (d) 1985, (e) 1990, (f) 1995, (g) 2000, (h) 2005, (i) 2010, and (j) 2015.**



**(a)**

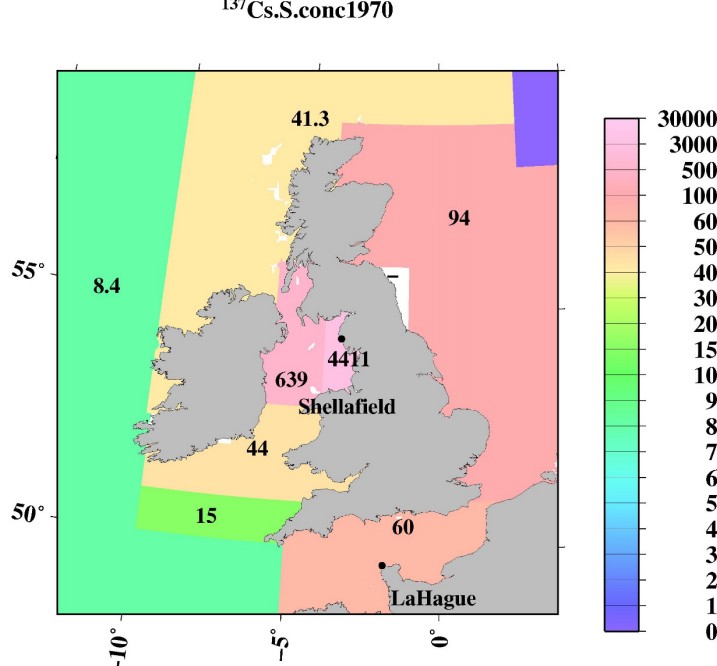


**(b)**

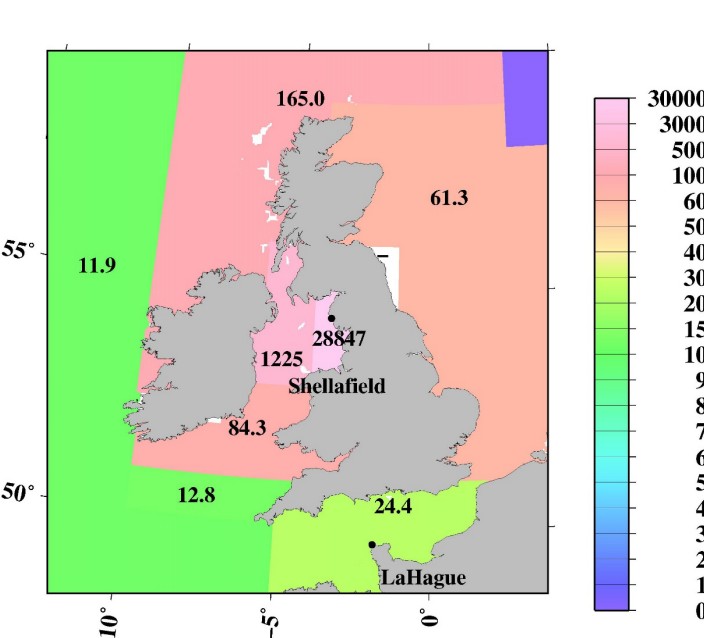

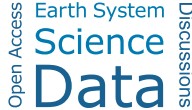

**(c)**

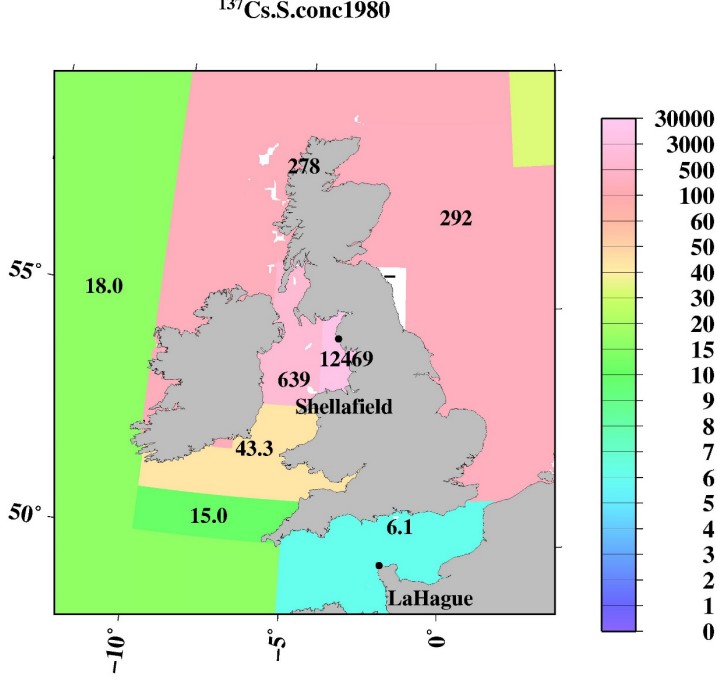

**(d)**

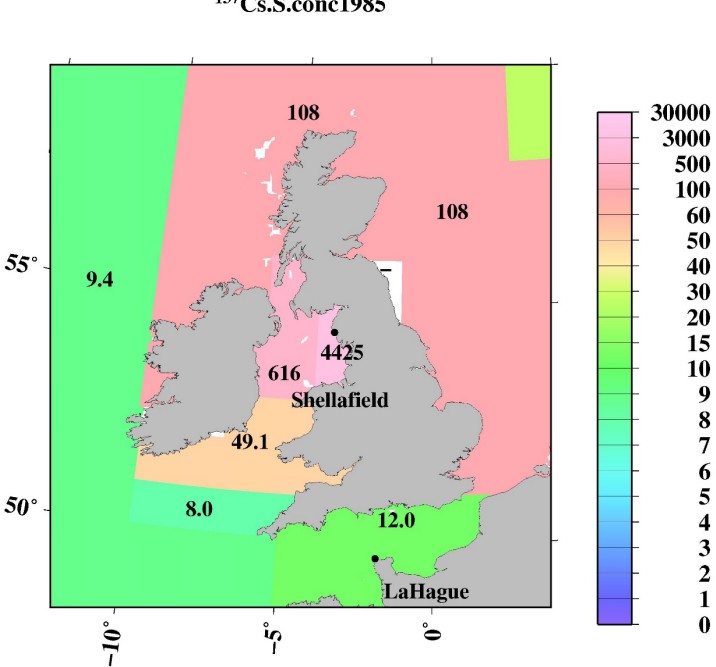





**(e)**

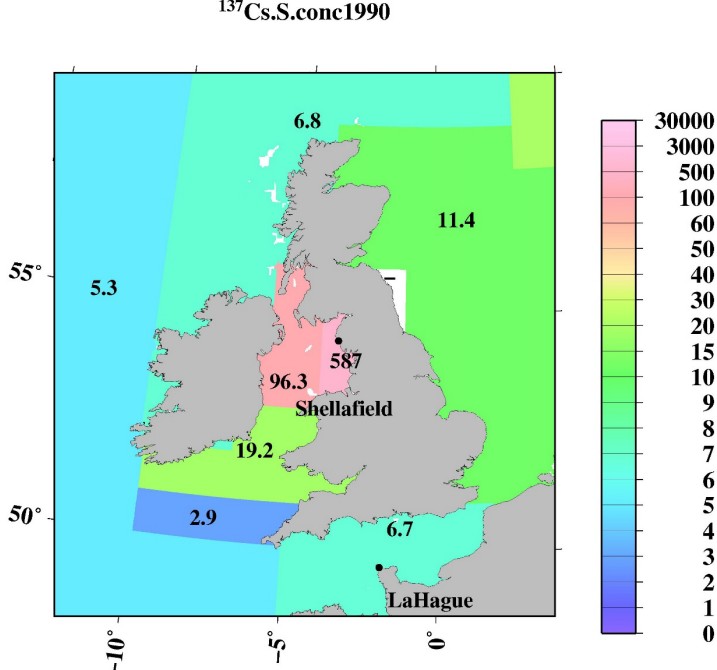

**(f)**

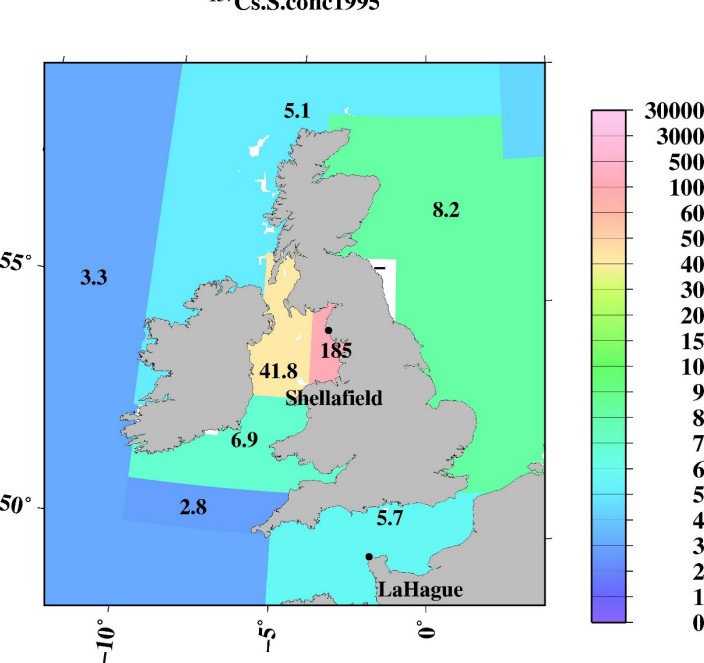



**(g)**

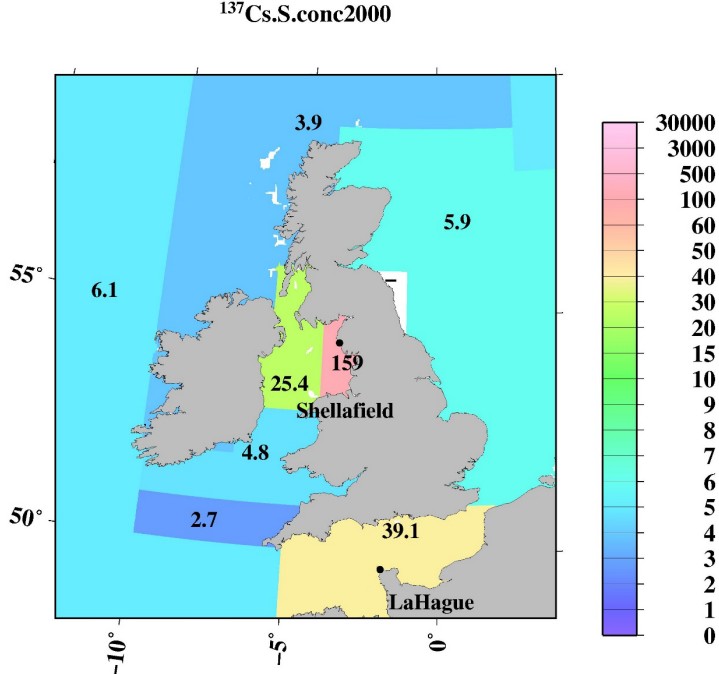

**(h)**

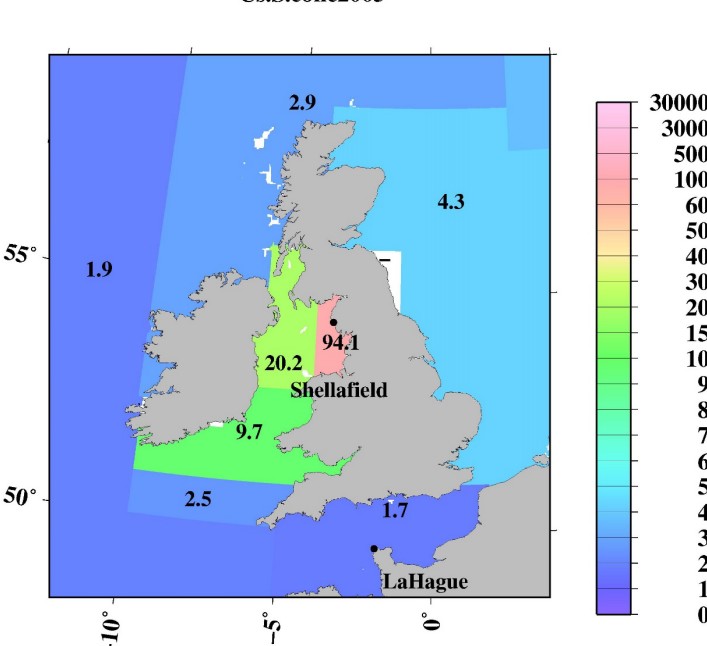





**(i)**

**(j)**

**Figure 9:** Horizontal distributions of the 0.5-yr average $^{137}$Cs value in the surface mixed layer in the Irish Sea. The unit is Bqm$^{-3}$. (a) 1970, (b) 1975, (c) 1980, (d) 1985, (e) 1990, (f) 1995, (g) 2000, (h) 2005, (i) 2010, and (j) 2015. The "-" mean that there is no available data.



### 3.14 $^{137}$Cs inventory in the surface mixing layer in the global ocean after 1970

The horizontal distribution of the surface mixed layer depth in the global ocean are shown in Fig. 10. The mixed layer depth in the open ocean shows a clear latitudinal distribution of deeper (~182 m) in the higher latitudes and shallower in the lower latitudes, particularly in the equatorial Pacific Ocean (48 m for the eastern equatorial Pacific Ocean and 58 m for the western equatorial Pacific Ocean). In the coastal sea, the mixed layer depths are shallower (33-76 m) than those in the open ocean. The mixed layer depth in each box is also listed in Table 5.

(a)

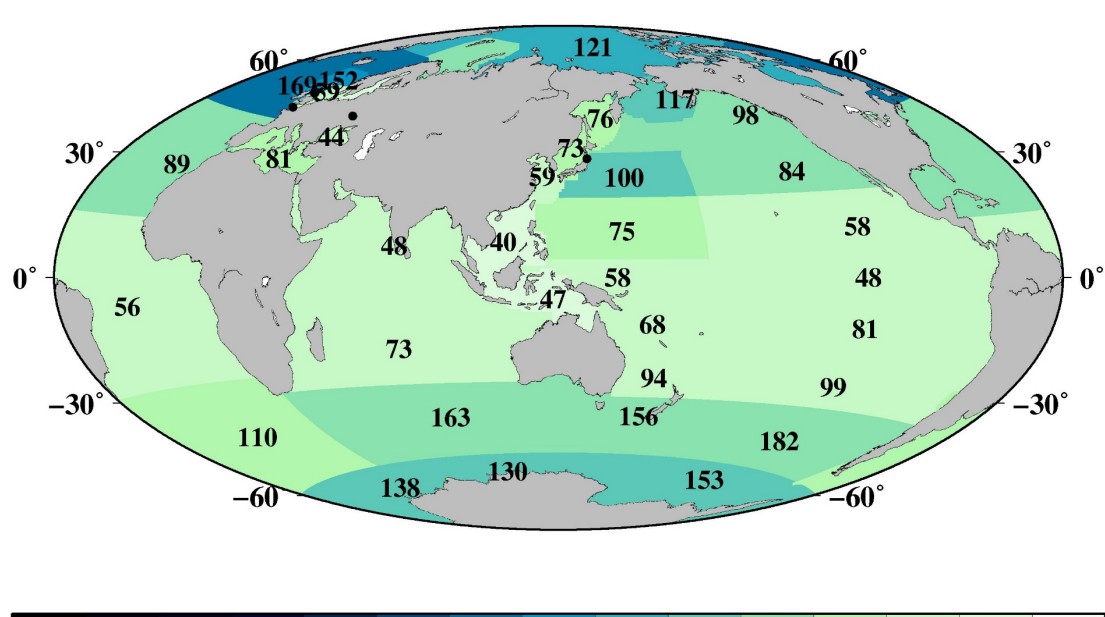

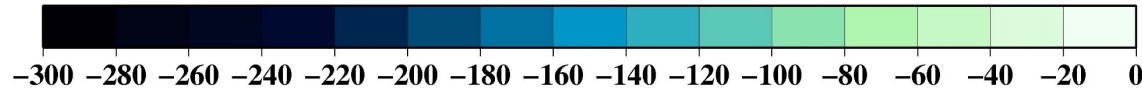






(b)




(c)




**Figure 10: Average mixed layer depth in each box in the global ocean. (a) Global, (b) North Atlantic Ocean and its marginal sea, and (c) Irish Sea. The unit is m. The "-" means that there is no mixed layer depth data.**



Table 5. $^{137}$Cs inventory in each box in the global ocean.

| Box Area | Area ($10^6$km$^2$) | Mixed Layer Depth (m) | 1970 | 1975 | 1980 | 1985 | 1990 | 1995 | 2000 | 2005 | 2010 | 2015 |
|---|---|---|---|---|---|---|---|---|---|---|---|---|
| 1 subarctic North Pacific Ocean | 10.66 | 98 | 10.4 | 7.3 | 5.1 | 3.5 | 3.4 | 2.8 | 2.3 | 1.9 | 1.6 | 2.5 |
| 2 western North Pacific Ocean | 7.14 | 100 | 6.3 | 5.1 | 4.2 | 3.4 | 2.5 | 2.0 | 1.6 | 1.3 | 1.1 | 2.1 |
| 3 eastern North Pacific Ocean | 9.85 | 84 | 12.2 | 8.2 | 5.5 | 3.7 | 3.0 | 2.5 | 2.0 | 1.7 | 1.4 | 2.2 |
| 4 western subtropical North Pacific Ocean | 13.41 | 75 | 7.1 | 6.4 | 5.8 | 5.2 | 2.9 | 2.6 | 2.2 | 1.9 | 1.7 | 1.3 |
| 5 eastern subtropical North Pacific Ocean | 20.46 | 58 | 11.6 | 11.6 | 11.6 | 11.6 | 3.5 | 3.0 | 2.6 | 2.2 | 1.9 | 1.6 |
| 6 western equatorial Pacific Ocean | 6.12 | 58 | 1.9 | 1.9 | 1.9 | 1.9 | 1.1 | 0.8 | 0.7 | 0.5 | 0.4 | 0.3 |
| 7 eastern equatorial Pacific Ocean | 12.34 | 48 | 2.6 | 2.6 | 2.6 | 2.6 | 1.4 | 1.3 | 1.1 | 0.9 | 0.8 | 0.7 |
| 8 western subtropical South Pacific Ocean | 7.81 | 68 | 7.2 | 4.8 | 3.2 | 2.2 | 1.5 | 1.0 | 0.7 | 0.4 | 0.3 | 0.2 |
| 9 eastern subtropical South Pacific Ocean | 24.91 | 81 | 6.0 | 10.5 | 10.5 | 4.9 | 4.9 | 4.3 | 3.7 | 3.3 | 2.8 | 2.0 |
| 10 western South Pacific Ocean | 4.31 | 94 | 3.8 | 1.7 | 1.9 | 1.4 | 1.0 | 0.7 | 0.6 | 0.6 | 0.5 | 0.4 |
| 11 eastern South Pacific Ocean | 16.86 | 99 | 5.2 | 5.5 | 4.3 | 3.3 | 2.6 | 2.2 | 2.0 | 1.9 | 1.8 | 1.7 |
| 12 eastern Southern Ocean | 16.92 | 182 | 10.7 | 8.4 | 6.6 | 5.2 | 4.0 | 3.2 | 2.5 | 2.0 | 1.5 | 1.2 |
| 13 Pacific sector of Antarctic | 9.87 | 153 | 4.1 | 2.1 | 1.1 | 0.6 | 0.3 | 0.2 | 0.1 | 0.0 | 0.0 | 0.0 |
| 14 Sea of Japan | 1.04 | 73 | 0.7 | 0.5 | 0.4 | 0.3 | 0.3 | 0.2 | 0.2 | 0.1 | 0.1 | 0.2 |
| 15 Arabian Sea | 20.23 | 48 | 4.0 | 3.8 | 3.7 | 3.4 | 2.8 | 2.3 | 1.9 | - | - | - |
| 16 Indian Ocean | 23.25 | 73 | 7.9 | 7.9 | 7.9 | 6.6 | 5.1 | 4.0 | 3.1 | 2.4 | 1.9 | 1.4 |
| 17 Southern Ocean | 26.55 | 163 | 15.2 | 15.2 | 15.2 | 11.4 | 8.9 | 7.0 | 5.5 | 4.3 | 3.4 | 2.7 |
| 18* Arctic Ocean | 12.03 | 121 | 1.7 | 8.8 | 16.5 | 12.0 | 9.7 | 4.7 | 2.1 | 5.3 | 2.4 | 2.2 |
| 19 Middle Southern Ocean | 5.09 | 156 | 3.5 | 2.6 | 1.9 | 1.5 | 1.1 | 0.8 | 0.6 | 0.5 | 0.3 | 0.3 |
| 20* Barents Sea and Coast of Norway | 1.85 | 81 | - | - | 5.0 | 3.9 | 3.0 | 0.7 | 0.7 | 0.6 | 0.4 | 0.3 |
| 21* Baltic Sea | 0.41 | 33 | - | 0.4 | 0.5 | 0.3 | 1.3 | 1.0 | 0.7 | 0.5 | 0.4 | 0.3 |
| 22* North Sea | 0.43 | 59 | 2.4 | 1.6 | 7.4 | 2.7 | 0.3 | 0.2 | 0.2 | 0.1 | 0.1 | 0.1 |
| 23.1* Irish Sea | 0.01 | 64 | 2.3 | 15.2 | 6.6 | 2.3 | 0.3 | 0.1 | 0.1 | 0.05 | 0.03 | 0.02 |
| 23.2* | 0.03 | 77 | 1.5 | 2.8 | 3.3 | 1.4 | 0.2 | 0.1 | 0.1 | 0.05 | 0.03 | 0.01 |
| 23.3* | 0.05 | 97 | 0.2 | 0.4 | 0.2 | 0.3 | 0.1 | 0.04 | 0.03 | 0.1 | 0.03 | 0.02 |
| 23.4* | 0.04 | 120 | 0.1 | 0.1 | 0.1 | 0.04 | 0.01 | 0.01 | 0.01 | 0.01 | 0.01 | 0.01 |
| 23.5* | 0.01 | - | - | - | - | - | - | - | - | - | - | - |
| 24* English Channel | 0.08 | 57 | 0.1 | 0.0 | 0.0 | 0.0 | 0.0 | 0.0 | 0.1 | 0.0 | 0.0 | 0.0 |
| 25.1* Northern North Atlantic Ocean | 0.36 | 152 | 2.3 | 9.1 | 15.3 | 5.9 | 0.4 | 0.3 | 0.2 | 0.2 | 0.1 | 0.1 |
| 25.2* | 8.59 | 169 | 11.8 | 16.7 | 23.8 | 13.1 | 7.4 | 4.5 | 7.4 | 2.8 | 2.3 | 1.5 |
| 26 Black Sea | 0.46 | 44 | - | - | - | - | 1.1 | 0.7 | 0.5 | 0.3 | 0.2 | 0.1 |
| 27* Mediterranean Sea | 2.51 | 81 | - | 1.0 | 1.0 | 0.9 | 0.9 | 0.7 | 0.6 | 0.5 | 0.3 | 0.2 |
| 28 North Atlantic Ocean | 23.03 | 89 | 17.8 | 17.8 | 17.8 | 6.4 | 5.1 | 4.2 | 3.4 | 2.7 | 2.2 | 1.8 |
| 29 Central Atlantic Ocean | 29.58 | 56 | 13.3 | 9.9 | 7.4 | 5.5 | 2.7 | 2.5 | 2.3 | 2.1 | 1.9 | 1.8 |
| 30 South Atlantic Ocean | 21.48 | 110 | 3.7 | 3.4 | 3.1 | 2.8 | 2.6 | 2.3 | 2.1 | 1.9 | 1.8 | 1.6 |
| 31 Sea of Okhotsk | 1.61 | 76 | 1.5 | 0.9 | 0.6 | 0.4 | 0.3 | 0.2 | 0.2 | 0.2 | 0.1 | 0.1 |
| 32 East China Sea | 1.18 | 59 | 0.5 | 0.4 | 0.4 | 0.3 | 0.2 | 0.2 | 0.2 | 0.1 | 0.1 | 0.1 |
| 33 South China Sea | 4.02 | 40 | 0.9 | 1.3 | 1.6 | 0.6 | 0.4 | 0.3 | 0.2 | 0.1 | 0.1 | 0.0 |
| 34 Bering Sea | 2.28 | 117 | 2.1 | 1.6 | 1.2 | 0.9 | 0.7 | 0.5 | 0.4 | 0.3 | 0.2 | 0.6 |
| 35 Indonesian Archipelago | 3.27 | 47 | 0.9 | 0.8 | 0.7 | 0.7 | 0.6 | 0.6 | 0.5 | 0.5 | 0.4 | 0.4 |
| 36 Atlantic sector of Antarctic | 5.61 | 138 | 1.2 | 1.0 | 0.8 | 0.6 | 0.5 | 0.4 | 0.3 | 0.3 | 0.2 | 0.2 |
| 37 Indian sector of Antarctic | 5.31 | 130 | 1.4 | 1.0 | 0.7 | 0.5 | 0.3 | 0.2 | 0.2 | 0.1 | 0.1 | 0.1 |

\* : 0.5yr average value without curve fitting

-: There is no available data.



By using these mixed layer depths, the estimated $^{137}$Cs inventory in the surface seawater from 1970 to 2015 every 5 years is shown in Figs. 11-13. The $^{137}$Cs inventory in the surface seawater is also listed in Table 5. In the Pacific Ocean, a higher $^{137}$Cs inventory exists in the subarctic North Pacific Ocean (10.4 PBq), western and eastern North Pacific Ocean (6.3 PBq and 12.2 PBq), and subtropical western and eastern North Pacific Ocean (7.1 PBq and 11.6 PBq) in 1970 (Fig.11a). In particular, the $^{137}$Cs inventory in the eastern North Pacific Ocean/subtropical eastern North Pacific Ocean is larger than that in

the western regions. In the South Pacific Ocean, a higher $^{137}$Cs inventory is observed in the western subtropical South Pacific Ocean (7.2 PBq) and eastern Southern Ocean (10.7 PBq) in 1970. After 1975, the $^{137}$Cs inventory in the eastern part is larger than that in the western part in the South Pacific Ocean. The $^{137}$Cs inventory in the surface seawater in the Southern Hemisphere is the highest in the Pacific Ocean, followed by the Indian Ocean, and it is the lowest in the Atlantic Ocean. In the Indian Ocean, the $^{137}$Cs inventory has a latitudinal gradient that is higher than that in the Southern Ocean (15.2 PBq) in 1970. These

latitudinal gradients of the $^{137}$Cs inventory in the surface mixed layer continued until 2015.

In the northern North Atlantic Ocean, its marginal seas, and the Arctic Ocean (Figs. 12, 13), the $^{137}$Cs inventory is strongly influenced by the discharged $^{137}$Cs from fuel reprocessing plants in the Irish Sea and English Chanel, as well as the global fallout from large-scale weapon tests (Northern North Atlantic Ocean.2 and Arctic Ocean). In the Irish Sea, the maximum $^{137}$Cs inventory occurred in the Irish Sea.1 (15.2 PBq) in 1975. The $^{137}$Cs discharged into the Irish Sea.1 was

transported into the Irish Sea.2, followed by transport to the northern North Atlantic Ocean.1, North Sea, and Barents Sea and coast of Norway. The $^{137}$Cs discharged from the La Hague was also transported into the North Sea.






(a)

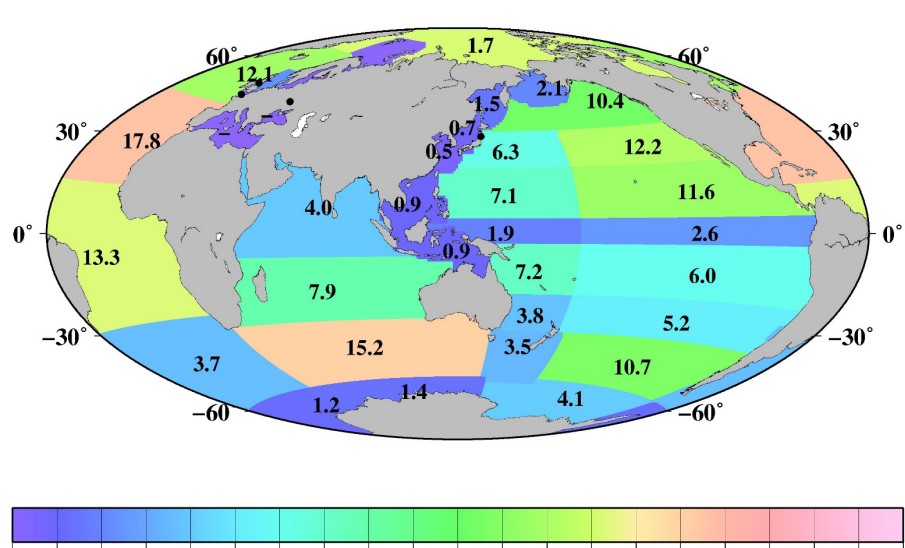


(b)

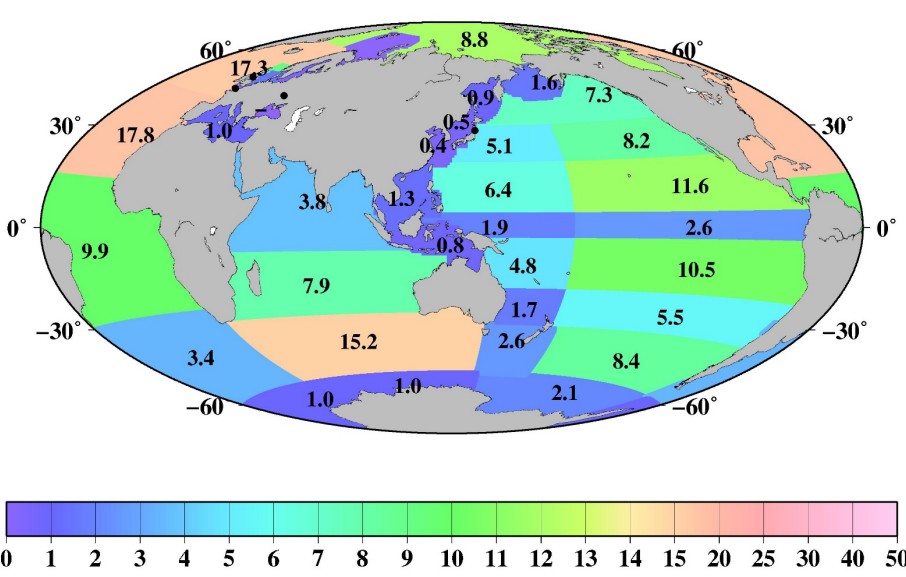



(c)

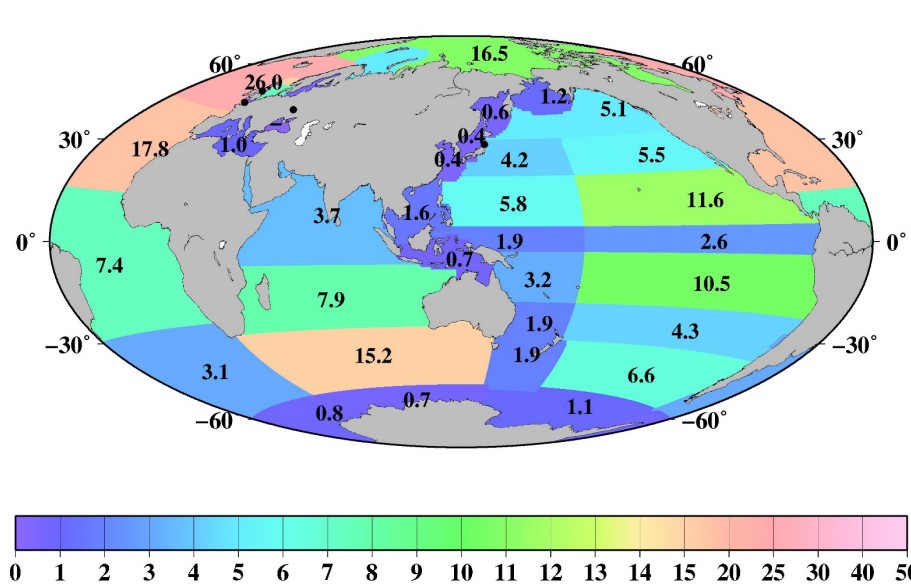

(d)

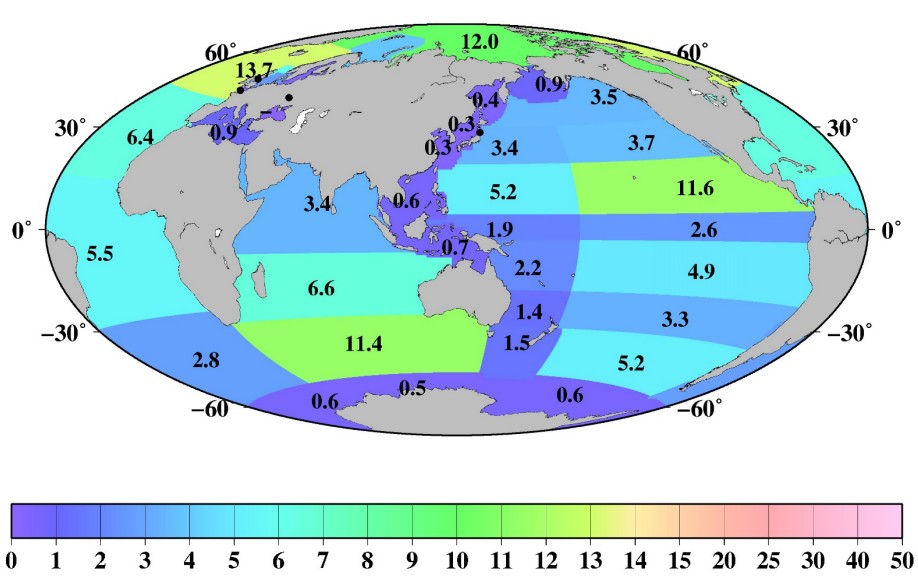



(e)

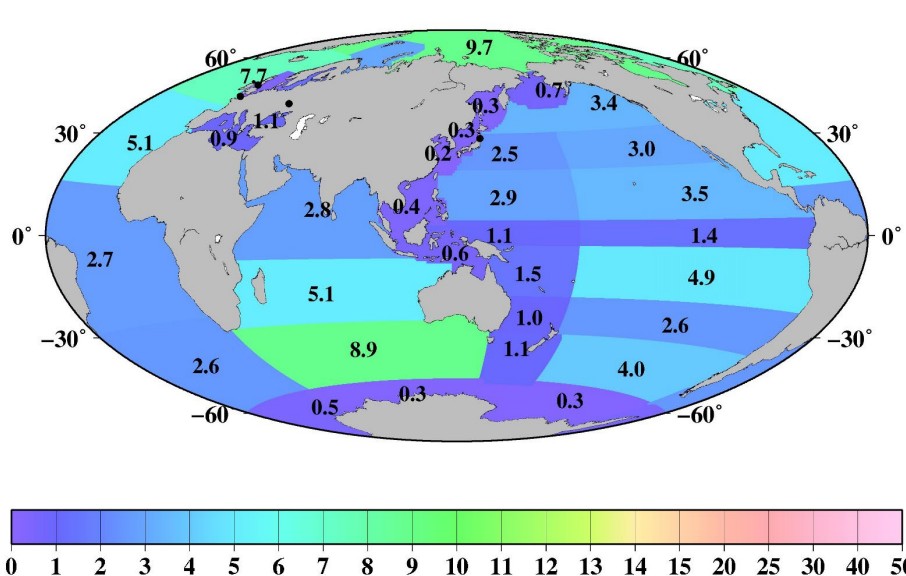

(f)

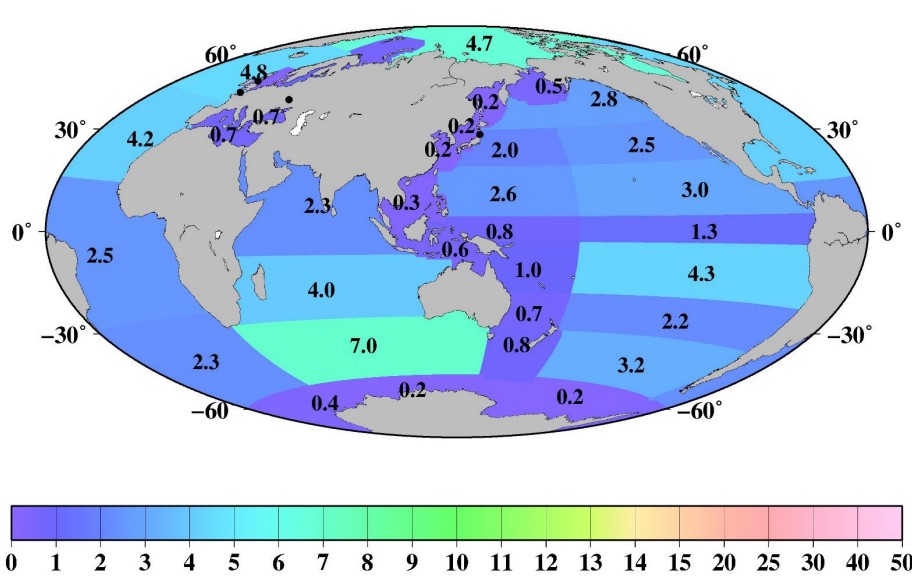




(g)

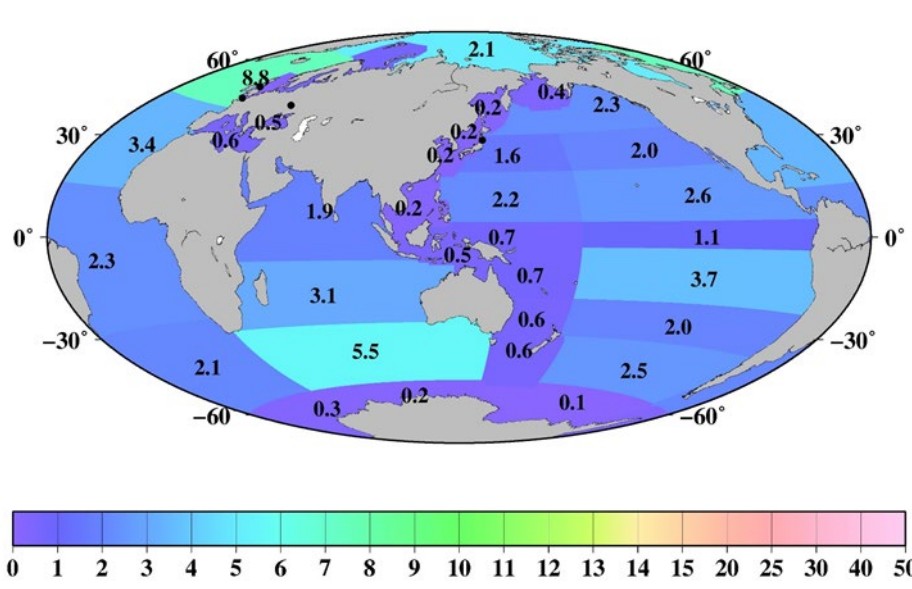

(h)

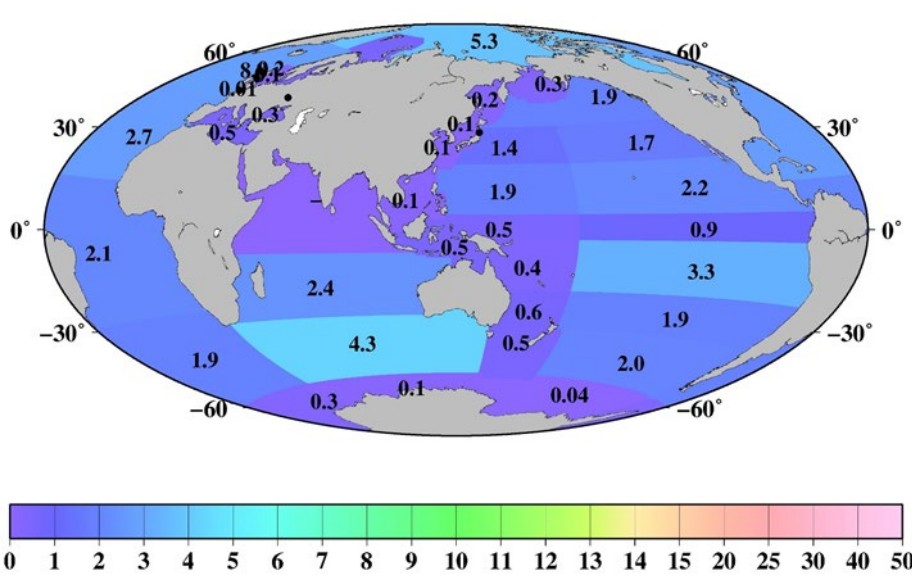



(i)

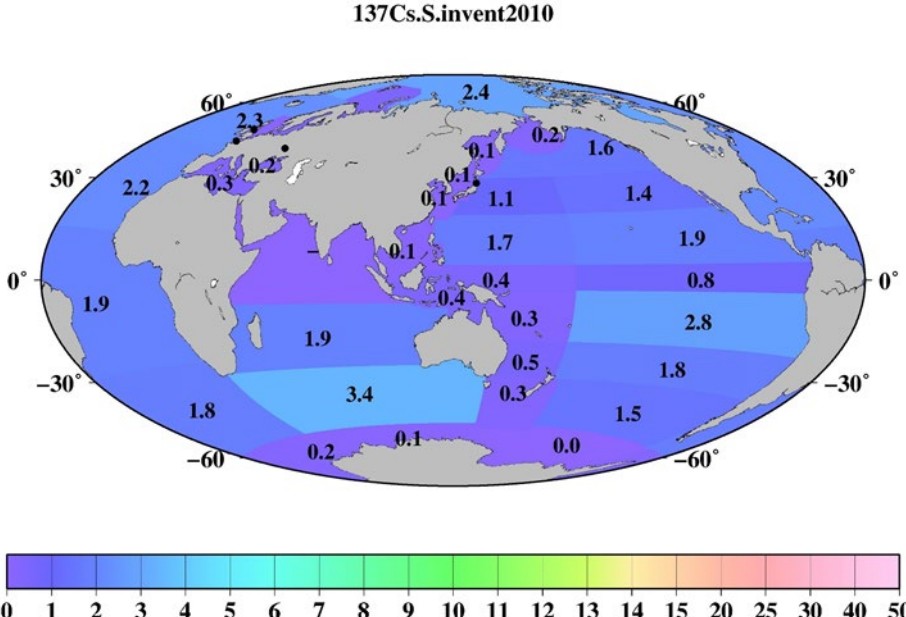

(j)




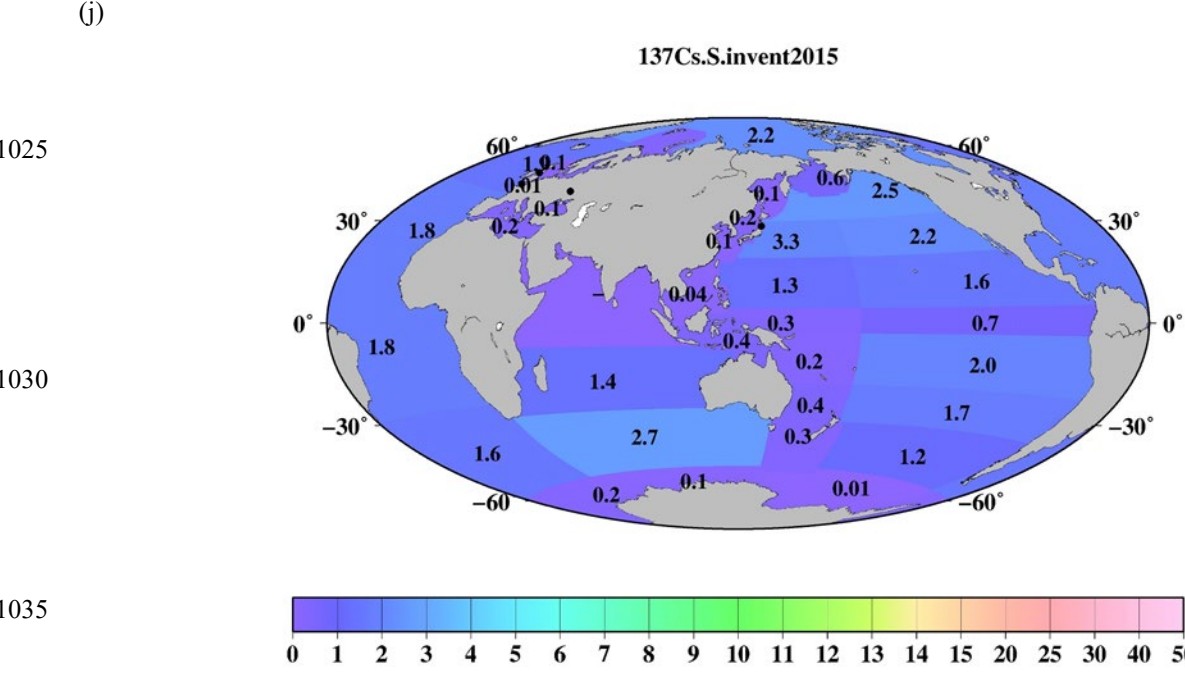

**Figure 11: Horizontal distributions of the ¹³⁷Cs inventory in the surface mixed layer in the global ocean.**



(a)


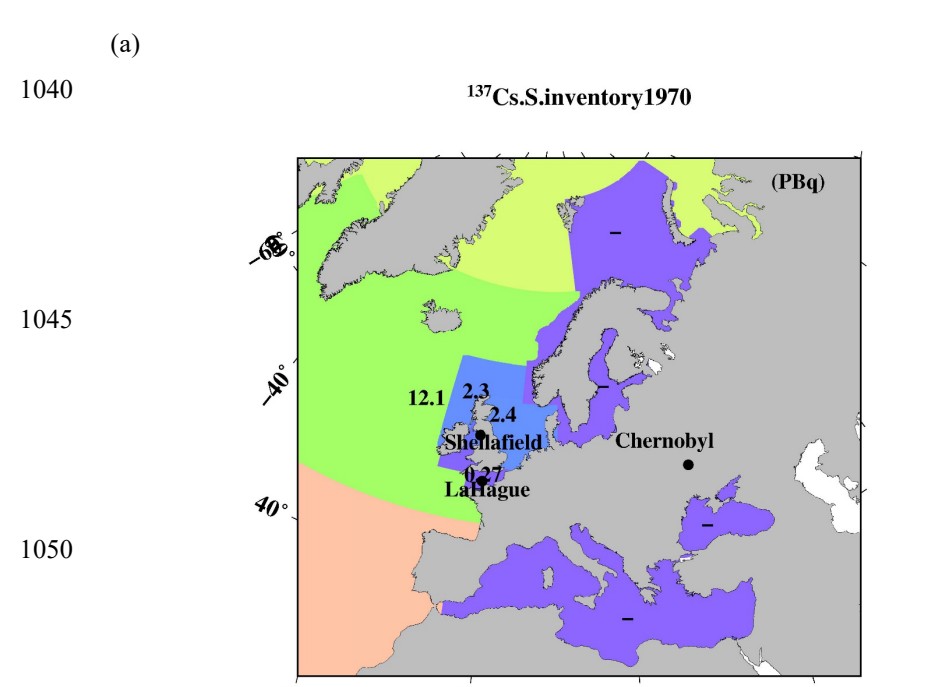



(b)

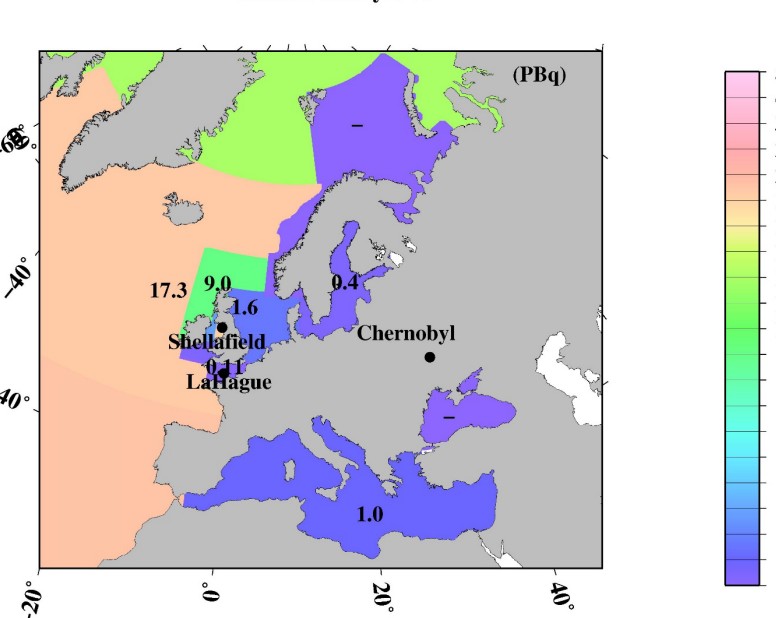



(c)



(d)



(e)

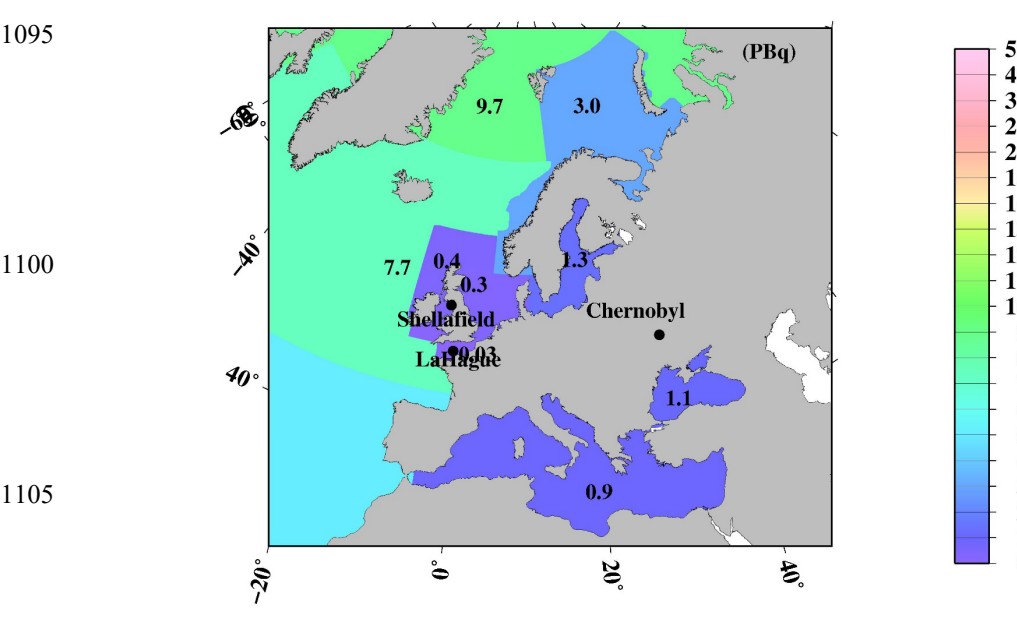

(f)

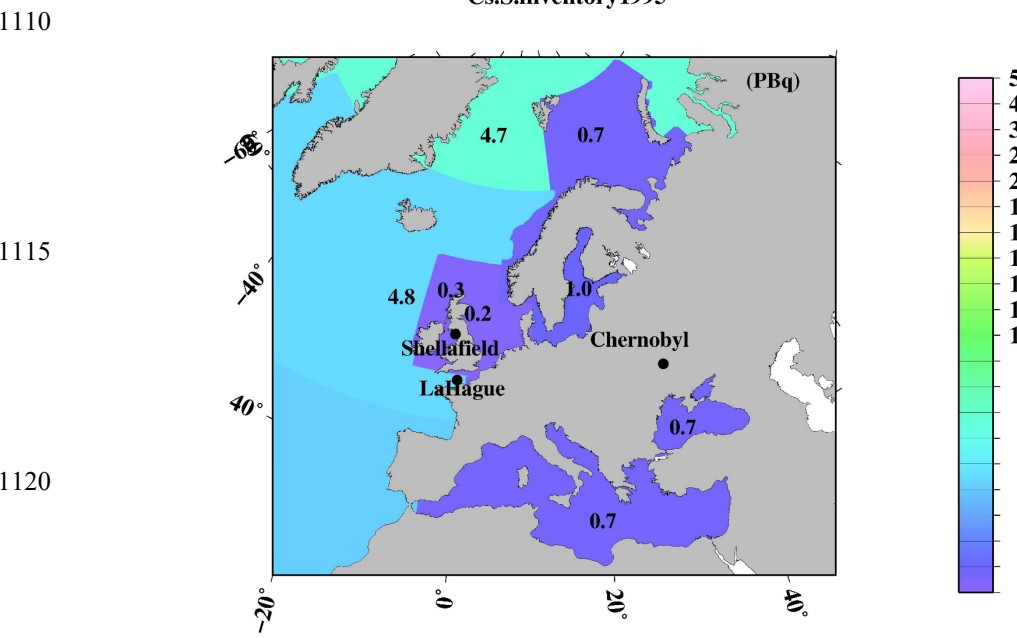

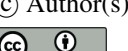



(g)

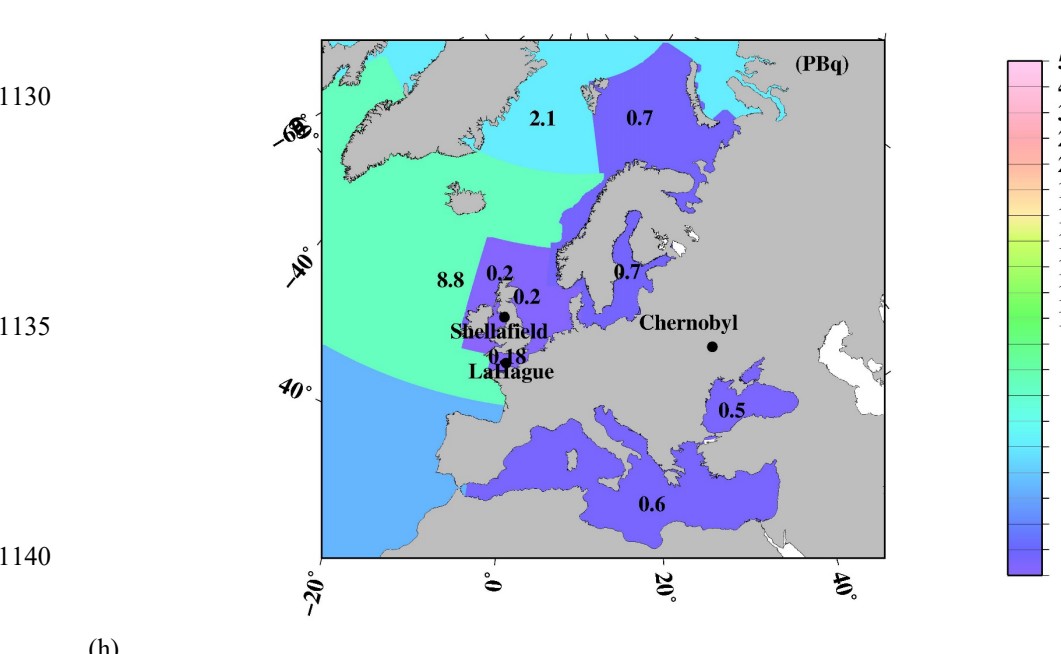

(h)

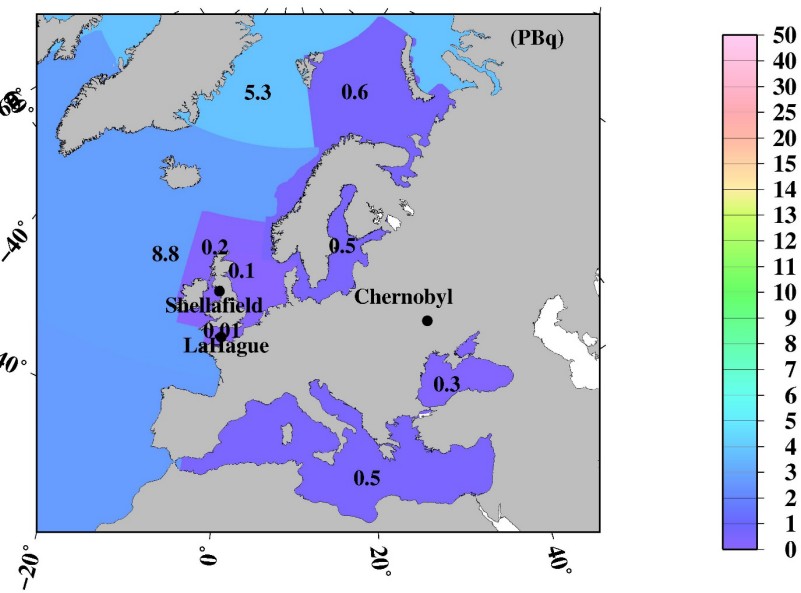





(i)

**<sup>137</sup>Cs.S.inventory2010**

![Map of 137Cs surface inventory 2010 over Europe and North Atlantic with values: 2.4, 0.4, 2.3, 0.1, 0.1, 0.4, 0.01, 0.2, 0.3. Shellafield, LaHague, Chernobyl locations marked. (PBq)]

(j)

**<sup>137</sup>Cs.S.inventory2015**

![Map of 137Cs surface inventory 2015 over Europe and North Atlantic with values: 2.2, 0.3, 1.9, 0.1, 0.06, 0.3, 0.01, 0.1, 0.2. Shellafield, LaHague, Chernobyl locations marked. (PBq)]

**Figure 12: Horizontal distributions of <sup>137</sup>Cs inventory in the surface mixed layer in the northern North Pacific Ocean and its marginal seas.**



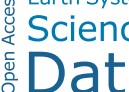

(a)

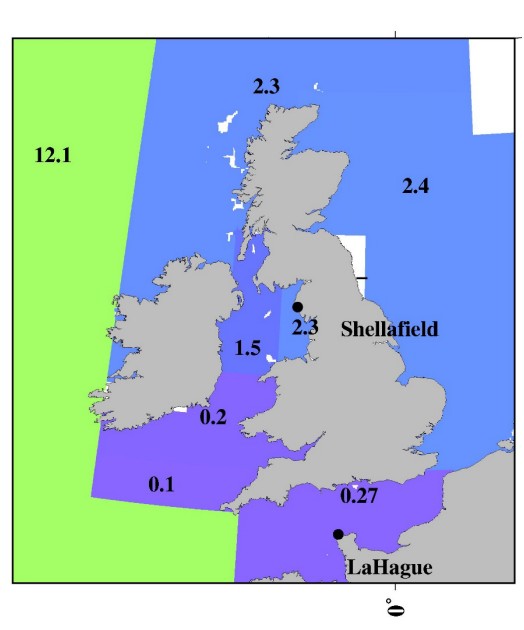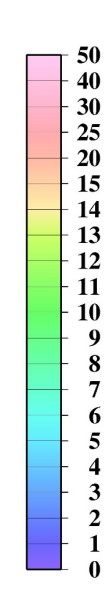

(b)

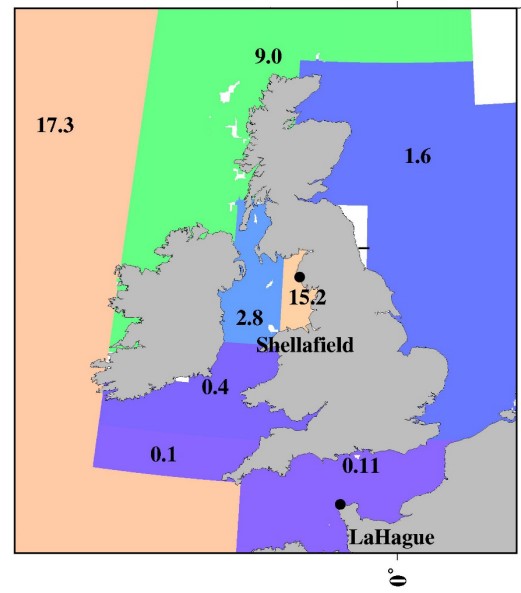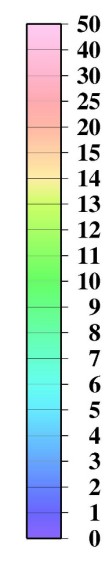



(c)

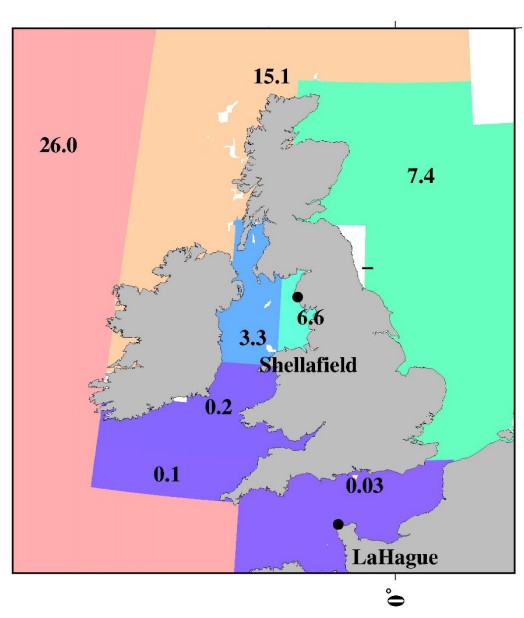
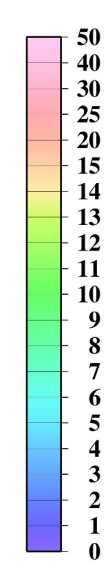

(d)

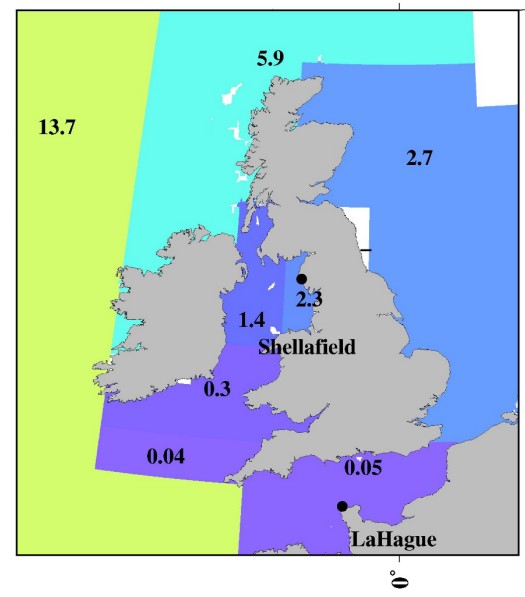
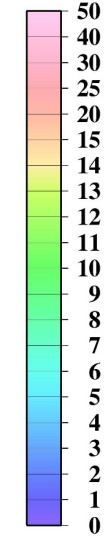



(e)

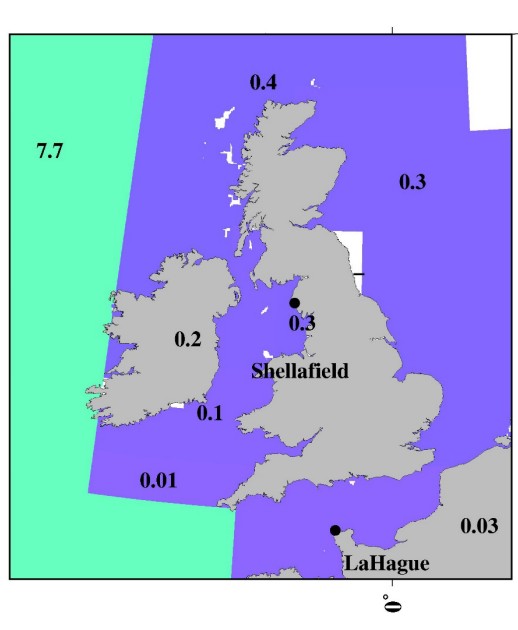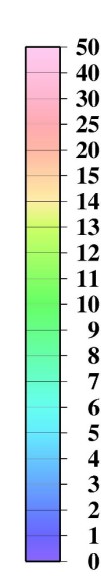

(f)

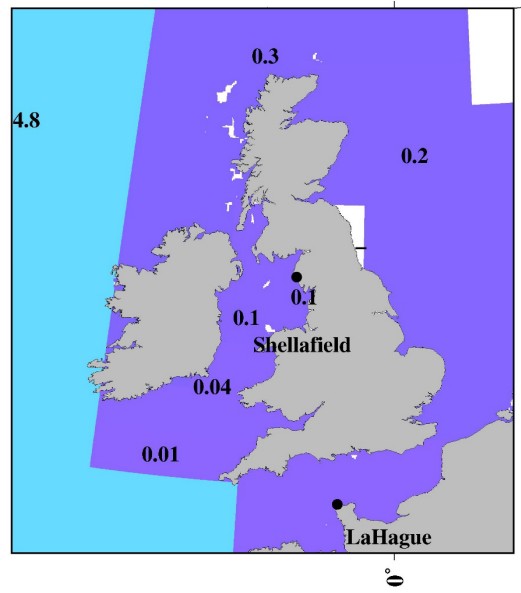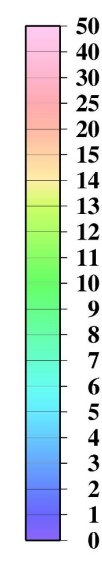





(g)

**¹³⁷Cs.S.inventory2010**



(h)

**¹³⁷Cs.S.inventory2015**

**Figure 13: Horizontal distributions of the ¹³⁷Cs inventory in the surface mixed layer in the Irish Sea and English Channel.**



Fig. 14 shows the time variation in [137]Cs inventories in the surface mixed layer during 1970-2015. The [137]Cs inventory in the surface mixed layer in the global ocean in 1970 was estimated to be 187 ± 26 PBq. This result indicates that 32% of the deposited [137]Cs remained in the surface mixed layer; in other words, 68% of the deposited [137]Cs was transported below the surface mixed layer on a decadal scale in 1970. In 1970, the [137]Cs inventory was the largest in the North Pacific Ocean, followed by the North Atlantic Ocean and the marginal sea, Atlantic Ocean, and South Pacific Ocean. The [137]Cs inventories

increase until 1980, and the inventory is estimated to be 201±28 PBq in 1975 and 210±12 PBq in 1980 due to the discharge of [137]Cs from the Sellafield and La Hague reprocessing plants. According to the estimation by Aaklog (2003), approximately 38 PBq (32 PBq until 1980) and 0.96 PBq (0.70 PBq until 1980) of [137]Cs were discharged from the Sellafield and La Hague plants from 1970 to 1998, respectively. The contribution from the nuclear fuel reprocessing plants and large-scale nuclear weapons tests (39%) observed in the Arctic Ocean and the northern North Atlantic Ocean and its marginal sea (the Arctic Ocean, Barents

Sea and coast of Norway, Baltic Sea, North Sea, northern North Atlantic Ocean, Irish Sea, and the English Channel) resulted in a large [137]Cs inventory in 1980. After 1980, the [137]Cs inventory decreased gradually and was estimated to be 35.1 ± 3.6 PBq in 2010, immediately before the F1NPS accident. Although the [137]Cs inventory decreased over time after 1980 until 2010, immediately before the F1NPS accident, the relative contributions of the [137]Cs inventory in the South Pacific Ocean, Indian Ocean, and Atlantic Ocean gradually increased and were estimated to be 21, 15, and 17% in 2010, respectively. After the

F1NPS accident, the [137]Cs inventory increased and was estimated to be 48.1 ± 12.1 PBq.

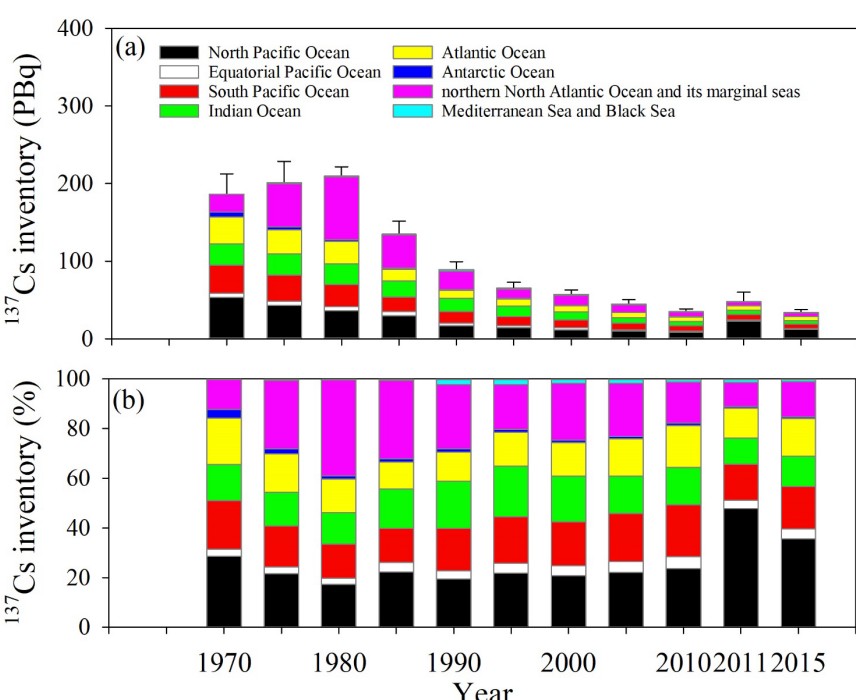

Figure 14: Temporal variations in the [137]Cs inventory every 5 years in the global ocean surface seawater.



## 4 Discussion

### 4.1 Basin-scale transport of $^{137}$Cs in surface seawater in the North Pacific Ocean, its marginal seas, and the equatorial Pacific Ocean


Fig. 15 shows the spatiotemporal variations in the $^{137}$Cs density in the surface mixed layer in the North Pacific Ocean, subtropical North Pacific Ocean, equatorial Pacific Ocean, and subtropical western South Pacific Ocean. In the western North Pacific Ocean, the $^{137}$Cs density decreases exponentially. However, in the eastern Pacific Ocean, the $^{137}$Cs density increased until the mid-1960s and then decreased exponentially. The time lag that reached the maximum value was caused by horizontal

transport from the western North Pacific Ocean and accumulated in the eastern North Pacific Ocean (Inomata et al., 2012). In the subtropical western and eastern North Pacific Ocean, the $^{137}$Cs density was almost constant in the 1970s and the 1980s. After the 1990s, the $^{137}$Cs density decreased gradually. In the equatorial western Pacific Ocean and subtropical western South Pacific Ocean, the $^{137}$Cs density increased gradually until the 1980s and then decreased after the 1990s. As shown in Table 3, Tap2 in the eastern North Pacific Ocean, which is estimated to be 8.8 years, is shorter than that in the western North Pacific

Ocean (17.1 years). These results can be explained by the following interpretation: $^{137}$Cs deposited in the western North Pacific Ocean, mainly in the Kuroshio Current and Kuroshio Extension regions (Aoyama et al., 2006), was transported eastwards along the Kuroshio Current and Kuroshio Extension, after which it was transported via the southwards-flowing California Current and accumulated around the coastal sites of western America (Inomata et al., 2012). The shorter Tap2 in the eastern North Pacific Ocean suggests that outflowed $^{137}$Cs is larger than the flow of $^{137}$Cs in comparison with that in the western North

Pacific Ocean. The seawater with higher $^{137}$Cs activity concentrations moves southwards with subsidence associated with the North Pacific subtropical gyre, followed by westwards transport and subduction in the central and eastern subtropical North Pacific Ocean. The increased $^{137}$Cs density in the equatorial western Pacific Ocean and the subtropical western South Pacific Ocean, as shown in Fig. 15, would result in a supply of seawater with higher $^{137}$Cs activity concentrations.




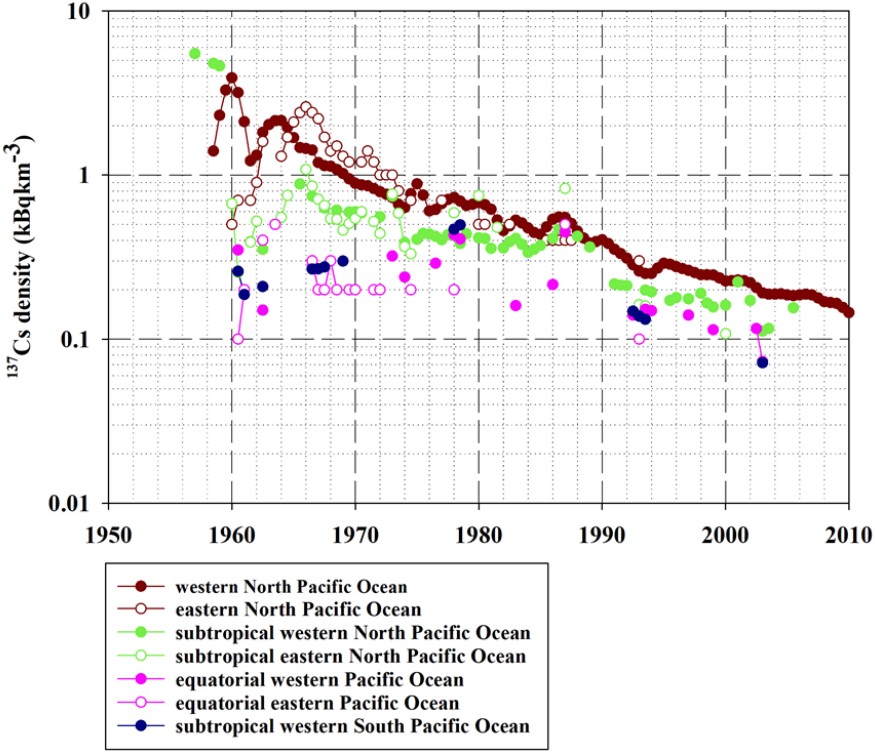

**Figure 15: $^{137}$Cs density in the surface mixed layer in the North Pacific Ocean, equatorial Pacific Ocean, and South Pacific Ocean.**


**4.2. Transport of $^{137}$Cs from the Pacific Ocean to the Indian Ocean via the Indonesian Sea throughflow**

As described in the previous sections, seawater with a relatively large $^{137}$Cs density ($^{137}$Cs activity concentrations) is transported westwards in the equatorial Pacific Ocean and the subtropical western South Pacific Ocean. It is known that the warm seawater in the equatorial Pacific Ocean is transported into the Indian Ocean by the wind-forcing circulation through

the Indonesian Archipelago, namely, the Indonesian throughflow (Gordon, 2005; Feng et al., 2018). Reportedly, the average volume of seawater in the Indonesian throughflow was estimated to be 15 Sv (1 Sv = $10^6$ m$^3$s$^{-1}$) (e.g., Gordon et al., 2010; Feng et al., 2018). The Indonesian throughflow consists of seawater derived from the North Pacific Ocean, South Pacific Ocean, and Antarctic Ocean. Approximately 9 Sv of seawater is transported to the Indian Ocean (Gordon et al., 2005). The transport of seawater from the Pacific Ocean into the Indian Ocean is revealed by the higher tritium concentrations in the seawater

transported from the North Pacific Ocean through the Makassae Strait with the Mindanao Current, whereas seawater with lower tritium concentrations is transported into the Indian Ocean from the South Pacific Ocean via the Hamahera Sea (Fin et





al., 1994). This suggests that most of the $^{137}$Cs in the surface seawater inflows into the Indian Ocean from the North Pacific Ocean. Furthermore, the South Indian Ocean is connected to the Atlantic Ocean around the Cape of Good Hope via the Agulhas Current (Sanchez-Cabeza et al., 2011) with an average seawater mass transport of 8.7 Sv, according to Stramma and England (1999).


In this section, we discuss the $^{137}$Cs inflow from the Pacific Ocean to the Indian Ocean, followed by the Atlantic Ocean. In the Indonesian Archipelago, the 0.5-yr average value of $^{137}$Cs in 2010 was 2.7 Bq m$^{-3}$ (Table 2, Fig. 7). This value is higher than those in the surrounding sea area, such as in the Eastern China Sea (1.7 Bq m$^{-3}$), western North Pacific Ocean (1.5 Bq m$^{-3}$), western subtropical North Pacific Ocean (1.7 Bqm$^{-3}$), and western equatorial Pacific Ocean (1.2 Bq m$^{-3}$). The 0.5-yr average value in the Indonesian Archipelago in 2010 decreased to approximately 53% compared to that in 1970. These decreasing rates are smaller than those in the surrounding sea area (78-93%).


Figure 16 shows the spatiotemporal variations in the $^{137}$Cs density in the regions related to the Indonesian throughflow (the South China Sea, Indonesian Archipelago, western subtropical South Pacific Ocean, Arabian Sea, and Indian Ocean). The $^{137}$Cs density in each box in the global surface seawater is also listed in Table 6. The $^{137}$Cs density in the western subtropical South Pacific Ocean (upstream region) is slightly higher or almost the same as that in the Indonesian Archipelago (downstream region). This result suggests that $^{137}$Cs derived from large-scale weapons tests in the western North Pacific Ocean flowed into the Indian Ocean by basin-scale transport. The $^{137}$Cs densities in the Indian Ocean (the Arabian Sea, Indian Ocean, and Southern Ocean) are almost similar to or higher than those in the Indonesian Archipelago. Furthermore, the $^{137}$Cs density in these regions decreased after the 1990s. It is likely that the main plume of $^{137}$Cs derived from the large-scale weapon tests is transported into the Indian Ocean, with a time scale of 20-30 years.



Although we used the average box value in this discussion, the signatures of $^{137}$Cs inflow from the North Pacific Ocean to the Indian Ocean via the Indonesian Archipelago were recognized by measurements. Evidently, higher concentrations were found at approximately 100°E in the subsurface layer, whereas lower concentrations were observed at approximately 70°E in the surface seawater (Povinec et al., 2011). In addition, $^{137}$Cs in the Indian Ocean is transported westwards at approximately 10–15°S latitude (Sanchez-Cabeza et al., 2011; Povinec et al., 2011).


Furthermore, several studies have found that $^{137}$Cs is transported into the South Atlantic Ocean via the Agulhas Current and then transported northwards with the Bengella Current (Sanchez-Cabeza et al., 2011; Strama and England, 1999). The transit times from the Pacific Ocean to the Atlantic Ocean via the Indian Ocean were estimated over four decades via model simulations (Tsumune et al., 2011). In this study, a slight increase in the $^{137}$Cs average values in the Central Atlantic Ocean and South Atlantic Ocean was detected in 2003, as shown in Fig. 2g. Furthermore, Tap3 values in the Indonesian Archipelago (36.7 yr) and South and Central Atlantic Ocean (43.5 and 37 years) are longer than those in the surrounding boxes. These results support the interpretation that the $^{137}$Cs deposited into the western North Pacific Ocean is transported into the equatorial Pacific Ocean, Indian Ocean, and Atlantic Ocean on an approximately four- to five-decade scale.







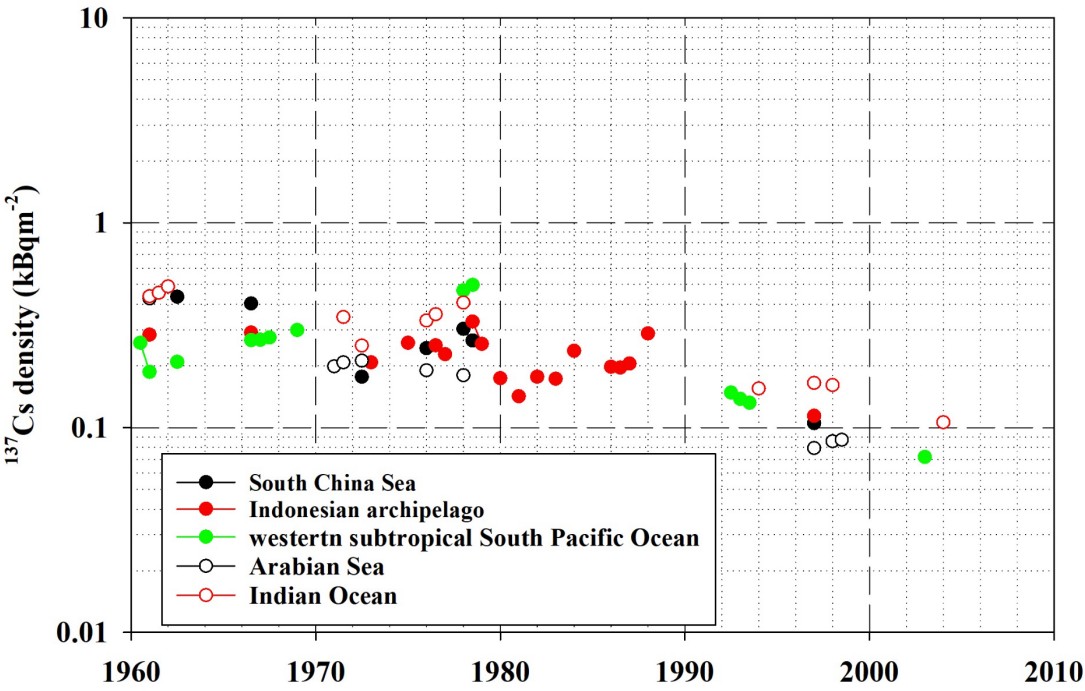

**Figure 16: $^{137}$Cs density in the Indonesian Archipelago and surrounding sea in the surface mixed layer.**






Table 6. $^{137}$Cs density in each box in the global ocean.

| Box | Area | $^{137}$Cs density (kBq m$^{-2}$) | | | | | | | | | |
|---|---|---|---|---|---|---|---|---|---|---|---|
| | | 1970 | 1975 | 1980 | 1985 | 1990 | 1995 | 2000 | 2005 | 2010 | 2015 |
| 1 | subarctic North Pacific Ocean | 0.98 | 0.68 | 0.47 | 0.33 | 0.32 | 0.26 | 0.22 | 0.18 | 0.15 | 0.24 |
| 2 | western North Pacific Ocean | 0.88 | 0.72 | 0.59 | 0.48 | 0.34 | 0.28 | 0.23 | 0.18 | 0.15 | 0.30 |
| 3 | eastern North Pacific Ocean | 1.24 | 0.84 | 0.56 | 0.38 | 0.31 | 0.25 | 0.21 | 0.17 | 0.14 | 0.23 |
| 4 | western subtropical North Pacific Ocean | 0.53 | 0.48 | 0.43 | 0.39 | 0.22 | 0.19 | 0.17 | 0.14 | 0.13 | 0.10 |
| 5 | eastern subtropical North Pacific Ocean | 0.57 | 0.57 | 0.57 | 0.57 | 0.17 | 0.15 | 0.12 | 0.11 | 0.09 | 0.08 |
| 6 | western equatorial Pacific Ocean | 0.31 | 0.31 | 0.31 | 0.31 | 0.17 | 0.14 | 0.11 | 0.09 | 0.07 | 0.05 |
| 7 | eastern equatorial Pacific Ocean | 0.21 | 0.21 | 0.21 | 0.21 | 0.12 | 0.10 | 0.09 | 0.08 | 0.07 | 0.06 |
| 8 | western subtropical South Pacific Ocean | 0.92 | 0.62 | 0.42 | 0.28 | 0.19 | 0.13 | 0.08 | 0.06 | 0.04 | 0.03 |
| 9 | eastern subtropical South Pacific Ocean | 0.24 | 0.42 | 0.42 | 0.20 | 0.20 | 0.17 | 0.15 | 0.13 | 0.11 | 0.08 |
| 10 | western South Pacific Ocean | 0.88 | 0.39 | 0.45 | 0.32 | 0.23 | 0.17 | 0.15 | 0.13 | 0.11 | 0.10 |
| 11 | eastern South Pacific Ocean | 0.31 | 0.33 | 0.25 | 0.20 | 0.15 | 0.13 | 0.12 | 0.11 | 0.11 | 0.10 |
| 12 | eastern Southern Ocean | 0.63 | 0.49 | 0.39 | 0.30 | 0.24 | 0.19 | 0.15 | 0.12 | 0.09 | 0.07 |
| 13 | Pacific sector of Antarctic | 0.41 | 0.22 | 0.11 | 0.06 | 0.03 | 0.02 | 0.01 | 0.00 | 0.00 | 0.00 |
| 14 | Japan Sea | 0.65 | 0.52 | 0.41 | 0.32 | 0.27 | 0.22 | 0.17 | 0.14 | 0.11 | 0.15 |
| 15 | Arabian Sea | 0.20 | 0.19 | 0.18 | 0.17 | 0.14 | 0.11 | 0.09 | - | - | - |
| 16 | Indian Ocean | 0.34 | 0.34 | 0.34 | 0.28 | 0.22 | 0.17 | 0.13 | 0.10 | 0.08 | 0.06 |
| 17 | Southern Ocean | 0.57 | 0.57 | 0.57 | 0.43 | 0.34 | 0.26 | 0.21 | 0.16 | 0.13 | 0.10 |
| 18 | Arctic Ocean | 0.15 | 0.73 | 1.37 | 1.00 | 0.81 | 0.39 | 0.17 | 0.44 | 0.20 | 0.18 |
| 19 | Middle Southern Ocean | 0.68 | 0.51 | 0.38 | 0.29 | 0.21 | 0.16 | 0.12 | 0.09 | 0.07 | 0.05 |
| 20 | Barents Sea and Coast of Norway | - | - | 2.70 | 2.09 | 1.60 | 0.36 | 0.40 | 0.31 | 0.23 | 0.17 |
| 21 | Baltic Sea | - | 1.00 | 1.26 | 0.72 | 3.21 | 2.38 | 1.76 | 1.30 | 0.96 | 0.71 |
| 22 | North Sea | 5.54 | 3.63 | 17.29 | 6.37 | 0.67 | 0.49 | 0.35 | 0.25 | 0.18 | 0.13 |
| 23.1 | Irish Sea | 281.40 | 1840.44 | 795.49 | 282.29 | 37.46 | 11.79 | 10.14 | 6.01 | 3.49 | 2.81 |
| 23.2 | Irish Sea | 49.13 | 94.19 | 112.09 | 47.36 | 7.41 | 3.22 | 1.96 | 1.55 | 1.08 | 0.41 |
| 23.3 | Irish Sea | 4.27 | 8.18 | 4.20 | 4.76 | 1.86 | 0.67 | 0.46 | 0.94 | 0.47 | 0.32 |
| 23.4 | Irish Sea | 1.83 | 1.54 | 1.25 | 0.97 | 0.35 | 0.34 | 0.32 | 0.30 | 0.29 | 0.27 |
| 23.5 | Irish Sea | - | - | - | - | - | - | - | - | - | - |
| 24 | English Channel | 3.39 | 1.39 | 0.35 | 0.68 | 0.38 | 0.32 | 2.23 | 0.10 | 0.10 | 0.08 |
| 25.1 | northern North Atlantic Ocean | 6.27 | 25.07 | 42.29 | 16.43 | 1.03 | 0.78 | 0.59 | 0.45 | 0.34 | 0.26 |
| 25.2 | northern North Atlantic Ocean | 1.41 | 2.01 | 3.03 | 1.59 | 0.90 | 0.56 | 1.03 | 0.32 | 0.26 | 0.22 |
| 26 | Black Sea | - | - | - | - | 2.46 | 1.58 | 1.01 | 0.65 | 0.42 | 0.27 |
| 27 | Mediterranean Sea | - | 0.41 | 0.41 | 0.35 | 0.35 | 0.29 | 0.23 | 0.21 | 0.11 | 0.10 |
| 28 | North Atlantic Ocean | 0.77 | 0.77 | 0.77 | 0.28 | 0.22 | 0.18 | 0.15 | 0.12 | 0.10 | 0.08 |
| 29 | Central Atlantic Ocean | 0.45 | 0.34 | 0.25 | 0.19 | 0.09 | 0.08 | 0.08 | 0.07 | 0.07 | 0.06 |
| 30 | South Atlantic Ocean | 0.17 | 0.16 | 0.14 | 0.13 | 0.12 | 0.11 | 0.10 | 0.09 | 0.08 | 0.07 |
| 31 | Sea of Okhotsk | 0.91 | 0.57 | 0.36 | 0.23 | 0.16 | 0.14 | 0.12 | 0.11 | 0.09 | 0.09 |
| 32 | Eastern China Sea | 0.47 | 0.38 | 0.31 | 0.25 | 0.20 | 0.17 | 0.14 | 0.12 | 0.10 | 0.11 |
| 33 | South China Sea | 0.23 | 0.32 | 0.41 | 0.16 | 0.10 | 0.06 | 0.04 | 0.03 | 0.02 | 0.01 |
| 34 | Berigng Sea | 0.92 | 0.70 | 0.53 | 0.41 | 0.31 | 0.24 | 0.18 | 0.14 | 0.11 | 0.26 |
| 35 | Indonesian Archipelago | 0.27 | 0.25 | 0.23 | 0.21 | 0.19 | 0.17 | 0.15 | 0.14 | 0.13 | 0.12 |
| 36 | Atlantic sector of Antarctic | 0.21 | 0.17 | 0.14 | 0.11 | 0.09 | 0.07 | 0.06 | 0.05 | 0.04 | 0.03 |
| 37 | Indian sector of Antarctic | 0.27 | 0.19 | 0.13 | 0.09 | 0.06 | 0.04 | 0.03 | 0.02 | 0.01 | 0.01 |

-: There is no data.

#: Estimated value based on the extrapolation of the trend line.




### 4.3. Recirculation of F1NPS [137]Cs associated with basin-scale transport in the North Pacific Ocean and its marginal sea

In this section, we focus on the temporal variations in the [137]Cs activity concentrations in the North Pacific Ocean and its marginal seas after 2011 to investigate the transport of [137]Cs from the F1NPS accident. Fig. 17a shows the 0.5-yr [137]Cs average values in the western and eastern North Pacific Ocean after 2011. These values were selected as typical cases because the main plume of F1NPS-[137]Cs is transported in these boxes and exists in many measurements. As described above, the 0.5-yr average values of [137]Cs decreased exponentially before the F1NPS accident. However, significantly high 0.5-yr average [137]Cs values were measured in the western North Pacific Ocean (8-59 Bq m$^{-3}$) and eastern North Pacific Ocean (2.4-7.6 Bq m$^{-3}$) in 2011/2012. Increases in [137]Cs in the subarctic North Pacific Ocean (15-28 Bq m$^{-3}$) and Bering Sea (4.2 Bq m$^{-3}$) were also observed in 2011/2012, although the data did not show these values due to the limited sample measurements. Slightly higher 0.5-yr average [137]Cs values were also observed in the Sea of Okhotsk (2.3 Bq m$^{-3}$; not shown in this figure) and the Japan Sea (1.6-1.9 Bq m$^{-3}$). These were caused by the atmospheric deposition of [137]Cs derived from the F1NPS accident, although the contribution of directly discharged [137]Cs from the F1NPS might be included in the values in the western North Pacific Ocean. After 2013, in the western North Pacific Ocean, the 0.5-yr average [137]Cs values decreased exponentially with seasonal variation. In the eastern North Pacific Ocean, the 0.5-yr average [137]Cs values increased after 2014 and reached a maximum in 2018. It is clear that the 0.5-yr average [137]Cs values in the eastern North Pacific Ocean were higher than those in the western region in 2018/2019. With the optimal interpretation analysis in Inomata et al. (2016), the main plume of F1NPS-[137]Cs exists in the centre in the North Pacific Ocean (longitude range of 165°E-170°W and latitude range of 30-50°N), which corresponds to the subarctic, western, and eastern North Pacific Ocean boxes in this study in 2012. The zonal transport speed of F1NPS-[137]Cs was estimated to be approximately 8 cm s$^{-1}$ from March 2011 to March 2012 (Aoyama et al., 2013). The time lag of the maximum 0.5-yr average values in these regions could be explained by our previous interpretation based on the behaviour of [137]Cs released in the surface seawater by a large-scale nuclear weapons test: [137]Cs was transported eastwards in the North Pacific Ocean and reached to the west of the American continent. Increased mixed layer depth deeper in the western North Pacific Ocean and shallower in the eastern North Pacific Ocean would result in the difference in the 0.5-yr average [137]Cs values (Inomata et al., 2012). The seasonal variation in the surface mixed layer depth might reflect the 0.5-yr average [137]Cs values in the western North Pacific Ocean. In addition, note that the behaviour, such as the transport pattern, of F1NPS-[137]Cs is almost the same as that of [137]Cs derived from large-scale weapons tests in the 1950s-1960s (Aoyama et al., 2013, 2016a,b, 2019, 2020; Inomata et al., 2012, 2018).

A slight increase in the 0.5-yr average value of [137]Cs also occurred in the Eastern China Sea and Japan Sea after 2013 and reached maximum values in 2015/2016 (Fig. 17b). The [137]Cs activity concentrations in the Eastern China Sea and Japan Sea increased following the processes elucidated in our previous studies (Inomata et al., 2018). The increase in [137]Cs activity concentrations was first observed in the subsurface layer in 2012/2013 around southern Japan in the western North Pacific Ocean. Based on the potential temperature density ($\sigma_\theta$), the [137]Cs peak existed in the subtropical mode water. In the Eastern China Sea, the increase in the [137]Cs activity concentrations also started in subsurface seawater (140 m) in 2013, and [137]Cs

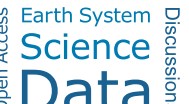

activity in the surface mixed layer (0-50 m) in the Eastern China Sea reached a maximum in 2014/2015 (Inomata et al., 2018).

Increased $^{137}$Cs in the Eastern China Sea is caused by the following processes: the $^{137}$Cs entrained into the subtropical mode water is transported westwards in the subsurface seawater and upwells along the continental shelf in the Eastern China Sea and the Kuroshio countercurrent around the meandering Kuroshio (Ito et al., 1994). Furthermore, the $^{134}$Cs/$^{137}$Cs ratios in subtropical mode water were almost the same as those in seawater in the Eastern China Sea and Japan Sea (Aoyama et al., 2017; Inomata et al., 2018). Then, the F1NPS-derived $^{137}$Cs flowed into the Japan Sea via the Tsushima Strait by the Tsushima

warm current and reached a maximum in 2015/2016. The propagation of F1NPS-derived $^{137}$Cs from the Eastern China Sea to the Japan Sea occurred over 1-2 years.






**Figure 17: Temporal variations in the 0.5-yr average values of surface ¹³⁷Cs in the North/equatorial Pacific Ocean and its marginal seas after 2011. (a) Western and eastern North Pacific Ocean, (b) Japan Sea and Eastern China Sea.**



### 4.4 Estimation of the amount of $^{137}$Cs outflow to the downstream box and/or below the surface mixed layer

The $^{137}$Cs deposited into the ocean surface is transported via advection in the surface seawater, then is transported to
deep water below the mixed layer depth, where it undergoes radioactive decay (T$_{1/2}$ = 30.17 yr). In 1970, $^{137}$Cs existing in the
surface seawater in the global ocean was estimated to be 187 ± 26 PBq. This value corresponds to 32% of the deposited $^{137}$Cs
until 1970, although the $^{137}$Cs released from reprocessing plants is included. The remaining approximately 68% of deposited
$^{137}$Cs, which is estimated to be 577 ± 60 PBq, would be transported downwards below the surface mixed layer in the global
ocean on a decadal timescale. According to the estimation by using a model simulation (Kamidaira et al., 2015), the amount
of F1NPS-$^{137}$Cs transported below the surface mixed layer within 4 months is almost half.

Fig. 18 shows the horizontal distributions of the inflow/outflow $^{137}$Cs amount in the surface mixed layer based on the
$^{137}$Cs deposition amount until the 1$^{st}$ of January 1970. Positive values (red) indicate that $^{137}$Cs inflows into the sea box from
the upstream box, whereas negative values (blue) indicate that $^{137}$Cs outflows into the downstream box and/or undergoes
downwards transport below the mixed layer depth. The amount of decrease was large in the North Pacific Ocean. The largest
decrease, which was estimated to be 61.2 PBq, occurred in the subarctic North Pacific Ocean. On the other hand, an increase
in the $^{137}$Cs amount occurred in the western subtropical South Pacific Ocean (2.8 PBq), eastern South Pacific Ocean (9.6 PBq),
middle Southern Ocean (2.6 PBq), Southern Ocean (12.4 PBq), and Antarctic Ocean (1.2-4.0 PBq). This suggests that some
of the $^{137}$Cs deposited into the Pacific Ocean was transported to the South Pacific Ocean (south of 40°N), followed by
movement to the Indian Ocean within 10-20 years. The outflow of $^{137}$Cs in the northern North Atlantic Ocean and North
Atlantic Ocean was also large (41.7 PBq and 52.0 PBq, respectively). The increase in $^{137}$Cs in the Irish Sea (Irish Sea.1; 2.3
PBq), North Sea (0.7 PBq), and northern North Atlantic Ocean (0.3 PBq) was due to the discharged $^{137}$Cs from the Sellafield
and La Hague plants (Fig. 18).

The temporal variation in the outflow/inflow pattern of the $^{137}$Cs amount in each box is shown in Figs. 19-21. These
main seawater transport routes using $^{137}$Cs behaviour in the surface seawater are summarized in Fig. 22. Fig. 19 shows the
inflowed or outflowed $^{137}$Cs amount in each box in the global ocean from 1975 to 2015 at 5-year intervals. The inflowed or
outflowed $^{137}$Cs amount corresponds to the sum of the $^{137}$Cs amount for the previous five years. In 1975, 1980 and 1985, the
values in the subarctic, western, and eastern North Pacific Ocean were negative (-0.04−-2.7 PBq; -0.03−-1.8 PBq; and -0.03−
-1.2 PBq, respectively), whereas the subtropical North Pacific Ocean and equatorial Pacific Ocean showed positive values
(0.1−1.3 PBq; 0.08−1.3 PBq; and 0.07−1.3 PBq, respectively). The inflowed $^{137}$Cs also occurred in the subtropical eastern
South Pacific Ocean (5.2 and 1.1 PBq) in 1975 and 1980 and the eastern South Pacific Ocean (0.8 PBq) in 1980. These results
can be explained by the following: $^{137}$Cs deposited in the surface mixed layer in the western North Pacific Ocean was
transported eastwards and accumulated into the eastern subtropical Pacific Ocean. Then, these were transported southwards
with subsidence and westwards in the equatorial Pacific Ocean. Considering the seawater current system, $^{137}$Cs moved
southwards due to subduction in the eastern subtropical North Pacific Ocean and upwelled in the western/eastern equatorial
Pacific Ocean. The negative values (-0.5−-1.6 PBq for 1975; -0.4−-1.1 PBq for 1980) in the western subtropical South



Pacific Ocean, western South Pacific Ocean, and eastern Southern Ocean and positive values (0.3−1.7 PBq for 1975; 0.3−1.7 PBq for 1980) in the Arabian Ocean, Indian Ocean and Southern Ocean would result in the transport of $^{137}$Cs from the Pacific Ocean into the Indian Ocean through the Indonesian Archipelago in 1975 and 1980. After 1990, the positive $^{137}$Cs values in the eastern/western subtropical North Pacific Ocean and equatorial Pacific Ocean became negative (Fig. 19d). This suggests that the main $^{137}$Cs plume derived from the large-scale nuclear weapon tests would pass through until 1990. In addition, note that a small amount of $^{137}$Cs inflowed into the South Atlantic Ocean in 1975 (0.1 PBq) and 1980 (0.01 PBq) (Fig. 19a,b). A small $^{137}$Cs increase also occurred in the Central Atlantic Ocean after 1995 (Fig. 19c). In 2015, increased $^{137}$Cs in the subarctic, western, and eastern North Pacific Ocean (0.02−1.2 PBq) would be caused by the $^{137}$Cs released from the F1NPS (Fig. 19i).

In the northern North Atlantic Ocean and its marginal seas (Fig. 20), the increase in the $^{137}$Cs amount due to the discharged $^{137}$Cs from the reprocessing plants was significant in the northern North Atlantic Ocean, North Sea, and Barents Sea and coast of Norway and transported to the Arctic Ocean (Fig. 20a-d). The contribution of the discharged $^{137}$Cs from reprocessing plants decreased after 1985. The contribution from the Chernobyl accident found in the Baltic Sea, Black Sea, and Mediterranean Sea in 1990 (Fig. 20d).

In the Irish Sea and English Chanel, the contributions of $^{137}$Cs released from reprocessing plants were large in 1975, and it appears that this $^{137}$Cs was transported to the northern North Atlantic Ocean and North Sea until 1980. In 1980, the $^{137}$Cs amount around the source region had negative values. After 1985, the contribution is negatively associated with the decreased discharged $^{137}$Cs amount (Fig. 21).





(a)

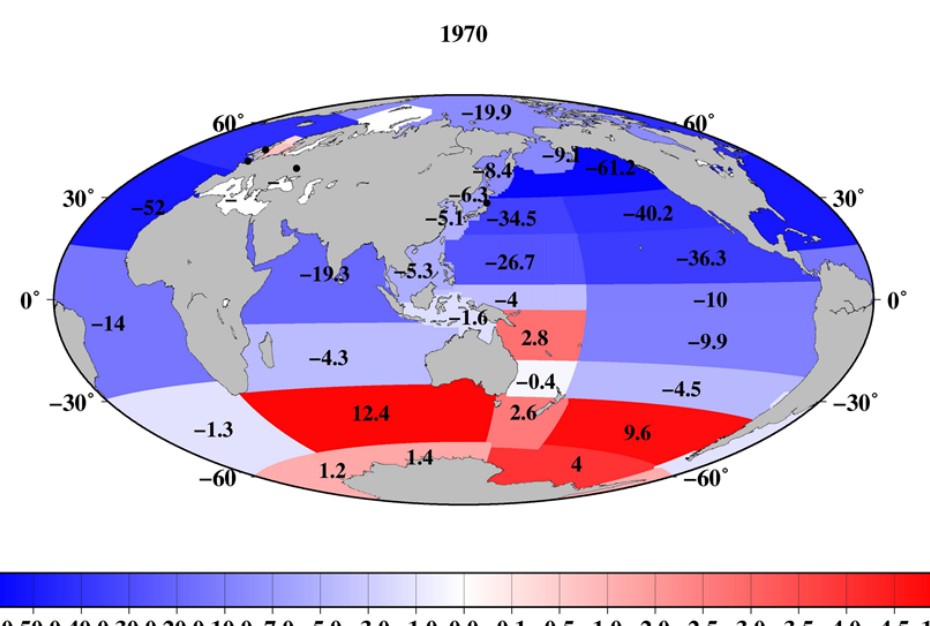

(b)




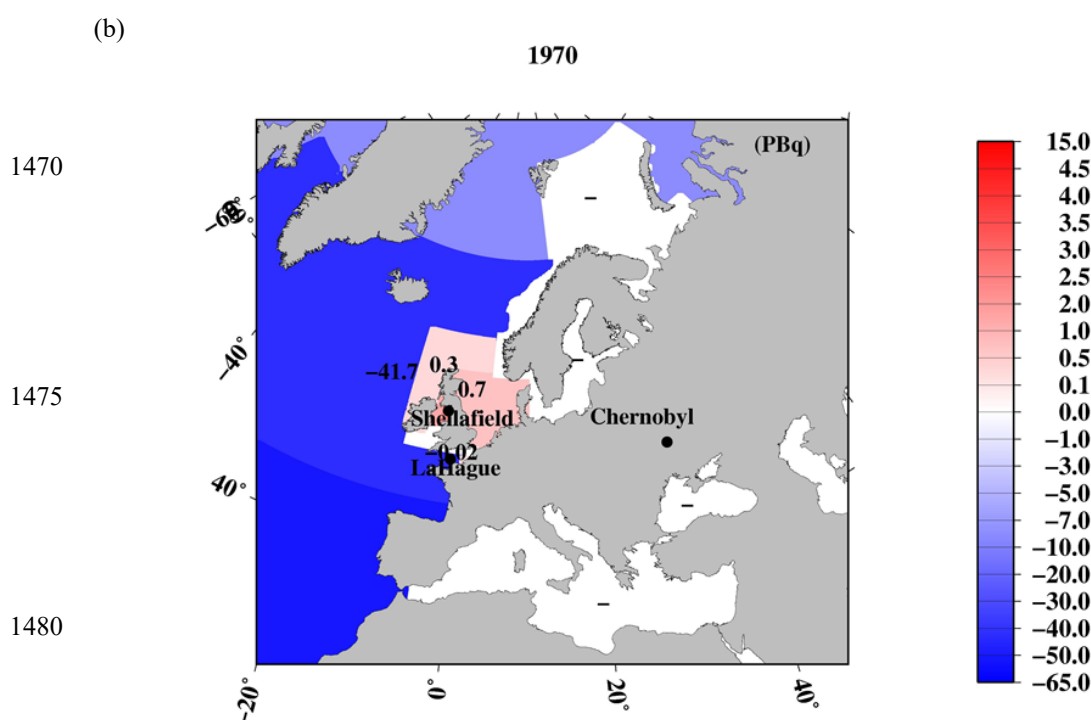



(c)

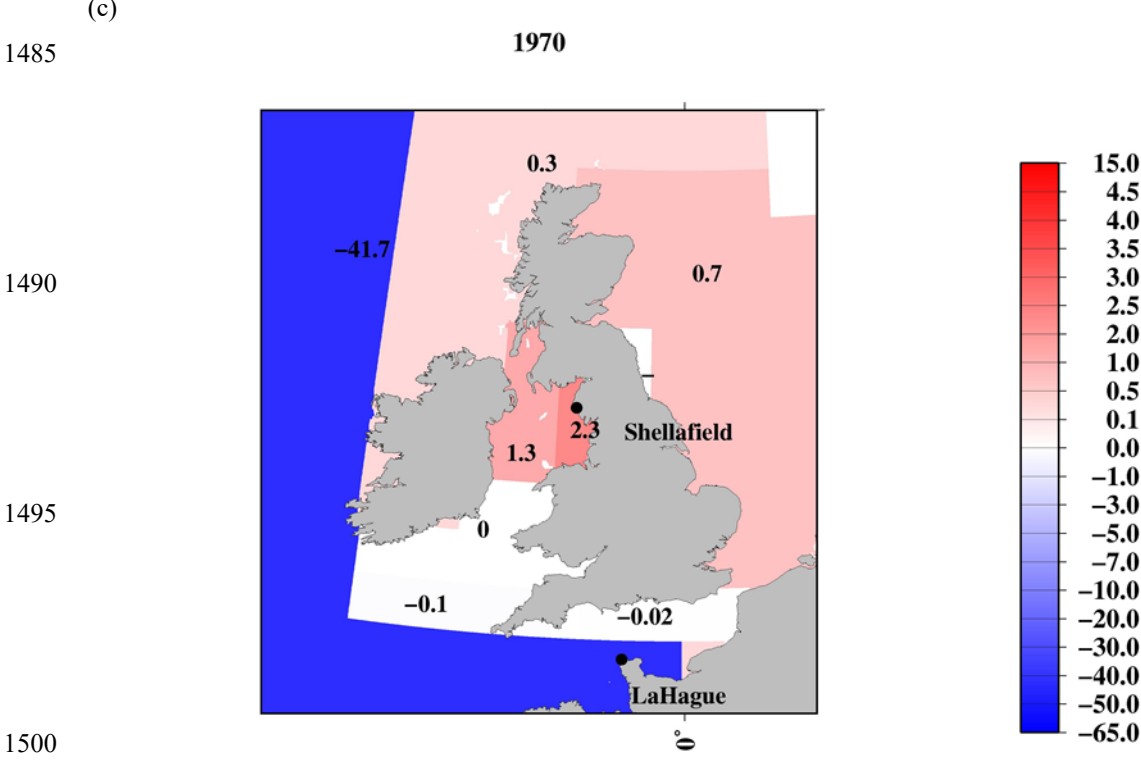

**Figure 18: Horizontal distribution of the $^{137}$Cs outflow amount in each box against the deposition amount in 1970 based on the 0.5-year $^{137}$Cs activity concentration data. The amount of $^{137}$Cs outflow includes the downwards transport portion below the surface mixed layer and horizontal transport in the surface mixed layer to the downstream boxes. A positive value (red) indicates the inflow amount, and negative values (blue) indicate the outflow amount. (a) Global ocean, (b) northern North Pacific Ocean and its marginal seas. The unit is PBq.**





(a)

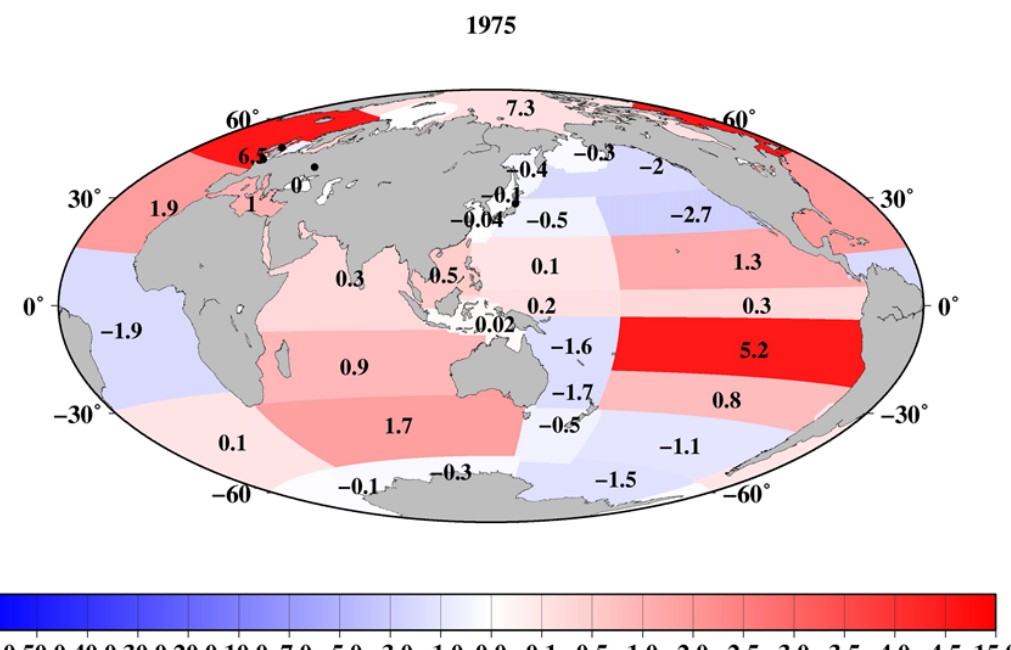

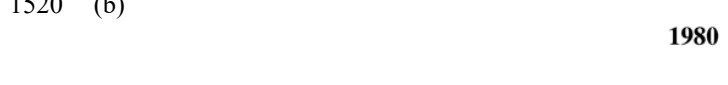

(b)

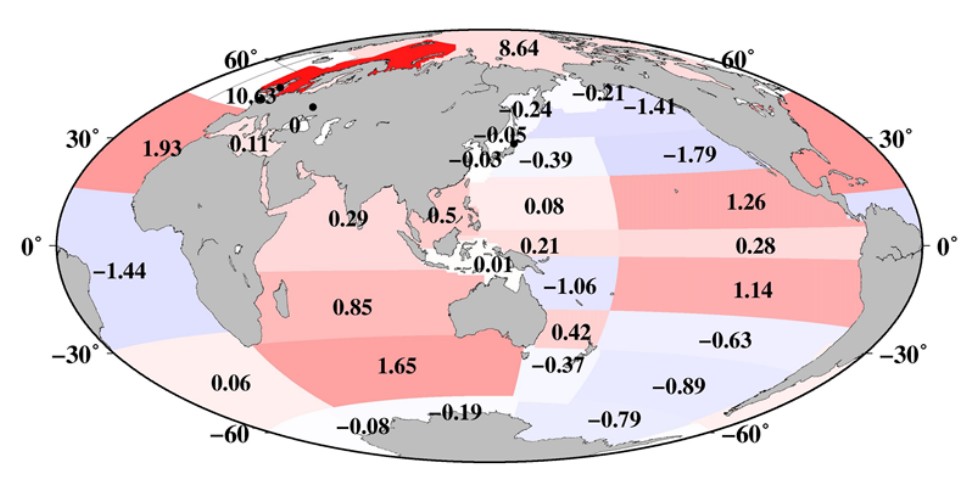



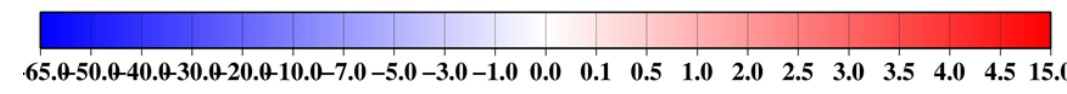


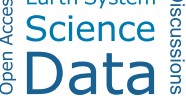

(c)




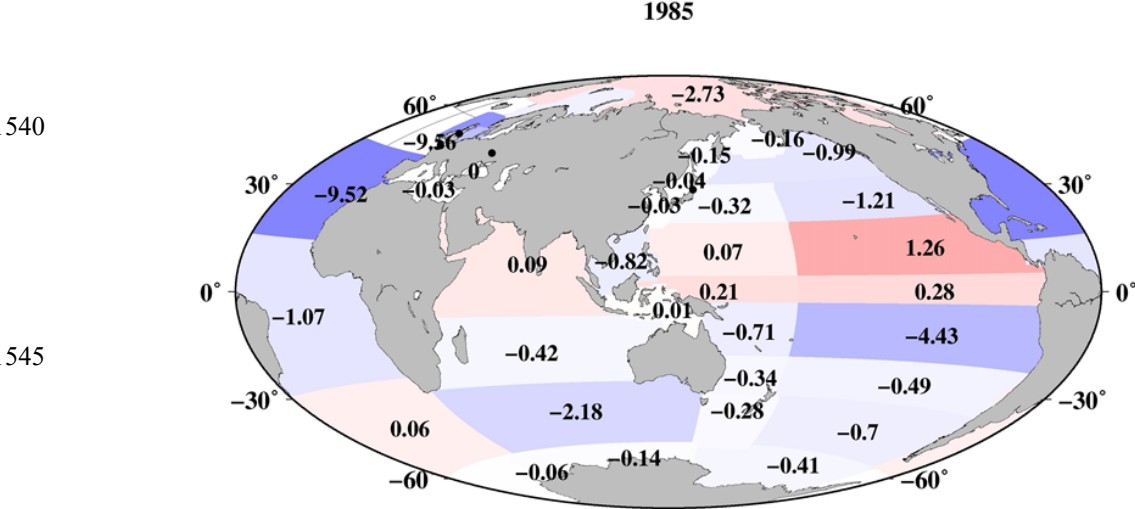

(d)




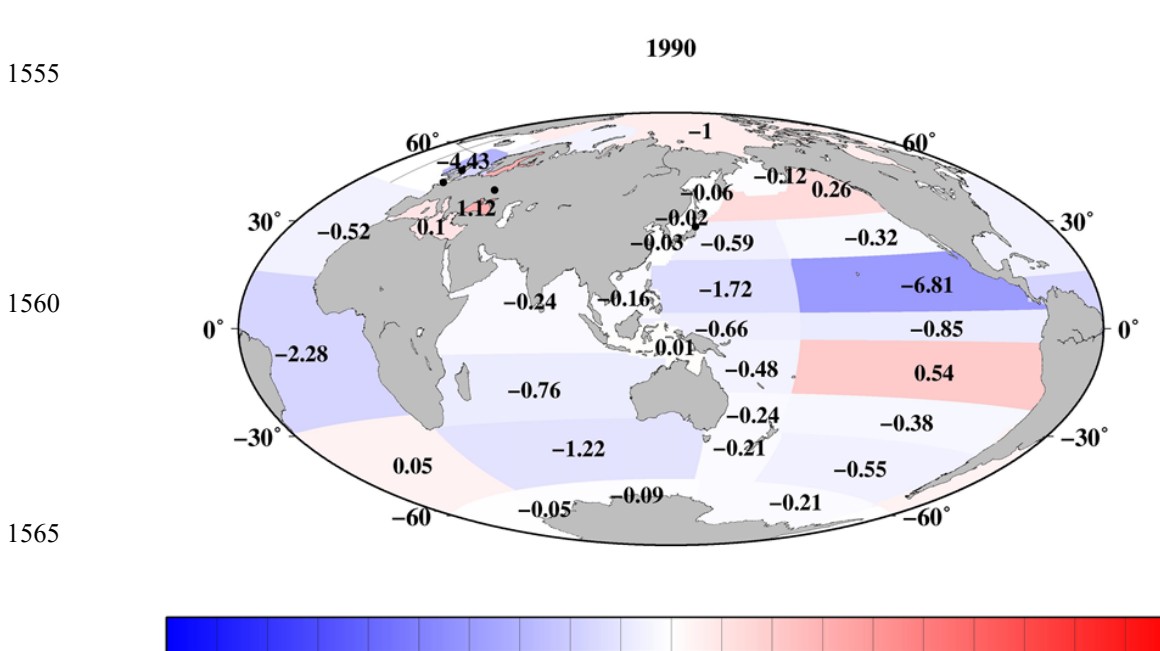





(e)

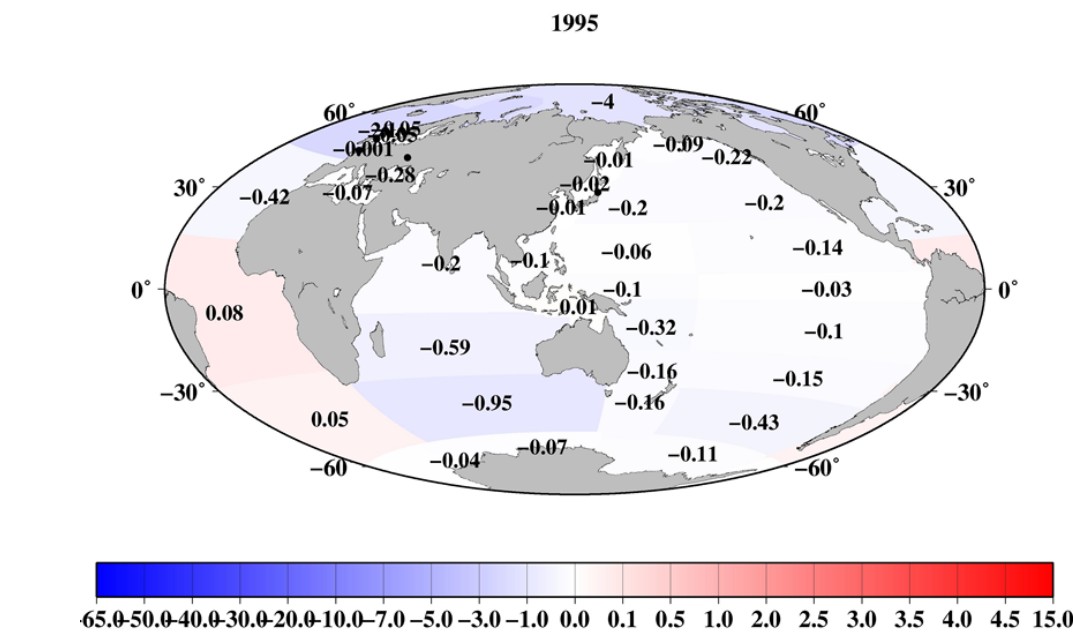

(f)

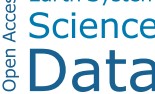

(g)

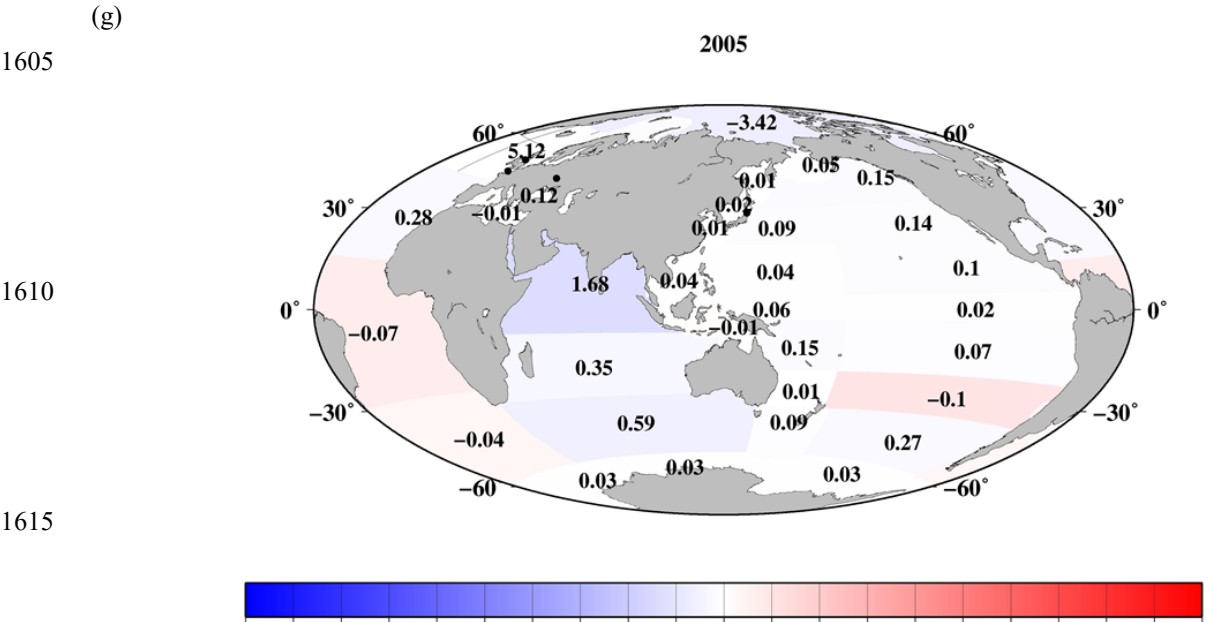

(h)

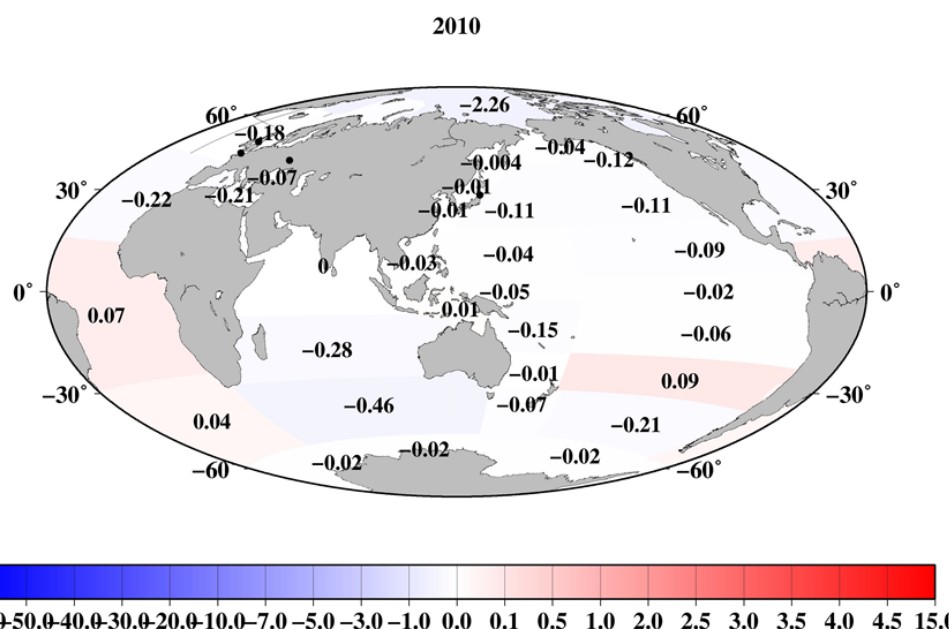





(i)

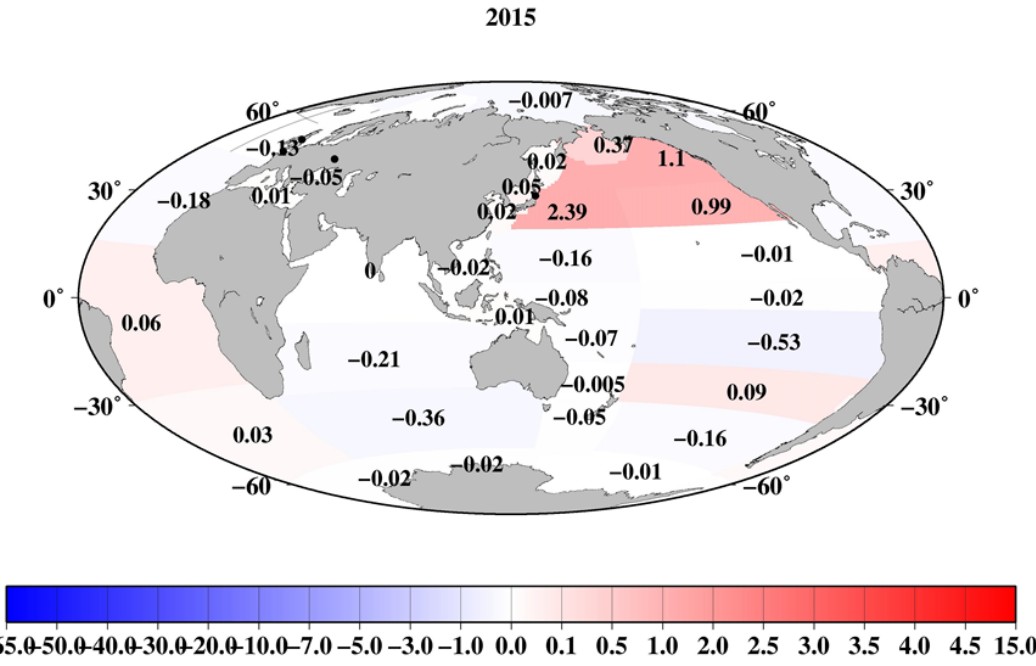

Figure 19: Mass balance of $^{137}$Cs in the surface seawater in each box in the global ocean. A positive value (red) indicates a larger inflow from the upstream boxes, and negative value (blue) indicates a larger outflow to the downstream boxes or below the surface mixed layer compared to the previous 5 years. The unit is PBq. (a) 1975, (b) 1980, (c) 1985, (d) 1990, (e) 1995, (f) 2000, (g) 2005, (h) 2010, and (i) 2015. The unit is PBq.



(a)

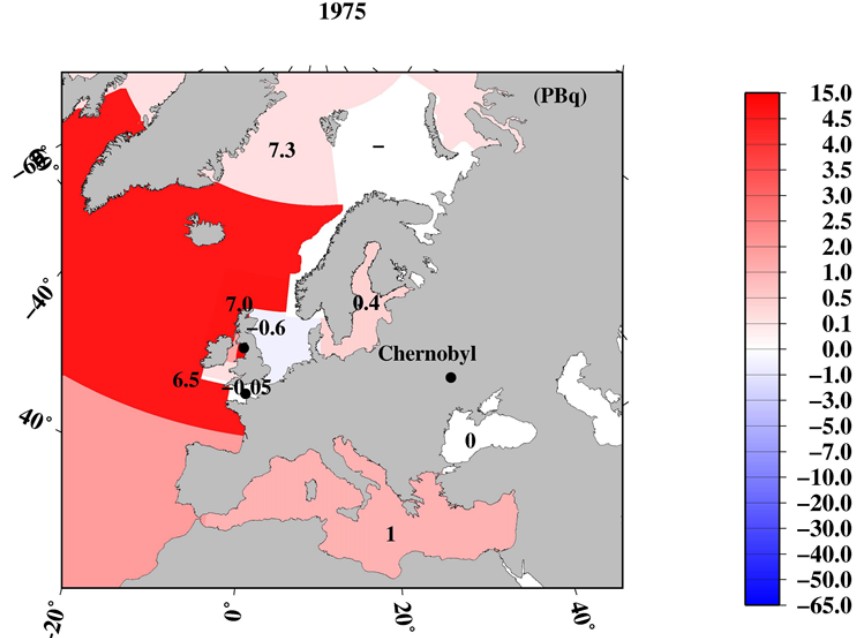

(b)

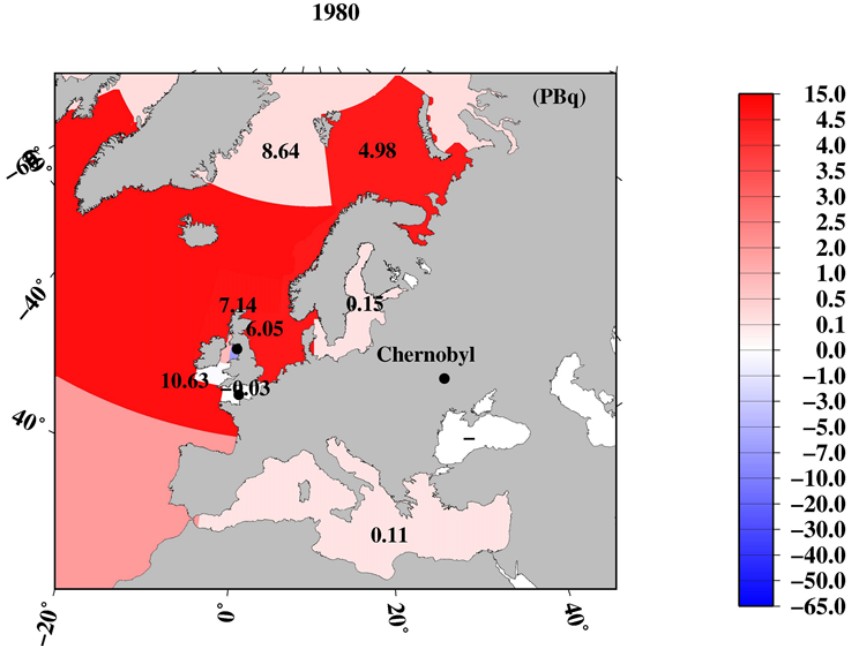



(c)

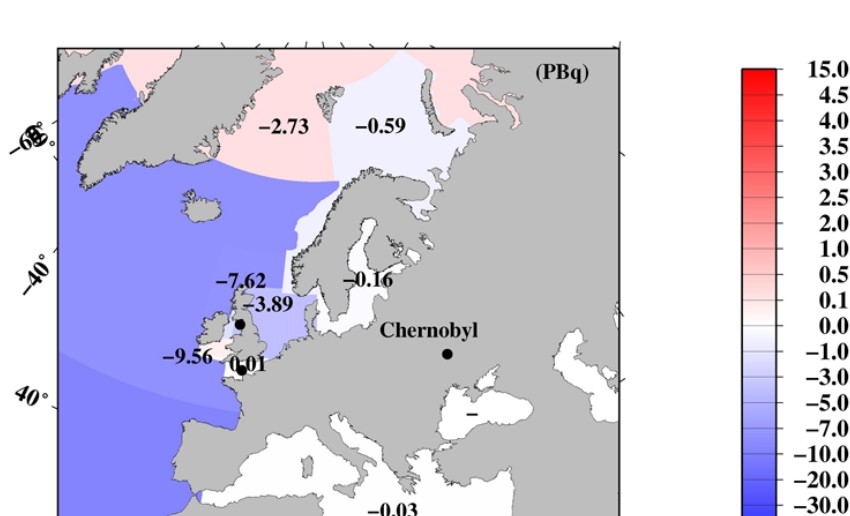

(d)

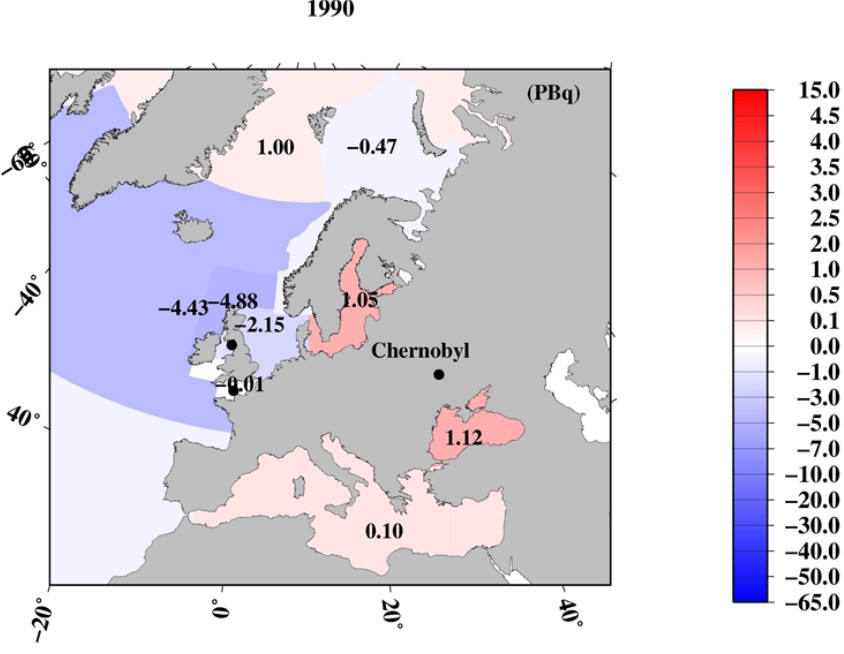



(e)

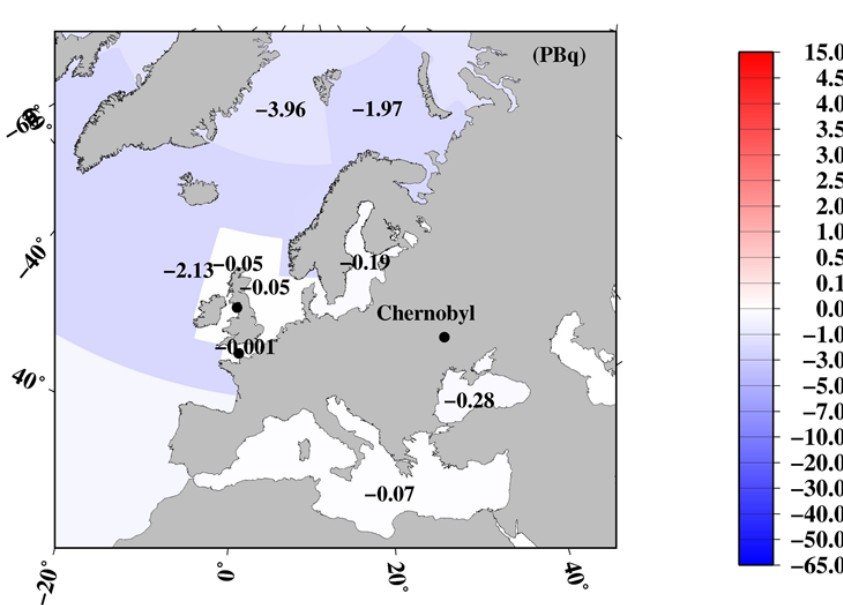


(f)



(g)

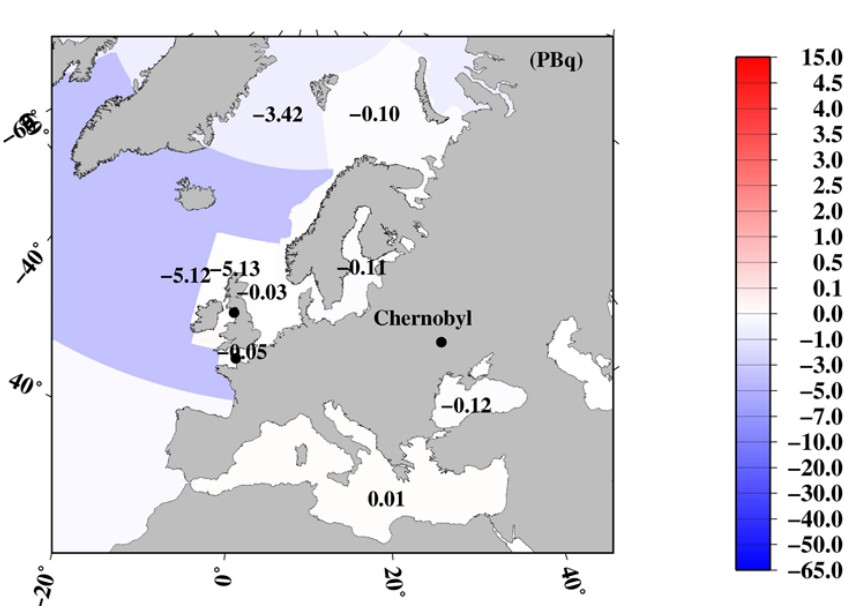

(h)

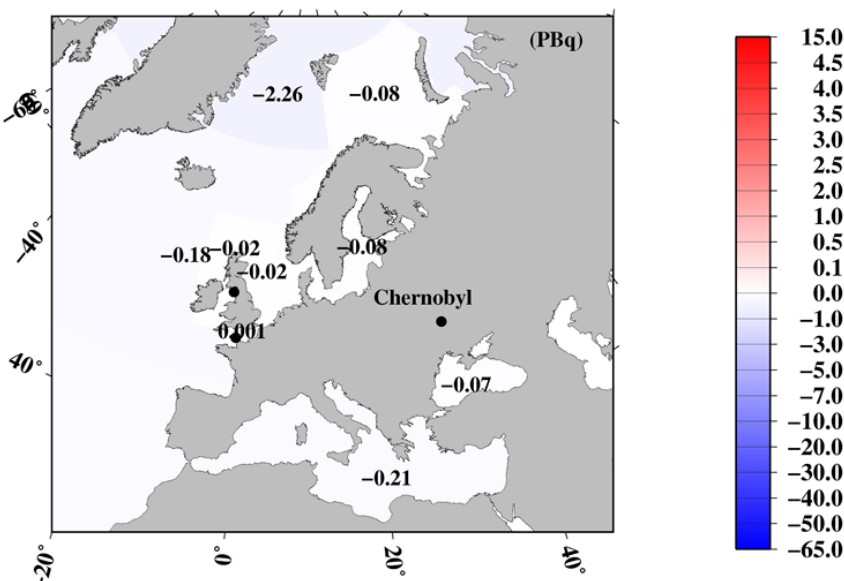



(i)

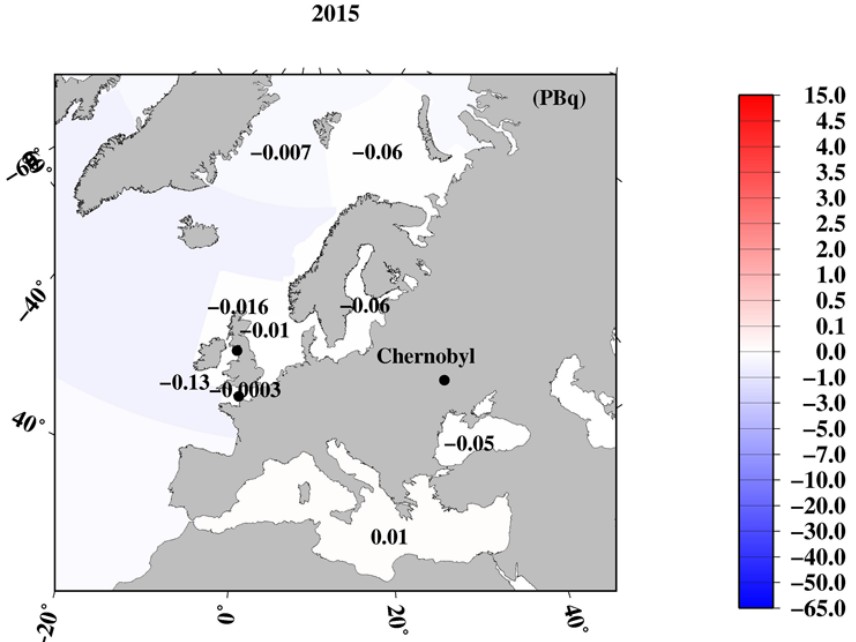

Figure 20: Mass balance of $^{137}$Cs in the surface seawater in each box in the northern North Atlantic Ocean and its marginal seas. A positive value (red) means the larger inflow from the upstream boxes, and a negative value (blue) indicates a larger outflow to the downstream boxes or below the surface mixed layer compared to the previous 5years. The unit is PBq. (a) 1975, (b) 1980, (c) 1985, (d) 1990, (e) 1995, (f) 2000, (g) 2005, (h) 2010, and (i) 2015.






(a)

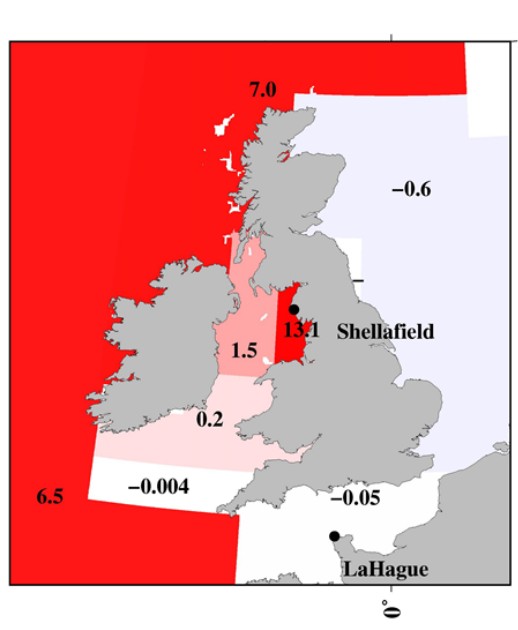
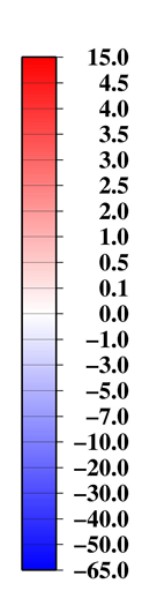

(b)

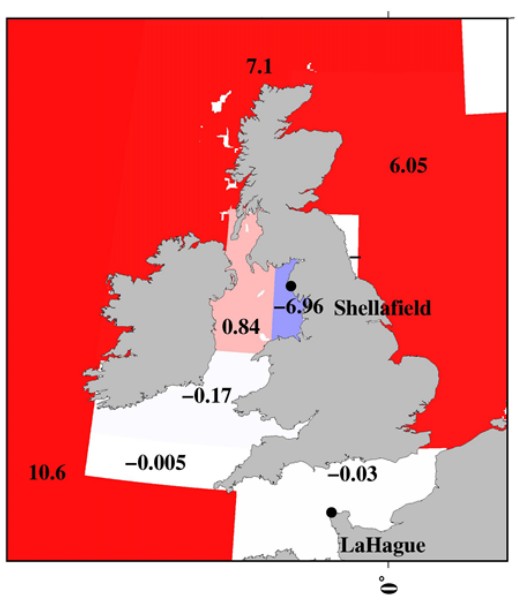
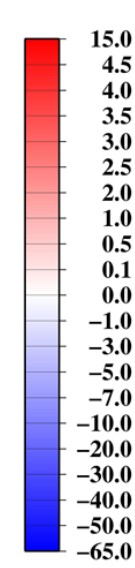



(c)

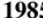

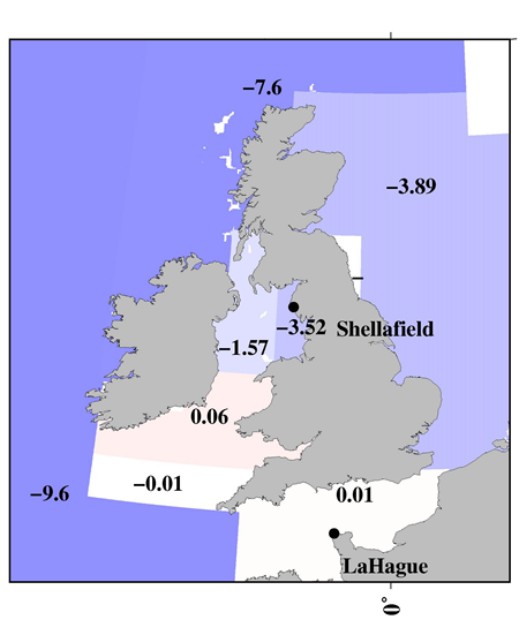
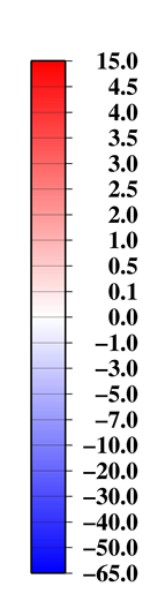

(d)

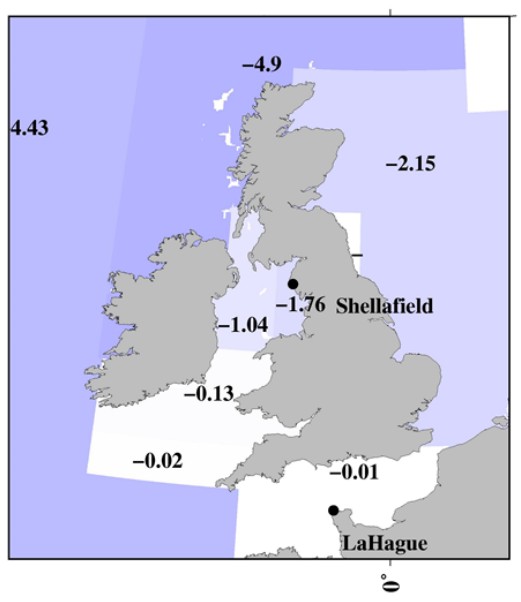
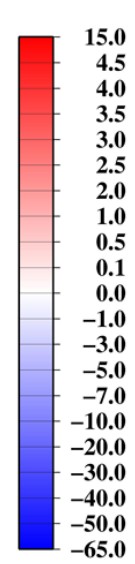



(e)

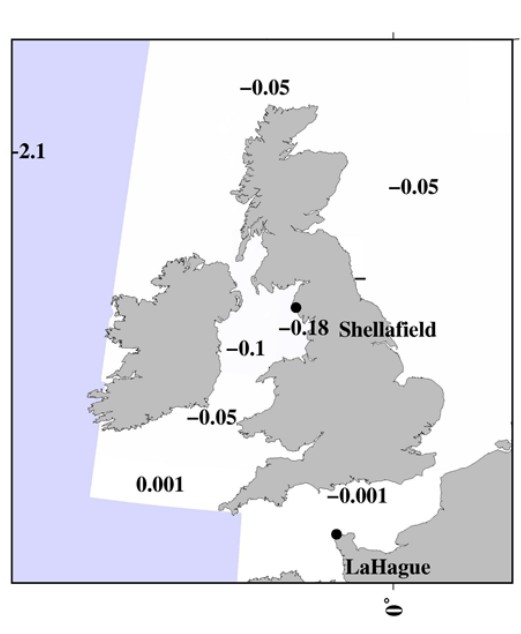
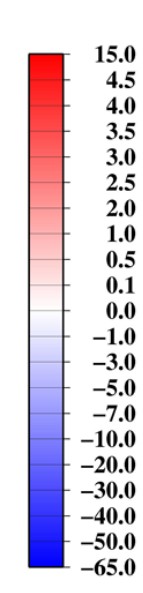


(f)

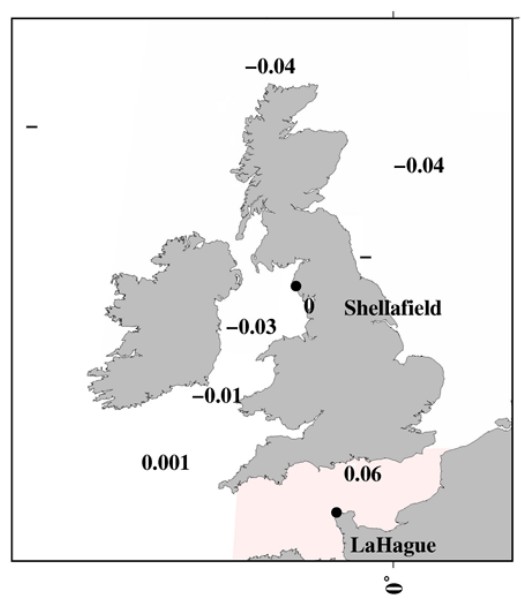
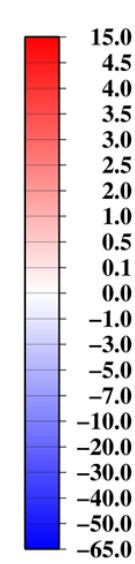



(g)

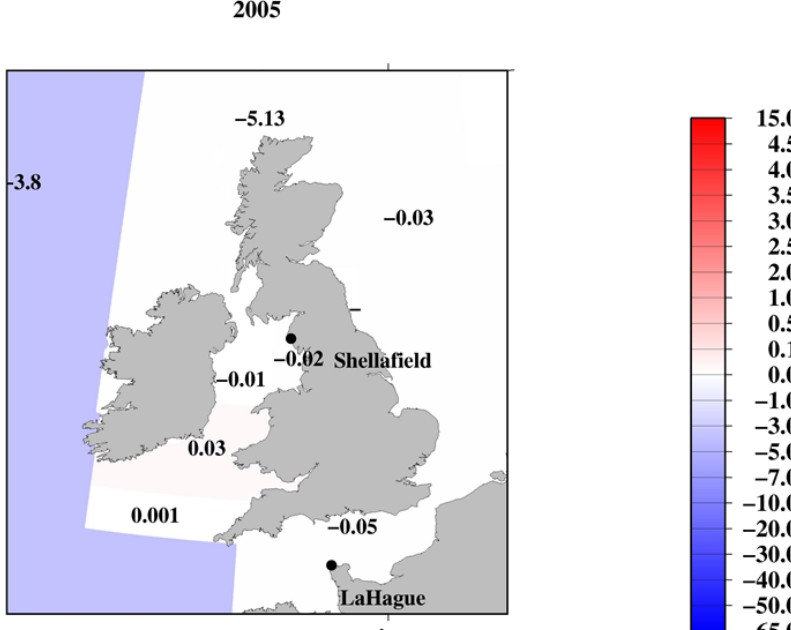

(h)

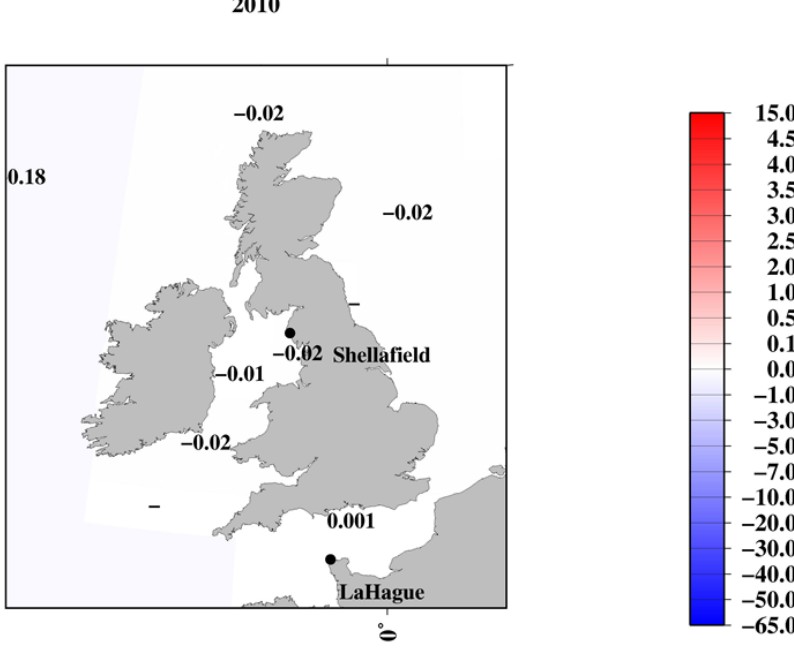





(i)

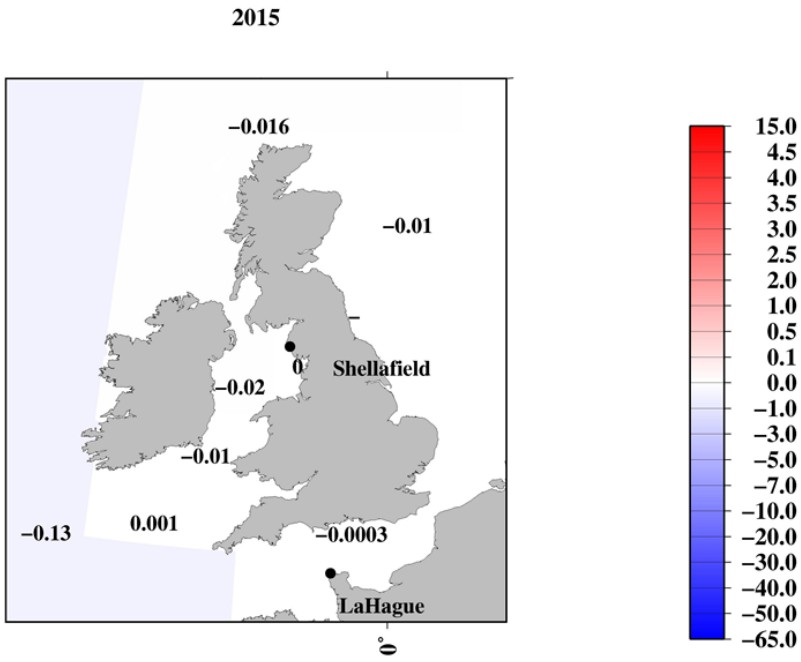

**Figure 21: Mass balance of** $^{137}$**Cs in the surface seawater in each box in the Irish Sea and English Chanel. A positive**
**value (red) indicates a larger inflow from the upstream boxes, and a negative value (blue) indicates a larger outflow to**
**the downstream boxes or below the surface mixed layer compared to the previous 5years. The unit is PBq. (a) 1975, (b)**
**1980, (c) 1985, (d) 1990, (e) 1995, (f) 2000, (g) 2005, (h) 2010, and (i) 2015.**








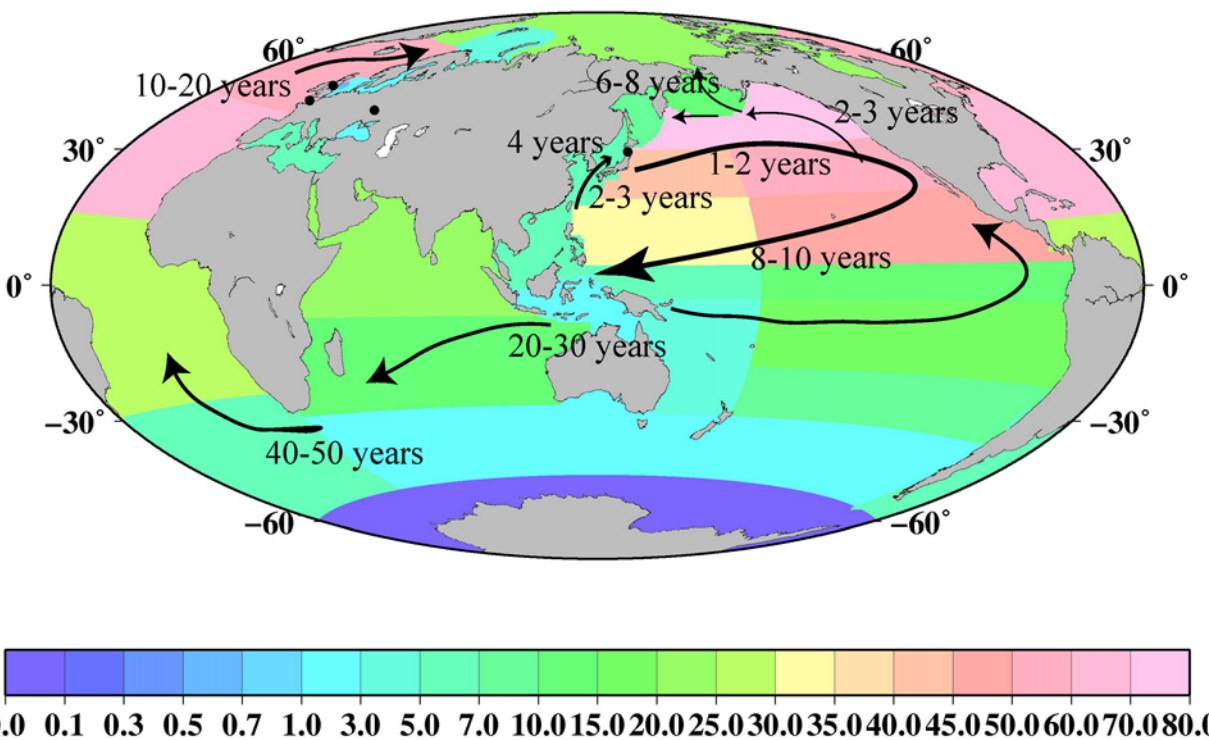

**Figure 22: Distribution of $^{137}$Cs deposition in 1 January 1970 and $^{137}$Cs transport route in the surface seawater of the global ocean deduced. Circles are location of the nuclear reprocessing plants, Chernobyl, and F1NPS.**

**5. Conclusions**

In this study, we analysed the $^{137}$Cs activity concentrations in the surface seawater in the global ocean by using almost all of the available historical data in the global ocean. The surface seawater was divided into 37 boxes, and the temporal variations in the 0.5-yr average $^{137}$Cs values in each box were investigated to determine the $^{137}$Cs distribution and transport in the global surface seawater.

The $^{137}$Cs deposition as of January 1970, with two × two minutes resolution, is estimated to be $874 \pm 90$ PBq. In 1970, due to the minor contribution of atmospheric deposition, the $^{137}$Cs inventory in the surface mixed layer in the global ocean was estimated to be $187\pm26$ PBq. This suggests that 68% of the $^{137}$Cs deposited into the surface seawater in the global ocean ($577 \pm 60$ PBq) had already been transported below the surface mixed layer on a decadal timescale. The $^{137}$Cs inventory increased

slightly and reached a maximum (210±12 PBq) in 1980. The increased $^{137}$Cs inventory was due to the discharged $^{137}$Cs from the reprocessing plants: Sellafield and La Hague. Then, the $^{137}$Cs inventory decreased, and the value in 2010, immediately before the F1NPS accident, was estimated to be 35 ± 3.6 PBq. The relative contributions in the South Pacific Ocean, Indian Ocean, and Atlantic Ocean to the $^{137}$Cs inventory in the surface mixed layer in the global ocean increased gradually. In 2011, the $^{137}$Cs inventory increased to 48.1 ± 12.1 PBq, in which the F1NPS-derived $^{137}$Cs accounted for 15.5 ± 3.9 PBq.

The spatiotemporal variation in the $^{137}$Cs inventory, density, and mass balance analysis suggests that $^{137}$Cs deposited by the large-scale nuclear weapons tests in the western North Pacific Ocean was transported eastwards and accumulated in the eastern subtropical North Pacific Ocean over 7-8 years. After arriving at $^{137}$Cs near the coast of California, this $^{137}$Cs was transported southwards with subsidence in the equatorial Pacific Ocean and transported westwards in the equatorial Pacific Ocean for 8-10 years. This $^{137}$Cs entered the Indian Ocean from the Pacific Ocean over the 2-3 decades. Then, $^{137}$Cs was transported into the South and Central Atlantic Ocean over a period of 4-5 decades. The F1NPS-derived $^{137}$Cs arrived near the coast of Canada and was transported with the Kuroshio Current, and its extension was northwards transport with the Alaska Current over 6-8 years. An increase in $^{137}$Cs activity concentrations is observed in the Bering Sea and Okhotsk Sea. Some of the $^{137}$Cs was transported into the Arctic Ocean. In the northern North Atlantic Ocean and its marginal seas, a significant amount of $^{137}$Cs was discharged from reprocessing plants transported to the North Sea, Barents Sea and coast of Norway, and the Arctic Ocean over approximately 1-2 decades. Contamination of the $^{137}$Cs released from the Chernobyl accident has caused higher activity concentrations in the Baltic Ocean until 2015.

Finally, because $^{137}$Cs is water soluble, its transport and distribution strongly depend on seawater circulation. The transport of $^{137}$Cs-labelled seawater can be examined to interpret the circulation of substances in seawater, as well as the climate change associated with gaseous exchange between the atmosphere and the ocean surface.

**Author contribution**

YI (corresponding author) conducted data analysis and the preparation of the manuscript. MA developed the database of radioactivity. All authors discuss about the results of the data analysis.





**Data availability**

Data described in this manuscript can be accessed at repository under data doi.

Aoyama (2021) HAM, Histroical Artifial radioacitity database in Marine environmnet, Global2021. Center for Research in Isotopes and Environmental Dynamics, University of Tsukuba, http://doi: 10.34355/CRiED.U.Tsukuba.00085.

Inomata and Aoyama (2022a) [137]Cs measurement points in the surface seawater in the global ocean based in the HAM database2021. Center for Research in Isotopes and Environmental Dynamics, University of Tsukuba, 2021. http://doi: 10.34355/ Ki-net.KANAZAWA-U.00149.

Inomata and Aoyama (2022b) Temporal variations of [137]Cs activity concentrations and these 0.5-yr average values in the surface seawater in the global ocean. Center for Research in Isotopes and Environmental Dynamics, University of Tsukuba, 2021. http://doi: 10.34355/ Ki-net.KANAZAWA-U.00150.

Inomata and Aoyama (2022c) Dataset of 0.5-yr average values of [137]Cs activity concentrations in the surface seawater in the global ocean during the period from 1957 to 2021. Center for Research in Isotopes and Environmental Dynamics, University of Tsukuba, 2021. http://doi: 10.34355/ Ki-net.KANAZAWA-U.0015.

**Competing interests**

The authors declare that they have no conflict of interest.

**Disclaimer**

Publisher's note: Copernicus Publications remains neutral with regard to jurisdictional claims in published maps and institutional affiliations.

**Acknowledgements1**

This research was financially supported by the Grant-in-Aid for Scientific Research on Innovative Areas, "Interdisciplinary study on environmental transfer of radionuclides from the Fukushima Dai-ichi NPP Accident" (Project No. 25110511) of the Japanese Ministry of Education, Culture, Sports, Science, and Technology (MEXT). This research was also supported by the cooperation program of the Environmental Radioactivity Research Network Centre (F-19-02, F-20-08, F-21-18, F-22-04) and the Institute of Nature and Environmental Technology, Kanazawa University (18009, 19022, 20043).





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

Figure 1: Boxes dividing the global ocean. (a) Global, (b) North Atlantic Ocean and its marginal Sea, and
(c) Irish Sea (Box 23) and English Chanel (Box 22).

Figure 2: 0.5-yr average $^{137}$Cs values of each box for (a) Boxes 1–3 (subarctic North Pacific Ocean, western North Pacific
Ocean, and eastern North Pacific Ocean), (b) Boxes 14 and 31–34 (Japan Sea, Sea of Okhotsk, Eastern China Sea, South China
Sea, and Bering Sea), (c) Boxes 4-9 (western subtropical North Pacific Ocean, eastern subtropical North Pacific Ocean, western
equatorial Pacific Ocean, eastern equatorial Pacific Ocean, eastern subtropical South Pacific Ocean, and western subtropical
South Pacific Ocean), (d) Boxes 10–12 and 19 (western subtropical South Pacific Ocean, eastern subtropical South Pacific
Ocean, eastern Southern Ocean, and middle Southern Ocean), (e) Boxes 13, 36 and 17 (Pacific sector of the Antarctic Ocean,
Atlantic sector of the Antarctic Ocean, and Indian sector of the Antarctic Ocean), (f) Boxes 15–17 (Arabian Sea, Indian Ocean,
and Southern Ocean), (g) Boxes 28–30 (North Atlantic Ocean, Central Atlantic Ocean, and South Atlantic Ocean), (h) Boxes
18, 20, 22 and 25 (Arctic Ocean, Barents Sea and coast of Norway, North Sea, and northern North Atlantic Ocean), (i) Boxes
23 and 24 (Irish Sea and English Channel), (j) Boxes 21, 26 and 27 (Baltic Sea, Black Sea and Mediterranean Sea).

Figure 3: Comparison with 0.5-yr average $^{137}$Cs values in the Pacific Ocean, Indian Ocean, and Atlantic Ocean.



Figure 4: Temporal variation in the 0.5-yr average $^{137}$Cs values. (a) Western North Pacific Ocean, (b) Japan Sea. The lines represent the exponential decay of the 0.5-yr $^{137}$Cs average values before 1970 (Tap1), 1970-1985 (Tap2), and 1990-2010 (Tap3).


Figure 5: Horizontal distributions of $^{137}$Cs deposition density (KBqm-2) as of the 1$^{st}$ of January 1970. (a) Global Ocean, (b) Northern North Pacific Ocean and its marginal seas. Black circles are locations of the F1NPS, Sellafield, La Hague, and Chernobyl power plants.

Figure 6: Horizontal distributions of the $^{137}$Cs deposition amount (PBq) in each box as of the 1$^{st}$ of January 1970. (a) Global Ocean, (b) Northern North Pacific Ocean and its marginal seas. Black circles are locations of the F1NPS, Sellafield, La Hague, and Chernobyl power plants.

Figure 7: Horizontal distributions of the 0.5-yr average $^{137}$Cs value in the surface mixed layer in the global ocean. The unit is
Bqm$^{-3}$. (a) 1970, (b) 1975, (c) 1980, (d) 1985, (e) 1990, (f) 1995, (g) 2000, (h) 2005, (i) 2010, and (j) 2015.

Figure 8: Horizontal distributions of the 0.5-yr average $^{137}$Cs value in the surface mixed layer in the northern North Atlantic Ocean and its marginal seas. The unit is Bqm$^{-3}$. (a) 1970, (b) 1975, (c) 1980, (d) 1985, (e) 1990, (f) 1995, (g) 2000, (h) 2005, (i) 2010, and (j) 2015.


Figure 9: Horizontal distributions of the 0.5-yr average $^{137}$Cs value in the surface mixed layer in the Irish Sea. The unit is Bqm$^{-3}$. (a) 1970, (b) 1975, (c) 1980, (d) 1985, (e) 1990, (f) 1995, (g) 2000, (h) 2005, (i) 2010, and (j) 2015. The "-" means that there are no available data.

Figure 10: Average mixed layer depth in each box in the global ocean. (a) Global, (b) North Atlantic Ocean and its marginal sea, and (c) Irish Sea. The unit is m. The "-" means that there are no mixed layer depth data.

Figure 11: Horizontal distributions of the $^{137}$Cs inventory in the surface mixed layer in the global ocean.


Figure 12: Horizontal distributions of the $^{137}$Cs inventory in the surface mixed layer in the northern North Pacific Ocean and its marginal seas.

Figure 13: Horizontal distributions of the $^{137}$Cs inventory in the surface mixed layer in the Irish Sea and English Channel.




Figure 14: Temporal variations in the [137]Cs inventory every 5 years in global ocean surface seawater.

Figure 15: [137]Cs density in the surface mixed layer in the North Pacific Ocean, equatorial Pacific Ocean, and South Pacific Ocean.


Figure 16: [137]Cs density in the Indonesian Archipelago and surrounding sea in the surface mixed layer.

Figure 17: Temporal variations in the 0.5-yr average values of surface [137]Cs in the North/equatorial Pacific Ocean and its marginal seas after 2011. (a) Western and eastern North Pacific Ocean, (b) Japan Sea and Eastern China Sea.


Figure 18: Horizontal distribution of the [137]Cs outflow amount in each box against the deposition amount in 1970 based on the 0.5-year [137]Cs activity concentration data. The amount of [137]Cs outflow includes the downwards transport portion below the surface mixed layer and horizontal transport in the surface mixed layer to the downstream boxes. A positive value (red) indicates the inflow amount, and negative values (blue) indicate the outflow amount. (a) Global ocean, (b) northern North

Pacific Ocean and its marginal seas. The unit is PBq.

Figure 19: Mass balance of [137]Cs in the surface seawater in each box in the global ocean. A positive value (red) indicates a larger inflow from the upstream boxes, and a negative value (blue) indicates a larger outflow to the downstream boxes or below the surface mixed layer compared to the previous 5 years. The unit is PBq. (a) 1975, (b) 1980, (c) 1985, (d) 1990, (e)

1995, (f) 2000, (g) 2005, (h) 2010, and (i) 2015. The unit is PBq.

Figure 20: Mass balance of [137]Cs in the surface seawater in each box in the northern North Atlantic Ocean and its marginal seas. A positive value (red) indicates a larger inflow from the upstream boxes, and a negative value (blue) indicates a larger outflow to the downstream boxes or below the surface mixed layer compared to the previous 5 years. The unit is PBq. (a) 1975,

(b) 1980, (c) 1985, (d) 1990, (e) 1995, (f) 2000, (g) 2005, (h) 2010, and (i) 2015.

Figure 21: Mass balance of [137]Cs in the surface seawater in each box in the Irish Sea and English Chanel. A positive value (red) indicates a larger inflow from the upstream boxes, and a negative value (blue) indicates a larger outflow to the downstream boxes or below the surface mixed layer compared to the previous 5 years. The unit is PBq. (a) 1975, (b) 1980, (c)

1985, (d) 1990, (e) 1995, (f) 2000, (g) 2005, (h) 2010, and (i) 2015.