# Peer review of "Evaluating the transport of surface seawater from 1956 to 2021 using 137Cs deposited in the global ocean as a chemical tracer"

_Earth System Science Data, 2022_

## Author Response (AR1)

Dear Reviewer (Citation: https://doi.org/10.5194/essd-2022-374-RC1)

Thank you very much for your valuable comments. Taking into account your comments, I modified the manuscript as follows. The modified part was shown in red text.

● Please, correct the UNSCEAR quote which is incorrect in almost all of the text.

A. I modified the term of "UNSCEAR".

● line 52 the dissolved $^{137}$Cs
● line 510 Papucci
● line 1850 Papucci, C., Salvi, S., Lorenzelli R.
● line 1857 Papucci

A. line 1427-1432

Delfanti, R., Papucci, C., Salvi, S., Lorenzelli R.: IAEA CRP "Worldwide Marine Radioactivity". Research Contract - Agreement No. ITA-26803, ENEA, Italy, 2000.

Delfanti, R., and Pappuci, C.: Mediterranean Sea, in: Atwood, D. (Ed.), Encyclopedia of Inorganic Chemistry. Wiley, pp. 401–414, 2010.

Delfanti, R., Özsoy, E., Kaberi, H., Schirone, A., Salvi, S., Conte, F., Tsabaris, C., and Papucci, C.: Evolution and fluxes of $^{137}$Cs in the Black Sea/Turkish Straits System/North Aegean Sea. J. Mar. Systems. 135, 117–123, 2014.

● please, define tap in a more explicit way: apparent half-life

L266-281, The regression line of the 0.5-yr median value of $^{137}$Cs for each box was determined and apparent half residence time (Tap) due to the radioactive decay and ocean physical processes were estimated.
The Tap of $^{137}$Cs was calculated using the following equations:

$$^{137}\text{Cs} = {}^{137}\text{Cs}_0 \exp(-\lambda_{cs,\ apparent}) \qquad (1)$$

$$\lambda_{Cs,apparent} = \lambda_{Cs,ocean} + \lambda_{Cs,decay} \qquad (2)$$

$$\text{Tap} = 0.693/(\lambda_{Cs,apparent}) \qquad (3)$$

$$\text{Tpo} = 0.693/(\lambda_{Cs,ocean}) \qquad (4)$$

where $\lambda_{Cs,apparent}$, $\lambda_{Cs,ocean}$, and $\lambda_{Cs,decay}$ are the decay constants for apparent decay, physical oceanographic decay, and radioactive decay, respectively. $\lambda_{Cs,apparent}$ is estimated by using the regression line of the 0.5-yr median value of $^{137}$Cs as shown in (1). Tpo is the apparent half residence time by causing the oceanic physical processes and $\lambda_{Cs,ocean}$ was estimated $\lambda_{Cs,apparent}$ and $\lambda_{Cs,decay}$ in equation (2). Considering that the half-life of $^{137}$Cs ($T_{1/2}$) is 30.17 years, the Tap should be shorter than the half-life if no source of $^{137}$Cs exists in the region of interest. A shorter Tap means that $^{137}$Cs is removed quickly in the area and/or the $^{137}$Cs inflow amount is small in the area compared with the $^{137}$Cs outflow amount. In other words, a Tap shorter than the radioactive decay time indicates that the variations in the $^{137}$Cs activity concentrations are strongly controlled by physical ocean processes. In contrast, a longer Tap as well as a negative Tpo value means that $^{137}$Cs is preserved in the region for a longer time and/or there is an influx of water mass with higher $^{137}$Cs in the region compared to the $^{137}$Cs outflow from the region.

- please, define the start of half year because layering could affect the mixed layer

In Table 3, the start and end year for estimate Tap were described.

| | | Start | End | Tap | Tpo |
|---|---|---|---|---|---|
| Tap | western North Pacific Ocean | 1970 | 2010 | 15.0 | 29.9 |
| | Eastern China Sea | 1970 | 2010 | 17.7 | 42.8 |
| | Japan Sea | 1970 | 2010 | 16.4 | 35.9 |
| Tap1 | subarctic North Pacific Ocean | 1957.5 | 1969.5 | 8.6 | 12.1 |
| | western subtropical North Pacific Ocean | 1957 | 1970 | 4.3 | 5.0 |
| | western equatorial Pacific Ocean | 1960.5 | 1966.5 | 52.0 | -71.8 |
| | eastern equatorial Pacific Ocean | 1963.5 | 1969.5 | 5.8 | 7.2 |
| | eastern subtropical South Pacific Ocean | 1966 | 1970 | 6.0 | 7.5 |
| Tap2 | subarctic North Pacific Ocean | 1970.5 | 1984.5 | 9.6 | 14.0 |
| | eastern North Pacific Ocean | 1970.5 | 1985 | 8.8 | 12.4 |
| | western subtropical North Pacific Ocean | 1970 | 1989 | 34.1 | -260.7 |
| | Indonesian Archipelago | 1973 | 1997 | 36.7 | -169.2 |
| Tap3 | subarctic North Pacific Ocean | 1990.5 | 2009.5 | 18.2 | 45.8 |
| | Sea of Okhotsk | 1992 | 2010 | 24.0 | 117.0 |
| | western subtropical North Pacific Ocean | 1990 | 2011 | 25.2 | 153.0 |
| | western equatorial Pacific Ocean | 1992 | 2003 | 15.6 | 32.5 |
| | Indonesian Archipelago | 1973 | 1997 | 36.7 | -169.2 |
| | Baltic Sea | 1990 | 2017 | 11.5 | 18.6 |
| | North Atlantic Ocean | 1992 | 2017 | 21.3 | 72.3 |
| | Central Atlantic Ocean | 1992 | 2016 | 38.0 | -146.5 |
| | South Atlantic Ocean | 1994 | 2013.5 | 15.4 | 31.4 |

- Correct the follow figure:

- fig 1 (a-b) unreadable labels and coordinates

[Figure]

[Figure]

- fig 2 label, Mediterranean

A. The spell of Mediterranean Sea is modified as follows.

[Figure]

- fig 5 b unreadable coordinates

Fig 5b was modified as follows.

[Figure]

- fig 6 b unreadable coordinates

[Figure]

- fig 7 (a-j) unreadable label

[Figure]

- fig 8 (a-j) unreadable coordinates

[Figure]

- fig 10 (a b) unreadable coordinates

[Figure]

[Figure]

- fig 12 (a-j) unreadable labels and coordinates

[Figure]

- fig 18 (b) unreadable coordinates

[Figure]

- fig 19 (a-i) unreadable label

[Figure]

- fig 20 (a-i) unreadable coordinates

1975

(PBq)

| | |
|---|---|
| | 15.0 |
| | 4.5 |
| | 4.0 |
| | 3.5 |
| | 3.0 |
| | 2.5 |
| | 2.0 |
| | 1.0 |
| | 0.5 |
| | 0.1 |
| | 0.0 |
| | −1.0 |
| | −3.0 |
| | −5.0 |
| | −7.0 |
| | −20.0 |
| | −65.0 |

- line 480

  the value that is almost equal to the value before Chernobyl accident demonstrates the importance of increased release from the rivers flowing through the area affected by the accident: in fact, in Mediterranean Sea, where this input is negligible, the value in 2002 were less than half pre-Chernobyl accident.

Thank you very much for your comments. The importance of $^{137}$Cs inflow into the Black Sea from the rivers were added in the manuscript.

A. Line 499-505

The 0.5-yr median value of $^{137}$Cs in the Black Sea (Box 26) in 1977 and 1978.5 was approximately 17 Bq m$^{-3}$, and in 1986, it increased to 299 Bq m$^{-3}$, which was at least 18 times higher than that before the Chernobyl accident (Fig. 2j). The 0.5-yr median value of $^{137}$Cs decreased rapidly to 60 Bq m$^{-3}$ in 1989. The 0.5-yr median $^{137}$Cs value in 2002 was almost equal (18.3 Bq m$^{-3}$) to that before the Chernobyl accident. The rapid decrease in surface $^{137}$Cs could be due to the strong intrusion of surface waters to the deep layers, $^{137}$Cs inflow into the Mediterranean Sea after passing through the Bosporus Strait, and radioactive decay (Egorov, 1999; Delfanti et al., 2014). However, Black Sea continues to receive $^{137}$Cs derived from Chernobyl by the runoff from rivers (Gulin et al., 2013).

**https://doi.org/10.5194/essd-2022-374-RC3**

**General comments**

The theme of this work is valuable in that its result is expected to be helpful to understand the long-term spatiotemporal variation of Cs137 in the surface water of global ocean associated with the global fallout, Chernobyl and Fukushima NPP accidents, and discharges from reprocessing facilities. The manuscript is reasonably well written though some improvements are obviously needed. Furthermore, some questions about mass balance and inflow/outflow Cs137 amount need to be explained in detail which are, reviewer think, the most essential information in this study.

The manuscript in present form is not sufficient enough to recommend the publication in ESSD. Major revision is recommended to provide the opportunity of strengthening the manuscript.

**Answer to the reviewer comments**

Thank you very much for your valuable comments. I modified the manuscript taking into account your comments. The modified part was shown in the blue text.

**Detailed specific comments**

**Abstract:**

- Line 10, Pg.1:

Reviewer understands that authors used mainly HAMGlobal2021 data and partly MARIS data, but authors mentioned only HAMGlobal2021 data It is better to describe this matter correctly.

A. MARIS data were also included in the HAMGLobal2021. After published the manuscript in ESSD, researcher in the IRSN send me their data set (about 2000 records). I asked to Editor to use these additional data. These new data were also used in this analysis.

Line 145-147, Page 5

In HAMGlobal2021, the dataset produced by the IAEA Marine Radioactivity Information System (MARIS) were combined. Furthermore, we used the $^{137}$Cs data reported in the IRSN database (Baily du Bois, P. et al., 2020). Finally, all these data were compiled into a single comprehensive database for this study.

Line 97-103, Page 4

The data used in this study were adopted from the HAMGlobal2021: Historical Artificial radioactivity database in the Marine environment, Global integrated version 2021 (Aoyama, 2021), which contains data from the F1NPS accident. The HAMGlobal2021 database contains information on several radionuclides ($^{134}$Cs, $^{137}$Cs, $^{90}$Sr, $^3$H, $^{239,240}$Pu, $^{241}$Am, and $^{14}$C) in the global ocean. The data were measured from 1956 to 2021. The dataset in International Atomic Energy Agency Marine Radioactivity Information System (IAEA MARIS) database were also compiled in the HAMGlobal2021. In addition to this, the data measured in the North Atlantic Ocean and its marginal seas developed by IRSN (Bois et al., 2020) were also contained in this study.

- Lines 11-17, Pg.1:

Author used data from 1956-2021 data but no statement before 1970 is found in the abstract. It is desirable to include comments before 1970.

A. Lines 12-13, Pg.1:

The 0.5-yr median value of $^{137}$Cs in each box in the Pacific Ocean, the values were gradually increased or almost constant levels in the 1950s and1960s, and then, except in the northern North Atlantic Ocean and its marginal sea, decreased exponentially in 1970–2010, immediately before the Fukushima Nuclear Power Plant (F1NPS) accident.

**Introduction:**

- Line 30, Lines 35-36, Line 46, Pg.2:

Authors repeatedly elaborated that the Cs137 originates from large-scale atmospheric weapons tests. Writing improvements are needed.

A. This part was modified as follows.

Lines 32-44, Pg.2:

$^{137}$Cs is regarded as one of the most abundant artificial radionuclides in the ocean because of its long half-life (30.17 yr) and large fission yield that originates from large-scale atmospheric weapons tests due to atmospheric nuclear weapon tests by the United States and Russian Federation. Atmospheric nuclear weapons tests occurred from 1945 to 1980. During 1945 to 1963, the large scale atmospheric nuclear weapons tests were conducted by the United States. In 1963, the Partial Nuclear Test Ban Treaty was signed and these tests in the atmosphere by the United States and Soviet Union, and Great Britain shifted to the underground. However, France continued the atmospheric test until 1974 and China until 1980. In addition, $^{137}$Cs has been released into the Pacific Ocean by local fallout from ground tests (e.g., UNCEAR 2000; Aoyama et al., 2006; Aoyama, 2010; Inomata, 2010) on Bikini Atoll in the Marshall Islands between 1946 and 1958 by the United States. Because the $^{137}$Cs released into the atmosphere fallout onto the ocean surface, the ocean is recognized as the largest receptor of $^{137}$Cs on Earth. Furthermore, other sources, such as the accidental release from nuclear facilities (the Three Mile Island nuclear power plant in 1979), sea dumping of nuclear wastes from nuclear facilities carried out in 1986 in the north–central East Sea/Japan Sea by the former Soviet Union and Russian Federation, lost nuclear weapons, and the use of radioisotopes in human activities, such as industry, medicine, and science, are recognized. These contributions in the environment are minor compared to those from the dominant sources listed above (UNSCEAR, 2000; IAEA, 2005).

- Lines 54-55, Pg.2:

Elaborate in more detail what the local fallouts represent.

A. Lines 37-39, Pg.2:
In addition, $^{137}$Cs has been released into the Pacific Ocean by local fallout from ground tests on Bikini Atoll in the Marshall Islands between 1946 and 1958 by the United States (e.g., UNSCEAR 2000; Aoyama et al., 2006; Aoyama, 2010; Inomata, 2010).

- Lines 80-83, Pg. 3:
-Reviewer is confused with authors statement such that the atmospheric deposition of 137Cs into the ocean was estimated to be 11.7-14.8 PBq (Aoyama et al., 2016b) and that the 137Cs inventory into the North Pacific Ocean was estimated to be 15.2-18.3 PBq (Aoyama et al., 2016b, Inomata et al., 2016; Tsubono et al., 2016). Clear description is required.

A. Atmospheric deposition of $^{137}$Cs into the ocean mean that $^{137}$Cs deposition amount from the atmosphere in the ocean surface. The $^{137}$Cs inventory means that $^{137}$Cs existed amount into the surface mixed layer. In order to clear these mean, the author modified as follows.

Line 76-83, Pages 3

The released 137Cs amount by the F1NPS accident and these distributions were investigated by numerous researches and summarized in Busseler et al. (2017). In this study, we used our estimation because these were considering the mass balances among atmosphere, land, and ocean. The atmospheric deposited amount of 137Cs into the ocean from the atmosphere was estimated to be 11.7-14.8 PBq (Aoyama et al., 2016b). Directly discharged liquid 137Cs from the F1NPS was estimated to be 3.6 ± 0.7 PBq by using the observation data around the F1NPS and model simulation (Tsumune et al., 2012, 2013). The 137Cs inventory into the North Pacific Ocean in the surface mixed layer was estimated to be 15.2-18.3 PBq (Aoyama et al., 2016b), which are consistent with the estimated values by optical statistical analysis (Inomata et al., 2016) and model simulation (Tsubono et al., 2016).

● Regarding the statement "directly discharged liquid 137Cs from the F1NPS was estimated to be 3.6 ± 0.7 PBq (Tsumune et al., 2012, 2013)", comments that there are various estimates in literature need to be added.

A. P76-81, Page 3

The released $^{137}$Cs by the F1NPS accident were investigated by numerous researches and summarized in Busseler et al. (2017). However, we used the values by considering the mass balances among atmosphere, land, and ocean. The atmospheric deposited amount of $^{137}$Cs into the ocean from the atmosphere was estimated to be 11.7-14.8 PBq (Aoyama et al., 2016b). Directly discharged liquid 137Cs from the F1NPS was estimated to be 3.6 ± 0.7 PBq by using the observation data around the F1NPS and model simulation (Tsumune et al., 2012, 2013). The 137Cs inventory into the North Pacific Ocean in the surface mixed layer was estimated to be 15.2-18.3 PBq

● Lines 148-151, Pg. 5:

-It is not clear what the statement "latitudinal and longitudinal distributions, and the locations of global fallout, reprocessing plants, and nuclear power plants" represents. "latitudinal and longitudinal distributions of global fallout, and the locations of reprocessing plants, and nuclear power plants"??

A. Actually, distribution of $^{137}$Cs are controlled by the ocean current and location of point source such as the reprocessing plants. Because $^{137}$Cs exists as water soluble, spatial distribution and temporal variation of $^{137}$Cs are related with these horizontally and vertically, and sources. This part was modified as, "The boxes were determined based on the ocean current and location of sources." In addition, author added the location and feature of each box related with ocean current in more detail.

Lines 152-190, Pg. 6:

[revised manuscript text omitted]

●   -Spatial distribution of Cs137 concentrations is uniform? In each box?

A. Measurement data are very limited in the global ocean. In order to investigate the distribution of [137]Cs activity concentrations in global scale, we made statistically median data for every 0.5 years in each box. "Spatial distribution" is deleted. Setting of each box was based on the dominant ocean current. Therefore, the [137]Cs activity concentrations data in each box is regarded as the same oceanographic environment.

P152-157. Page, 5.

Measured [137]Cs data, however, was very limited and it is impossible to cover the distribution of [137]Cs in the global ocean. In this study, the global ocean was divided into 37 boxes to investigate the temporal variations in [137]Cs activity concentrations in surface seawaters by using the available almost all data (Inomata and Aoyama, 2022a) (Figure 1). These boxes were divided by showing the latitudinal and longitudinal distributions based on the known ocean currents (IAEA, 2005; Open University, 2004), the latitudinal distributions of global fallout, location of reprocessing plants and F1NPS under the assumptions that [137]Cs activity concentrations in the box is almost same (Hirose et al., 2003; Inomata et al., 2009; IAEA, 2005).

- -Horizontal and vertical transport of ocean water is almost same? In each box? It is needed to elaborate in detail the basis of this assumption somewhere in this manuscript. There are several questions on the assumption. What is the assumption in shallow marginal seas? Are the horizontal and vertical transports of ocean water time-invariant or time-varying? If each box is further divided into two, the assumption such that horizontal and vertical transports of ocean water are almost same is still valid?

A. Measured [137]Cs data was very limited and it is impossible to cover the distribution of [137]Cs in the global ocean unlike the remote sensing data and/or oceanographic physical parameter data. Model simulation is also effective tool to investigate the spatial and temporal variation of trace species such as [137]Cs. However, the results of model simulation also include large uncertainty. In the manuscript, we explain the feature of [137]Cs activity concentrations in each box associated with ocean current. Marginal Seas such as Japan Sea are based on the definition of IHO (1953). In the boxes divided into subboxes, the box received significant large discharge such as Irish Sea and northern North Pacific Ocean by reprocessing plants, western North Pacific Ocean by the F1NPS accident. The sea area and mixed layer depth were also calculated for each sub box.

Line 152-190, Page 5-6.
Measured [137]Cs data, however, was very limited and it is impossible to cover the distribution of [137]Cs in the global ocean unlike the remote sensing data and/or weather data. In this study, the global ocean was divided into 37 boxes to investigate the temporal variations in [137]Cs activity concentrations in surface seawaters (Inomata and Aoyama, 2022a) (Figure 1). These boxes were divided based on the known ocean currents (IAEA, 2005; Open University, 2004), location of reprocessing plants and F1NPS under the assumptions that [137]Cs activity concentrations in the box is almost same and sources of [137]Cs are established (Hirose et al., 2003; Inomata et al., 2009; IAEA, 2005). Marginal Seas such as Japan Sea are based on the

definition of IHO (1953). The temporal variation of $^{137}$Cs activity concentrations in the surface seawater in each box were investigated by using the available almost all data. The box divided by showing the latitudinal and longitudinal distributions, oceanographic parameters, and the latitudinal distributions of global fallout.Subarctic ocean (north 40°N) is the highest atmospheric deposition of $^{137}$Cs occurred in the 1960s in the Pacific Ocean, western North Pacific Ocean and eastern North Pacific Ocean (25-40°N) are upstream and downstream of Kuroshio extension. These three regions are influenced the $^{137}$Cs contamination derived from the F1NPS accident. Subtropical western and eastern North Pacific Ocean (5-25°N) are downstream and upstream of the north Equatorial Current associated with the subtropical Gyre. Subtropical western and eastern North Pacific Ocean includes the California Gyre. These boxes include the contamination of local fallout such as the Bikini Atoll. Western and eastern equatorial Pacific Ocean are downstream and upstream of the South Equatorial Current. And upwelling od sweater occurs in the eastern Southern Pacific Ocean. Subtropical western and eastern North Pacific Ocean are down stream and upstream of the weak South Equatorial Current. Eastern subtropical South Pacific Ocean includes the French nuclear weapons test sites. The eastern Southern Pacific Ocean (25-40°S) is Tasmania Sea. Eastern South Pacific Ocean (25-40°S) is mid-latitude region of the South Pacific Ocean and includes South Pacific Current. Eastern Southern Ocean (40-60°S) is affected by the Antarctic Circumpolar Current. Antarctic Ocean (below 60°S) are divided into three; Antarctic sector for Pacific, Indian, and Atlantic and locate the polar front and continental water boundary. Middle Southern Ocean is connected to the Indian sector of the Southern Ocean. Indian Ocean is connected to Indonesian Archipelago by Indonesian through flow and also connected to the Arabian Sea. In the marginal seas of the North Pacific Ocean, South China Sea, Eastern China Sea, Japan Sea, and Okhotsk Sea are classified. The Eastern China Sea influence of the bifurcation of Kuroshio Current and downstream of western North Pacific Ocean and connected to the Japan Sea via Tsushima Warm Current. The northward transported seawater in the Japan Sea is connected to the Sea of Okhotsk. The Bering Sea is downstream of the subarctic North Pacific Ocean and upstream of the Arctic Ocean. The Atlantic Ocean was divided into three, South Atlantic Ocean (60°S-30°S), Central Atlantic Ocean (30°S-15°N), and North Atlantic Ocean (15°N-45°N). The South Atlantic Ocean is connected with the Southern Ocean. The Irish Sea and English Chanel are considered as the $^{137}$Cs direct discharged region. The North Sea, Barents Sea and Coast of Norway, Baltic Sea, Arctic Ocean are down stream of the northern North Atlantic Ocean and affected the inflow the $^{137}$Cs derived from the Irish Sea and English Chanel. The northern North Atlantic Ocean received $^{137}$Cs global fallout by the large scale weapons tests in the 1950s and 1960s. The Baltic Sea, the Mediterranean Sea, the Black Sea received the fallout of $^{137}$Cs from the Chernobyl accident.

- Lines 148-160, Pg. 5:

Elaborate why the box configuration is changed.

A. Taking into account the ocean current, these boxes (such as Box13, 36,37, 17, 19) were
   further divided. The location of Box and its ocean current, and seawater flow pattern are
   explained in the manuscript related with the above question. In Box2, 23, and 25, we set
   several sub-boxes, because the significantly larger values cause the larger $^{137}$Cs values in
   the box.

Line, 198-201, Page 7.

The boxes corresponding to the source region, such as the Irish Sea (Box 23; Boxes 23.1-
23.5) for the Sellafield plant and the northern North Atlantic Ocean (Box 25; Boxes 25.1 and
25.2) and western North Pacific Ocean (Box 2; Boxes 2.0-2.6) for the F1NPS accident, were
divided into several sub regions, because significantly large values around the dischared
region cause to larger values to estimate the $^{137}$Cs inventory.

**2. Data and methods:**

- Lines 193-194, Pg. 10:

Authors state that the currents and major source of Cs137 in the surface water has a 0.5-year
time interval. It is not clear what "the currents" and the major source? Describe the detailed
information. One more thing, are the 5-year interval data instantaneous values or some mean
values?

A. When we calculate the values in t-yr, the data within t±0.5 yr were used to the calculate
   median value. The t year interval data is instantaneous values at the target year.

Lines 248-251, Pg. 11

The 0.5-yr median values of the surface $^{137}$Cs concentrations in each box were produced by
the grid value producing command of block median programs (Wessel et al., 2013). The block
median reads the arbitrary data (x, y, z) and calculates the median value in a grid defined in

the setting range. In the case of t-year, the data within t±0.5 years were used to calculate the median values.

- Lines 217, Pg. 11:

 is used without definition and without physical meaning (Eq. (3) should appear in advance)

A. Definition of Tap is firstly described.

Lines 269-289 (This part was shown in the red and blue text because this part was related with the reviewer 1's comment as red and blue color text), Pg. 12:

The apparent half time (Tap) of $^{137}$Cs was calculated using the following equations:

$$^{137}Cs = {}^{137}Cs_0 exp(-\lambda cs, apparent) \tag{1}$$

$$\lambda Cs, apparent = \lambda Cs, ocean + \lambda Cs, decay \tag{2}$$

$$Tap = 0.693/(\lambda Cs, apparent) \tag{3}$$

$$Tpo = 0.693/(\lambda Cs, ocean) \tag{4}$$

where $\lambda_{Cs,apparent}$, $\lambda_{Cs,ocean}$, and $\lambda_{Cs,decay}$ are the decay constants for apparent decay, physical oceanographic decay, and radioactive decay, respectively. $\lambda_{Cs,apparent}$ is estimated by using the regression line of the 0.5-yr median value of $^{137}$Cs as shown in (1). Tpo is the apparent half residence time by causing the oceanic physical processes and $\lambda_{Cs,ocean}$ was estimated $\lambda_{Cs,apparent}$ and $\lambda_{Cs,decay}$ in equation (2). Considering that the half-life of $^{137}$Cs ($T_{1/2}$) is 30.17 years, the Tap should be shorter than the half-life if no source of $^{137}$Cs exists in the region of interest. A shorter Tap means that $^{137}$Cs is removed quickly in the area and/or the $^{137}$Cs inflow amount is small in the area compared with the $^{137}$Cs outflow amount. In other words, a Tap shorter than the radioactive decay time indicates that the variations in the $^{137}$Cs activity concentrations are strongly controlled by physical ocean processes. In contrast, a longer Tap as well as a negative Tpo value means that $^{137}$Cs is preserved in the region for a longer time and/or there is an influx of water mass with higher $^{137}$Cs in the region compared to the $^{137}$Cs outflow from the region.

However, the exponentially decreasing trend from 1970 to 2010, before the F1NPS accident, did not estimate for all boxes. Tap from 1970 to 2010 were estimated for the western North Pacific Ocean, Japan Sea, and Eastern China Sea. For other boxes, Tap, therefore, was

estimated for several periods, taking into account the source contribution as follows. Tap1 is before 1970 (periods with nuclear weapon tests at a global scale), Tap2 is the period from 1970 to 1986-1990 (until the Chernobyl accident), Tap3 is from 1990 to 2010 (after the Chernobyl accident), and Tap4 is after 2011 (after F1NPS accident). There were some regions, where did not estimate to Tap, such as the northern North Atlantic and surrounding waters, because decreasing trend of $^{137}$Cs could not be approximated by Equation (1).

- Lines 221-224, Pg. 10:

-What is the definition of ?

-What is the definition of Tpo and its physical meaning?

A. Tpo is physical oceanographic apparent half residence time. In equation (1)-(4), decreased $^{137}$Cs activity concentrations are controlled by radioactive decay and oceanographic physical processes.

$$^{137}Cs = {}^{137}Cs0exp(-\lambda cs, apparent) \qquad (1)$$

$$\lambda Cs,apparent = \lambda Cs,ocean + \lambda Cs,decay \qquad (2)$$

$$Tap = 0.693/(\lambda Cs,apparent) \qquad (3)$$

$$Tpo = 0.693/(\lambda Cs,ocean) \qquad (4)$$

The longer Tap and negative values of Tpo mean that $^{137}$Cs is flowed into the downstream region to the upstream region.

Line 278-281 (red text). Pages 12

A shorter Tap means that $^{137}$Cs is removed quickly in the area and/or the $^{137}$Cs inflow amount is small in the area compared with the $^{137}$Cs outflow amount. In other words, a Tap shorter than the radioactive decay time indicates that the variations in the $^{137}$Cs activity concentrations are strongly controlled by physical ocean processes. In contrast, a longer Tap as well as a negative Tpo value means that $^{137}$Cs is preserved in the region for a longer time and/or there is an influx of water mass with higher $^{137}$Cs in the region compared to the $^{137}$Cs outflow from the region.

- Lines 279-280, Pg. 13:

Authors state that the maximum monthly mixed layer depth was used because Cs137 is easily transported to the subsurface under deeper mixed layer. Hard to understand why easier subsurface transport is necessary. How about using mean monthly mixed layer depth?

A. Line 330-334, Page 13

The mixed layer depth was the monthly time interval with seasonal variation that is deeper in winter and shallower in summer. It is recognised that sea water subducted from the ocean surface in the mode water formation region associated with the winter convective mixing because of the lower buoyancy from the ocean surface (Hanawa and Tally, 2001). The flow through the winter mixed layer ventilate the sea water into the ocean interior. The maximum monthly mixed layer depth in each box, mainly winter month, was used to calculate the $^{137}$Cs inventory in the mixed layer.

- Lines 293, Pg. 13:

Authors described "horizontal" transport as "outflow to the downstream box" transport. Reviewer thinks it is incorrect because there can be inflow-related transport. "horizontal (net outflow to the downstream box) transport" needs to be used.

A. Line 342-343, Pages 14;

In the marine environment, $^{137}$Cs activity concentrations after 1970 were dominantly controlled by radioactive decay and physical ocean processes, such as horizontal, which mean net outflow to the downstream box for sea water current U and V component, and downwards transport below the surface mixed layer for seawater current W component.

- Lines 296-302, Pg. 13:

-Subscripts in $C_{i, box}$ and $C_{0, box}$ in (8) are a little bit confusing. Better to use notations with two subscripts (including box number I and time).

A. $C_{box,t1}$, $C_{box,t0}$ were used in the equation.

Line 347-353, Page 14

$[^{137}\text{C inventory}_{box, t1}] = [^{137}\text{Cs inventory}_{box, t0}] - [^{137}\text{Cs inventory}_{box, t0} \times \exp(-0.693/T_{1/2} \times \Delta t)] - [\text{net outflow to the downstream box of } CI_{box, t0}] - [\text{downwards transport of } ^{137}\text{Cs inventory}_{box, t0} \text{ below the mixed layer}]$ (8)

where $^{137}$Cs inventory $_{box, t0}$ is the $^{137}$Cs inventory by using 0.5-yr $^{137}$Cs average value and mixed layer depth in each box in the initial year and $^{137}$Cs inventory $_{box, t1}$ is the $^{137}$Cs inventory by using 0.5-yr $^{137}$Cs average value in each box after the $\Delta t$ year. This mass balance was estimated to every 5 years from 1975 to 2015. In the case of 1970, the value of the initial year in each box was $^{137}$Cs deposition amount until 1970. In fact, distinguishing between net outflow to the downstream box and downwards-transported $^{137}$Cs amounts was very difficult in this study.

- Describe how the "initial year" is defined.

Line 350-352, Page 14

This mass balance was estimated to every 5 years from 1975 to 2015. In the case of 1970, the value of the initial year in each box was $^{137}$Cs deposition amount until 1970. In fact, distinguishing between net outflow to the downstream box and downwards-transported $^{137}$Cs amounts was very difficult in this study.

- To reviewer's knowledge, the transport is composed of advective and diffusive fluxes. Explain which data were used for the fluxes. If only advective flux was used, elaborate which current (u,v and w) data were used. Furthermore, authors mentioned that distinguishing the horizontal and downward transports were difficult. Explain why. No w velocity?

A. Actually, seawater transport was controlled by ocean current. And seawater speed and direction consists with u,v, and w. However, in this study, I did not analysis of ocean current, because of the data is discrete. I mean that "the net outflow to the downstream box" correspond to u and v component, and "downward transport below the surface sea water" correspond to w component. "The net outflow to the downstream box" is controlled by advective and diffusive fluxes.

Line 342-346 Page 14.

In the marine environment, $^{137}$Cs activity concentrations after 1970 were dominantly controlled by radioactive decay and physical ocean processes, such as horizontal, which mean net

outflow to the downstream box for sea water current U and V component, and downwards transport below the surface mixed layer for seawater current W component, except for the contribution from accidental release (the Chernobyl accident in 1986 and the Fukushima accident in 2011) and direct discharge from nuclear reprocessing plants.

**3. Results**

- Line 308, Pg. 13:

Authors state that correlation coefficient is between 0.51 and 1.0. It appears that the range is very large. Elaborate why. And clarify the correlation coefficient between what?

A. Additional data against to the previous analysis were caused to the difference of the average data. However, this is not an essential issue in this manuscript. I delete this description.

- Lines 561-562, Pg. 28:

Describe why the model results are considerably different from the estimate by Aoyama et al. (2006) and this study.

Line 577-584 Page 28.

However, these estimations are almost 1.4 times larger than those in the estimation by using a model simulation (UNSCEAR, 1993), with an estimated value of 545 PBq (Aoyama, 2019). The large difference in the meridional distribution in the mid-latitude. These corresponds to have larger 137Cs fallout region, where Kuroshio Current and its extension areas (latitude 20-40°N) in the Pacific Ocean and Gulf stream transport area (latitude 30-50°N) in the Atlantic Ocean. It was also reported that the 137Cs water column inventory in the North Pacific Ocean was 2-3 times larger than those in the cumulative 137Cs fallout amount in the same latitude in the modelling results in UNSCEAR (1993) (Aoyama, 2019). Because reconstructed 137Cs deposition in Aoyama et al. (2006) was based on the historical observed data, uncertainty of model would cause the underestimation of 137Cs deposition amount.

- Lines 630-631, Pg. 30:

Better to use 0.0 PBq instead of 0 PBq.

Lines 613-614, Pg. 30:

The $^{137}$Cs deposition amount is the lowest in the Pacific sector (0.05 PBq), Atlantic sector (0.0 PBq), and Indian sector (0.0 PBq) of the Antarctic Ocean.

● Lines 770, 785, Pg. 37:

Hard to see the max. value of scale bars.

These          Figures          modified          as          follows:

[Figure]

● Lines 923-924, Pg. 48:

The mixed layer depth can have strong seasonal variability in some marginal seas where

vertically well mixed in winter while stratified in summer. Elaborate how authors deal with.

A. The monthly mixed layer depth shows seasonal variation that is deeper in winter and shallower in summer. It is recognized that sea water subducted from the ocean surface in the mode water formation region associated with the winter convective mixing because of the lower buoyancy from the ocean surface (Hanawa and Tally, 2001). The flow through the winter mixed layer ventilate the sea water into the ocean interior. The maximum monthly

mixed layer depth in each box was used to calculate the $^{137}$Cs inventory in the mixed layer. In fact, $^{137}$Cs activity concentrations were increased after 1-2 years at the F1NPS accident in the Eastern China Sea and the Japan Sea. It appeared that $^{137}$Cs released into the F1NPS accident deposited in the subtropical mode water formation region in the North Pacific Ocean, and then subducted into the subsurface layer, following then transported westward, and reached to the bottom of the Eastern China Sea. The F1NPS derived $^{137}$Cs were upward transport into the surface seawater and then transported into the Japan Sea (Inomata et al., 2018a,b). Therefore, maximum mixed layer depth associated with the mode water formation was used in this study.

Line 329-333
The mixed layer depth was the monthly time interval with seasonal variation that is deeper in winter and shallower in summer. It is recognised that sea water subducted from the ocean surface in the mode water formation region associated with the winter convective mixing because of the lower buoyancy from the ocean surface (Hanawa and Tally, 2001). The flow through the winter mixed layer ventilate the sea water into the ocean interior. The maximum monthly mixed layer depth in each box, mainly winter month, was used to calculate the $^{137}$Cs inventory in the mixed layer.

- Lines 984-986, Pg. 51:

Authors state that the $^{137}$Cs discharged into the Irish Sea.1 was transported into the Irish Sea.2, followed by transport to the northern North Atlantic Ocean.1, North Sea, and Barents Sea and coast of Norway. Reviewer is interested in its transport direction in Irish Sea. To reviewer's knowledge, simulation by Prandle (1983) showed that Cs137 moved to the north channel of Irish Sea. How about in this study? Discuss this matter.

A.  Actually Irish Sea.2 area include the north channel of Irish Sea. Also this $^{137}$Cs transport pattern analyzed in this study is consistent with the general circulation of seawater transport by Bois et al. (2020).  Unfortunately, I did not find the paper, Prandle (1983).

Line  794-795, Page 51

This pattern is also consistent with general pattern of seawater transport in this region (Bois et al., 2020).

**4. Discussion**

- Line 1259, Pg. 67:

Hard to understand the expression "outflowed 137Cs is larger than the flow of 137Cs". Improvement in writing is required.

Line 932-947

      Fig. 15 shows the spatiotemporal variations in the $^{137}$Cs density in the surface mixed layer in the North Pacific Ocean, subtropical North Pacific Ocean, equatorial Pacific Ocean, and subtropical western South Pacific Ocean. In the western North Pacific Ocean, except for the highest $^{137}$Cs density in 1960, $^{137}$Cs density increased and reached to 1964, and then decreases exponentially. However, in the eastern North Pacific Ocean, the $^{137}$Cs density increased until 1966 and then decreased exponentially. The 2 years timelag that reached the maximum value was caused by horizontal transport from the western North Pacific Ocean and accumulated in the eastern North Pacific Ocean (Inomata et al., 2012). In the subtropical western and eastern North Pacific Ocean, and eastern equatorial Pacific Ocean, the $^{137}$Cs density was almost constant in the 1970s and the 1980s. After the 1990s, the $^{137}$Cs density decreased gradually. In the western equatorial Pacific Ocean and western subtropical South Pacific Ocean, the $^{137}$Cs density increased gradually until the 1980s and then decreased after the 1990s. As shown in Table 3, Tap2 in the eastern North Pacific Ocean, which is estimated to be 8.8 years, is shorter than that in the western North Pacific Ocean (16.9 years). This suggests that the outflowed $^{137}$Cs amount in the eastern North Pacific Ocean was larger than the inflowed 137Cs amount from the western North Pacific Ocean. The Tap2 in the western subtropical North Pacific Ocean is estimated to be 34.1 years and Tpo is estimated to be -260.7 years. This mean that the $^{137}$Cs was accumulated in this region: The seawater with higher $^{137}$Cs activity concentrations moves southwards with subsidence associated with the North Pacific subtropical gyre, followed by westwards transport and subduction in the central and eastern subtropical North Pacific Ocean (Inomata et al.,2012). The increased $^{137}$Cs density in the western equatorial Pacific Ocean and the western subtropical South Pacific Ocean, as shown in Fig. 15, would result in a supply of seawater with higher $^{137}$Cs activity concentrations.

- Lines 1356-1357, Pg. 67:

Regarding the statement "although the contribution of directly discharged 137Cs from the F1NPS might be included in the values in the western North Pacific Ocean", comment when

it happened. 2011? 2012? That is, how long does it take the directly released Cs137 moves from Fukushima NPP to the southern sea region of Kuroshio extension?

A. Direct discharge of 137Cs continued. However, most of discharged 137Cs occurred from 26March to 6 April, 2011, and then decreased (Tsumune, 2013). Because the atmospheric deposited $^{137}$Cs and in flowed $^{137}$Cs did not distinguished in this analysis, this part was deleted.

Line 1041-1042

These were caused by the atmospheric deposition of $^{137}$Cs derived from the F1NPS accident.

● Lines 1399-1357, Pg. 75:

The expression "Cs137 is transported via advection" is not correct. "via advection" needs to be changed to "via advection and diffusion". Authors appear to neglect diffusion, right?

A. I added the tem "diffusion".

Line 1071, Page 75

The $^{137}$Cs deposited into the ocean surface is transported via advection and diffusion in the surface seawater

● Lines 1406-1408, Pg. 75:

Authors mentioned inflow/outflow, upstream box and downstream box. Reviewer hardly understand how such information can be obtained. Up to now authors had never mentioned current data. Clarify this point.

Line 1079-1080, Page 75

Positive values (red) indicate that inflowed $^{137}$Cs, whereas negative values (blue) indicate that outflowed $^{137}$Cs in each area.

- Line 1428, Pg. 75:

Authors mention "considering the current system". What is the source of the current system?

Line 1099-1101, Page 75

Then, these were transported southwards with subsidence associated with California Current and westwards in the equatorial Pacific Ocean. 137Cs moved southwards due to subduction in the eastern subtropical North Pacific Ocean and upwelled in the western/eastern equatorial Pacific Ocean.

- Lines 1011-1714, Pg. 94:

What is represented by "1~2 years" in Fig. 22 shown in North Pacific?

Line 1271-1272, Pg. 93

$^{137}$Cs deposited by the large-scale nuclear weapons tests in the western North Pacific Ocean was transported eastwards within 1-2 years.

Reply

**Citation**: https://doi.org/10.5194/essd-2022-374-RC3

---

## Author Response (AR2)

**Public justification (visible to the public if the article is accepted and published):**

In the referee's opinion, chapter 2.7 contains essential information on this study and is only incompletely described. The author has revised the chapter accordingly. Unfortunately, the new text is insufficiently understandable linguistically. For example, the first sentence from line 340 with various text inserts stretches over 5 lines. The equation can hardly be captured. Now, what is the message of the chapter regarding mass balance and inflow and outflow? Are figures referred to and, if so, which ones? This text needs to be revised again before publication

**Dear Prof. Dagmar Hainbucher**

Thank you very much for your comment and your decision.

I modified the chapter 2.7.

In this mass balance equation, I was estimated the net outflowed $^{137}$Cs from each box. In the surface sea water, $^{137}$Cs activity concentrations (or inventory) are controlled by radioactive decay, inflow from the upstream box, outflow to the downstream box, and downward transport below the surface mixed layer.

$$[^{137}Cs \text{ inventory}]_{box,ti+5} = [^{137}Cs \text{ inventory}]_{box,ti} - [\text{radioactive decayed } ^{137}Cs]_{box,\Delta t} - [\text{net outflowed } ^{137}Cs]_{box,\Delta t} \qquad (8)$$

Because estimates of $^{137}$Cs transport amount in these processes were very difficult in this study, outflowed $^{137}$Cs by these processes were represented as "net outflowed $^{137}$Cs" in each box. In this study, I showed the estimation in every 5 years after 1975 by using the previous 5years inventory value. In the case of 1970, $^{137}$Cs deposition amount until 1970 was used as an initial value.

I modified this chapter as follows.

2.7. Mass balance; inflow and outflow of $^{137}$Cs from each box

In the marine environment, $^{137}$Cs activity concentrations after 1970 were dominantly controlled by radioactive decay and physical ocean processes, except for the release by accident and reprocessing plants. As the physical oceanographic processes, $^{137}$Cs in the surface seawater in each box receive inflow from the upstream box, outflow to the downstream box, and downward transport below the surface mixed layer. In fact, estimates of $^{137}$Cs transport amount in these processes were very difficult in this study. Therefore, outflowed $^{137}$Cs by these processes were represented as net outflowed $^{137}$Cs in each box. Mass balance of $^{137}$Cs in the surface mixed layer was considered as following equations.

$[^{137}\text{Cs inventory}]_{\text{box, ti+5}} = [^{137}\text{Cs inventory}]_{\text{box,ti}} - [\text{radioactive decayed } ^{137}\text{Cs}]_{\text{box, }\Delta t} - [\text{net outflowed } ^{137}\text{Cs}]_{\text{box, }\Delta t}$ \qquad (8)

$[\text{radioactive decay}]_{\text{box, }\Delta t} = [^{137}\text{Cs inventory} \times \exp(-0.693/T_{1/2} \times \Delta t)]_{\text{box}}$ \qquad (9)

$[\text{net outflowed } ^{137}\text{Cs}]_{\text{box, ti+5}} = [\text{inflowed } ^{137}\text{Cs}]_{\text{box, ti}} + [\text{outflowed } ^{137}\text{Cs}]_{\text{box, ti}}$ \qquad (10)
$+ [\text{downwards transport of } ^{137}\text{Cs below the surface mixed layer}]_{\text{box,ti}}$

where,

$\Delta t$ : 5 years

ti : $1970 + i \times 5$ (i=0,1,$\cdots$, 9).

$[^{137}\text{Cs inventory}]_{\text{box, ti}}$ is the value at initial year and $[^{137}\text{Cs inventory}]_{\text{box, ti+5}}$ is the $^{137}\text{Cs}$ inventory after the $\Delta t$ year in each box. In this study, this mass balance was estimated to every 5 years from 1970 to 2015. In the case of 1970, $^{137}\text{Cs}$ deposition amount until 1970 was used as the value of the initial year in each box. In the northern North Atlantic Ocean and Arctic Ocean, an extremely large inflow was estimated in 2000 due to the extremely large values included in the dataset. These data in 2000 and 2005 were removed from the figures.

Thank you very much again. \hfill Yayoi Inomata

---

## Author Response (AR3)

Public justification (visible to the public if the article is accepted and published):
In the referee's opinion, chapter 2.7 contains essential information on this study and is only incompletely described. The author has revised the chapter accordingly. Unfortunately, the new text is insufficiently understandable linguistically. For example, the first sentence from line 340 with various text inserts stretches over 5 lines. The equation can hardly be captured. Now, what is the message of the chapter regarding mass balance and inflow and outflow? Are figures referred to and, if so, which ones? This text needs to be revised again before publication

**Dear Prof. Dagmar Hainbucher**

Thank you very much for your comment and your decision.

I modified the chapter 2.7 and chapter 4.4.

In this mass balance equation, I was estimated the net in/outflowed $^{137}$Cs in each box. In the surface sea water, $^{137}$Cs activity concentrations (or inventory) are controlled by radioactive decay, inflow from the upstream box, outflow to the downstream box, and downward transport below the surface mixed layer.

$[^{137}\text{Cs inventory}]_{\text{box},t_i+5} = [^{137}\text{Cs inventory}]_{\text{box},t_i} - [\text{radioactive decayed } ^{137}\text{Cs}]_{\text{box},\Delta t} - [\text{net in/outflowed } ^{137}\text{Cs}]_{\text{box},\Delta t}$    (8)

Because estimates of $^{137}$Cs transport amount in these processes were very difficult in this study, outflowed $^{137}$Cs by these processes were represented as "net in/outflowed $^{137}$Cs" in each box. In this study, I showed the estimation in every 5 years after 1975 by using the previous 5years inventory value. In the case of 1970, $^{137}$Cs deposition amount until 1970 was used as an initial value.

I modified these chapters as follows.

The modified parts were described by green (for ver 5) and purple (for ver 6) colors text.

**2.7. Mass balance; net in/outflow of $^{137}$Cs in each box**

In the marine environment, $^{137}$Cs activity concentrations after 1970 were dominantly controlled by radioactive decay and physical ocean processes, except for the release by accident and reprocessing plants. As the physical oceanographic processes, $^{137}$Cs in the surface seawater in each box receive inflow from the upstream box, outflow to the downstream box, and downward transport below the surface mixed layer. In fact, estimates of $^{137}$Cs transport amount in these processes were very difficult in this study. Therefore, outflowed $^{137}$Cs by these

processes were represented as net in/outflowed $^{137}$Cs in each box. Mass balance of $^{137}$Cs in the surface mixed layer was considered as following equations.

$$[^{137}\text{Cs inventory}]_{\text{box, ti+5}} = [^{137}\text{Cs inventory}]_{\text{box,ti}} - [\text{radioactive decayed } ^{137}\text{Cs}]_{\text{box, }\Delta t} - [\text{net in/outflowed } ^{137}\text{Cs}]_{\text{box, }\Delta t} \quad (8)$$

$$[\text{radioactive decay}]_{\text{box, }\Delta t} = [^{137}\text{Cs inventory} \times \exp(-0.693/T_{1/2} \times \Delta t)]_{\text{box}} \quad (9)$$

$$[\text{net in/outflowed } ^{137}\text{Cs}]_{\text{box, ti+5}} = [\text{inflowed } ^{137}\text{Cs}]_{\text{box, ti}} + [\text{outflowed } ^{137}\text{Cs}]_{\text{box, ti}} \quad 10)$$
$$+ [\text{downwards transport of } ^{137}\text{Cs below the surface mixed layer}]_{\text{box,ti}}$$

where,

$\Delta t$ : 5 years

ti : 1970+i×5 (i=0,1,···, 9).

[revised manuscript text omitted]

Thank you very much again.                                              Yayoi Inomata